# Lactylation of METTL16 promotes cuproptosis via m⁶A-modification on *FDX1* mRNA in gastric cancer

Lianhui Sun[1,2,7], Yuan Zhang ®[1,7], Boyu Yang[3,7], Sijun Sun[1,7], Pengshan Zhang[1], Zai Luo[1], Tingting Feng[4], Zelin Cui ®[5], Ting Zhu[2], Yuming Li[6], Zhengjun Qiu[1], Guangjian Fan ®[2] ✉ & Chen Huang ®[1] ✉

Cuproptosis, caused by excessively high copper concentrations, is urgently exploited as a potential cancer therapeutic. However, the mechanisms underlying the initiation, propagation, and ultimate execution of cuproptosis in tumors remain unknown. Here, we show that copper content is significantly elevated in gastric cancer (GC), especially in malignant tumors. Screening reveals that METTL16, an atypical methyltransferase, is a critical mediator of cuproptosis through the m⁶A modification on *FDX1* mRNA. Furthermore, copper stress promotes METTL16 lactylation at site K229 followed by cuproptosis. The process of METTL16 lactylation is inhibited by SIRT2. Elevated METTL16 lactylation significantly improves the therapeutic efficacy of the copper ionophore– elesclomol. Combining elesclomol with AGK2, a SIRT2-specific inhibitor, induce cuproptosis in gastric tumors in vitro and in vivo. These results reveal the significance of non-histone protein METTL16 lactylation on cuproptosis in tumors. Given the high copper and lactate concentrations in GC, cuproptosis induction becomes a promising therapeutic strategy for GC.

Copper, an essential trace element in organisms, is involved in cell growth and metabolism[1–6]. Copper homeostasis disorder is relevant to various tumors, such as gynecological tumors[7,8] and colorectal cancer[9]. Indeed, increased serum copper levels in cancer patients are correlated with cancer grade and chemotherapy resistance[10]. Copper becomes toxic if its concentration exceeds a certain threshold, inducing a newly discovered form of cell death, named cuproptosis[11]. Cuproptosis occurs when copper binds directly to lipoylated components of the tricarboxylic acid cycle, leading to subsequent loss of iron-sulfur cluster-containing proteins, proteotoxic stress and ultimately cell death. A recent study using genome-wide CRISPR-Cas9 loss-of-function screening identified ten critical factors of cuproptosis[11]. Among them, ferredoxin 1 (FDX1) encodes a reductase known to reduce $Cu^{2+}$ to its more toxic form $Cu^{1+}$, lipoylates dihydrolipoamide S-acetyltransferase (DLAT) that is essential for cuproptosis[12], and is the direct target of elesclomol that is an important copper ionophore[13]. However, the mechanisms underlying the initiation, propagation, and execution of cuproptosis in tumors remain unknown.

RNA modification, as an epigenetic regulator, has gained increasing attention in recent years[14]. N6-methyladenosine (m⁶A), the

[1]Department of Gastrointestinal Surgery, Shanghai General Hospital, Shanghai Jiao Tong University School of Medicine, Shanghai 201620, China. [2]Precision Research Center for Refractory Diseases, Institute for Clinical Research, Shanghai General Hospital, Shanghai Jiao Tong University School of Medicine, Shanghai 201620, China. [3]Department of Urology, Beijing Friendship Hospital, Capital Medical University, Beijing 100050, China. [4]Department of Clinical Pharmacy, Shanghai General Hospital, Shanghai Jiao Tong University School of Medicine, Shanghai 201620, China. [5]Department of Laboratory Medicine, Shanghai General Hospital, Shanghai Jiao Tong University School of Medicine, Shanghai 201620, China. [6]Department of Critical Care Medicine, Shanghai General Hospital, Shanghai Jiao Tong University School of Medicine, Shanghai 201620, China. [7]These authors contributed equally: Lianhui Sun, Yuan Zhang, Boyu Yang, Sijun Sun. ✉e-mail: gjfan@shsmu.edu.cn; richard-hc@hotmail.com

most abundant RNA modification in eukaryotic cells, plays a crucial role in tumorigenesis[15,16]. The expression of regulatory proteins associated with m6A is abnormal in most tumors, leading to drug resistance and tumor development[17,18]. METTL16, followed by classic METTL3/METTL14 methyltransferases[19,20], was recently identified as a second m6A writer[21]. Previous studies have demonstrated that METTL16 is responsible for m6A deposition in many transcripts, including the MAT2A transcript that encodes SAM synthetase[22] and U6 snRNA[23]. METTL16 shows tumorigenesis and tumor-promoting capabilities in an m6A-dependent manner in multiple tumors[24–29], while the regulatory mechanisms controlling METTL16 activity remain unclear. A comprehensive study about the regulation and function of METTL16 may therefore provide better insights into the prevention of tumor therapy and metastasis.

Recently, lactate-derived lactylation has been identified as a newly discovered post-translational modification (PTM)[30]. Lactate is taken up by tumor cells and transported to mitochondria for oxidation to provide energy[31], meanwhile derives lactylation of histone lysine (K) residue to stimulate gene transcription[30]. Histone lactylated modification has been proven to involve multiple pathological processes, such as macrophage polarization[30,32] and tumorigenesis[33]. Delactylases and lactyltransferases are also gradually being discovered. Delactylases mainly include SIRT1-3 and HDAC1-3[34–36] while lactyltransferases mainly include P300, MOF, ME-PCT, and RE-PCT[30,36,37] The abundance and specificity of non-histone proteins in cells yet were higher than that of histones. Whether there are numerous lactylated modifications on non-histone proteins and how these lactylated non-histone proteins operate and are regulated in tumor progression are urgently needed to be explored[38].

In this study, we found that copper content is significantly elevated in gastric cancer (GC), especially in malignant tumor types. Screening revealed that METTL16 is a critical mediator of cuproptosis in GC through the m6A modification on *FDX1* mRNA. Interestingly, we found that high copper content promotes non-histone protein METTL16-K229 lactylation through increasing the interaction of potential lactyltransferases AARS1 / AARS2 to METTL16, and ultimately leads to cuproptosis. In addition, delactylases SIRT2 inhibits the process of METTL16 lactylation. Given that gastric tumors (especially malignant tumor types) have higher copper and lactate concentrations than normal tissues, combined treatment with copper ionophore–elesclomol and SIRT2-specific inhibitors to trigger cuproptosis obviously improves the efficacy of gastric cancer treatment. The study findings provide insights into the mechanisms underlying the initiation and execution of cuproptosis and suggest a promising therapeutic strategy for GC.

## Results

### Copper content is high in gastric cancer and related to tumor progression

Cuproptosis is caused by excessively high copper concentration, but the regulatory mechanisms of cuproptosis in tumors still need to be explored[11]. Given copper is mainly absorbed through the stomach and upper small intestine, tumors of the gastrointestinal system are suitable for studying the initiation and propagation of cuproptosis. We analyzed copper concentrations in 48 pairs of GC tissues and adjacent tissues, finding higher copper concentrations in GC tissues than in normal gastric tissues (Fig. 1a, $P < 0.01$). Further analysis revealed that the relative copper content in patients with stage III GC was higher than that in patients with stage I and II GC (Fig. 1b, $P < 0.05$), indicating that copper content is correlated with tumor progression. Additionally, the copper content was negatively correlated with the overall survival (OS) and disease-free survival (DFS) in patients with GC (Fig. 1c, d; log-rank $P < 0.05$). Meanwhile, the copper content was positively correlated with Ki-67 expression, an indicator of cell proliferation (Fig. 1e). Similarly, the copper content was positively

correlated with platelet/lymphocyte and neutrophil/lymphocyte ratios in serum specimens of patients, both of which are inflammation-related prognostic biomarkers for invasive malignancy for GC (Fig. 1f, g). These results indicated that the high content of copper is involved in the progression and development of GC.

Additionally, we examined the differences of copper concentrations in different GC types, especially in malignant tumor types. Intriguingly, we discovered higher copper concentrations in mucinous adenocarcinomas than in overall gastric tumors (Fig. 1h). The copper content was positively correlated with invasive indicators such as lymph node metastasis (Fig. 1i) and vascular invasion (Fig. 1j) in mucinous adenocarcinomas. However, the copper content was not significantly associated with these invasive indicators or tumor differentiation in non-mucinous adenocarcinoma (Supplementary Fig. 1a–h). Mucinous adenocarcinoma, a rare histological subtype of GC, is associated with a more advanced stage and worse 5-year OS and DFS than non-mucinous adenocarcinoma[39]. As a refractory GC, mucinous adenocarcinoma is characterized by chemoradiotherapy resistance and more effective therapeutic approaches are urgently needed.

Recently, researchers have found that high intracellular copper concentration triggers a reductase FDX1 to reduce $Cu^{2+}$ to a more toxic form $Cu^{1+}$, leading to cuproptosis[11]. We found that FDX1 protein level was higher in GC compared to normal gastric tissues (Fig. 1k). Given both FDX1 protein level and copper content are high in GC, cuproptosis may be more easily triggered. It provides a potential therapeutic strategy for GC, especially for malignant tumors–mucinous adenocarcinomas.

### METTL16 is a critical mediator of cuproptosis

It has been reported that total m6A levels were significantly increased in GC groups compared with benign gastric disease and healthy control patients[40], and promoted the process of GC tumorigenesis, growth, invasion, epithelial mesenchymal transformation (EMT), metastasis, and even multidrug resistance[41–46]. In addition, m6A modification is involved in several cell death types, including apoptosis, necroptosis, ferroptosis and pyroptosis[47]. However, the relationship of m6A modification and cuproptosis remains elusive. Interestingly, we found that copper content was positive correlated with total m6A levels in GC tissues ($R = 0.5116$) (Fig. 2a). Furthermore, copper significantly upregulated total levels of m6A modification in GC cells (Fig. 2b). Given that the methyltransferase-like proteins (METTL) family plays a dominant role in promoting m6A modifications[48], we analyzed the viability of METTLs knockdown cells when treated with elesclomol/ disulfiram and copper in a 1:1 ratio. The results indicated that METTL16 knockdown led to resistance to elesclomol/ disulfiram-Cu treatment (Fig. 2c–e). Moreover, treatment with tetrathiomolybdate (TTM), a cuproptosis inhibitor, suppressed cuproptosis in control and METTL16-knockdown cells, indicating that METTL16 is involved in cuproptosis (Fig. 2f). To further verify these results, METTL16-knockout clone cell lines were successfully constructed and validated (Supplementary Fig. 2c) with similar results observed as in METTL16-knockdown cells (Fig. 2g, h). These results demonstrated that METTL16 plays a crucial role in cuproptosis.

METTL16 is an m6A methyltransferase that modifies several pre- and non-coding RNAs, promotes translation and carcinogenesis, and is an important potential target for cancer treatment[21,29]. The mRNA levels of METTL16 were relatively higher in GC than in peri-tumors through TCGA and GTEx databases (Supplementary Fig. 2d and 2e). METTL16 protein levels were also significantly higher in GC tissues than in normal gastric tissues according to the immunohistochemical (IHC) scores (Supplementary Fig. 2f). METTL16 protein levels had no relevance with TNM staging, pathological patterns, nerve invasion, or vascular invasion (Supplementary Fig. 2g–2j). Interestingly, we found

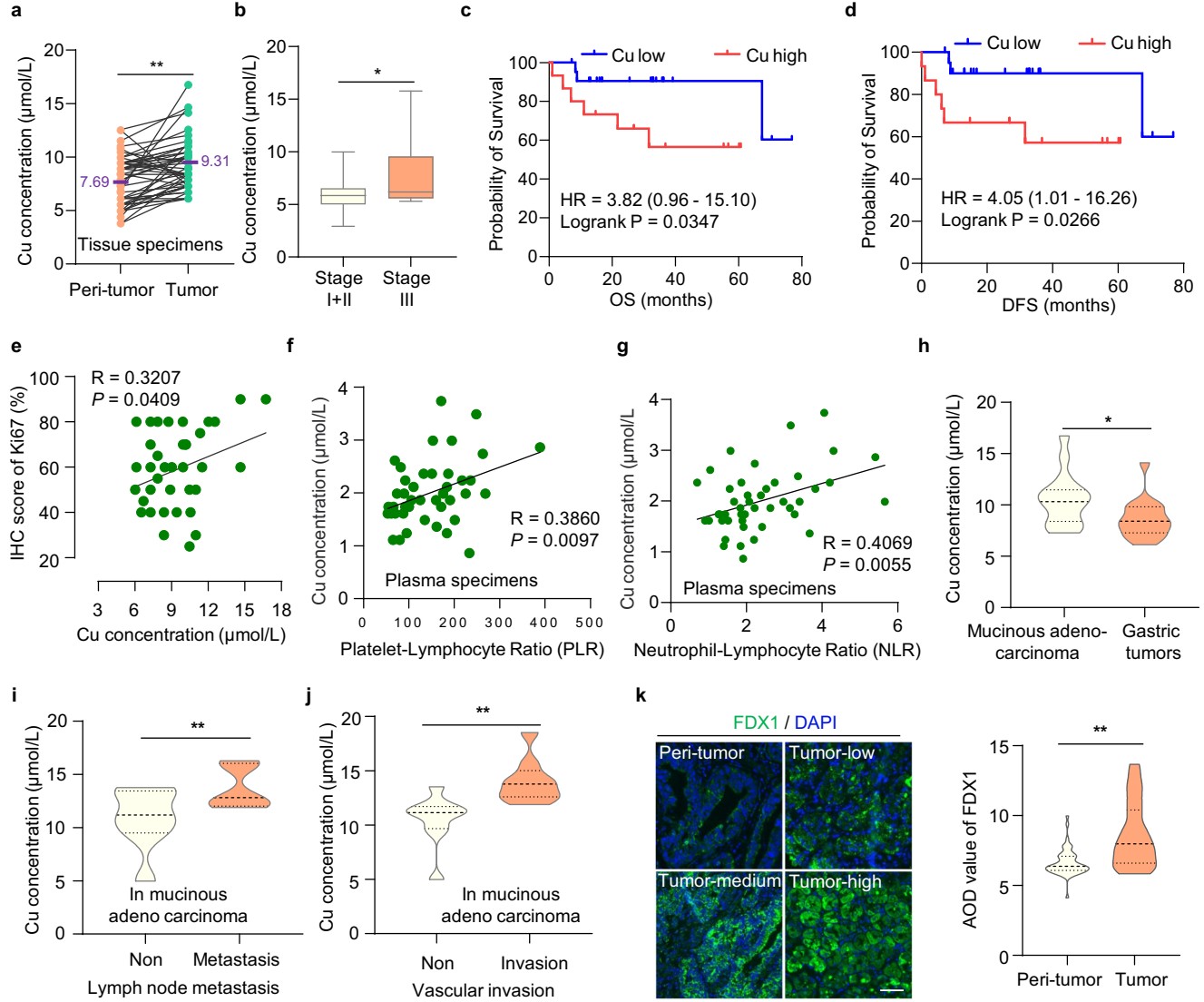

**Fig. 1 | Copper content is high in gastric cancer and related to tumor progression. a**, **b** Statistical graphs of correlations between Cu concentrations and GC tissue samples. **a** Paired line scatter plot showing Cu concentrations in GC (green dot) ($n = 48$) and adjacent normal tissues (orange dot) ($n = 48$). Cu concentrations in GC tissues are much higher than that in adjacent normal tissues. The means were added (Purple font). Paired *t*-test, $P < 0.0001$. **b** Box plot showing relative Cu concentrations between patients with Stage I + II ($n = 42$) and Stage III gastric cancer ($n = 42$). Stage I + II: minima = 2.909595, maxima = 9.987638, mean = 5.944. Stage III: minima = 5.293519, maxima = 15.78, mean = 7.687. whiskers: Min to Max. Unpaired *t*-test, $P = 0.0221$. **c**, **d** Survival analysis of Overall Survival (OS) and Disease-free Survival (DFS) in GC patients. **c** The difference in Overall Survival (OS) between patients with high and low Cu concentrations (HR = 3.82). **d** The difference in disease-free survival (DFS) between patients with high and low Cu concentrations (HR = 4.05). **e** Pearson correlation analysis between Ki-67 expression and Cu concentration ($R = 0.3207$). Simple linear regression. **f**, **g** Correlation between Cu concentrations with pathological indicators in the gastric serum specimens.

**f** Correlation between Cu concentration and platelet to lymphocyte ratio (PLR). Simple linear regression. **g** Correlation between Cu concentration and neutrophil to lymphocyte ratio (NLR). Simple linear regression. **h**–**j** Statistical graphs of correlation between Cu concentrations and pathological characteristics in mucinous adenocarcinoma samples. **h** Difference in Cu concentrations between mucinous adenocarcinoma and overall gastric tumors. $P = 0.0255$. Relative Cu level is positively correlated with lymph node metastasis ($P = 0.0387$) (**i**) and vascular invasion ($P = 0.0072$) (**j**) in mucinous adenocarcinoma. **k** FDX1 expression in gastric tissue microarray. Immunofluorescence staining of the tissue microarray (T16-425) containing 54 pairs of GC and adjacent normal tissues. Representative images showing the FDX1 expression in tumor and peri-tumor tissues. Scale bar, 50 μm. Violin plot showing the difference in the average optical density (AOD) values of FDX1 between gastric tumor and peri-tumor tissues. $P < 0.0001$. Statistical data presented in this figure show mean values ± SD of three times of independent experiments. Statistical significance was determined by Two-tailed *t* test, *$P < 0.05$, **$P < 0.01$. Source data are provided as a Source data file.

that the OS and DFS of gastric cancer patients with high METTL16 protein level was significantly increased (Supplementary Fig. 2k and 2l), which were consistent with those previously reported by Zhang et al.[49], implying that high METTL16 protein level has potential toxic effects on tumors.

## METTL16 promotes cuproptosis by targeting FDX1

METTL16 is responsible for m6A deposition in many transcripts[50]. Hence, methylated RNA immunoprecipitation (MeRIP) sequencing was performed in stable Ctrl-Sh and METTL16-Sh cell lines (Fig. 3a). The metagene plot showed that m6A abundance reduced in METTL16-Sh cells (Supplementary Fig. 3a). The potential m6A modifications in each sample were predicted using HOMER software and m6A modifications mapped primarily to the classical GGAC motif (Fig. 3b). We further analyzed the total m6A distribution patterns of mRNAs according to the MeRIP-seq results and found that m6A peaks were mainly enriched in CDS and 3'UTR regions (Supplementary Fig. 3b). KEGG pathway analysis indicated that genes methylated by METTL16

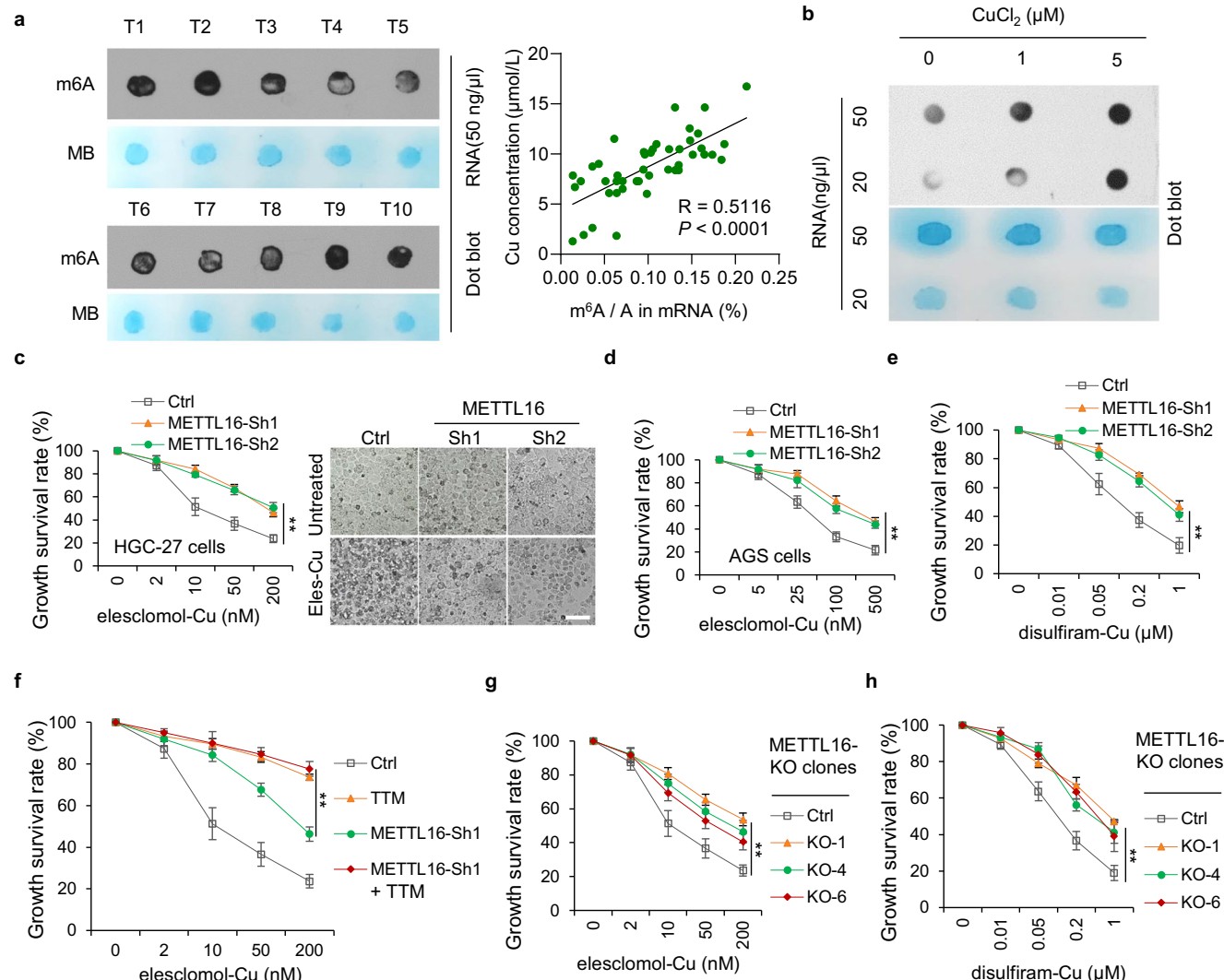

**Fig. 2 | METTL16 is a critical mediator of cuproptosis. a** m6A dot blot assays showing global m6A level of RNAs extracted from 40 pairs of GC tissues. RNAs were serially diluted and loaded equally with the concentration of 50 ng/μl. The methylene blue staining (below) was used to detect input RNA, while the intensity of dot immunoblotting (above) represented the level of m6A modification. Correlation between Cu concentration and m6A modification level in 48 pairs of GC tissues was shown on the right. $P < 0.0001$. **b** Changes in total m6A RNA methylation levels in HGC-27 cells treated with 1 or 5 μM $CuCl_2$. The m6A contents in GC were quantitated by RNA dot blotting. **c–g** METTL16 plays a crucial role in cuproptosis. Stable METTL16-knockdown HGC-27 (**c**, **e–g**) or AGS (**d**) cells were treated with indicated different concentrations of elesclomol-Cu (ratio=1:1) for 72 h and the growth survival rate was detected by CCK-8 assay. **c** The growth survival rate (left) and cell survival observed by microscope (right) in stable METTL16 knockdown HGC-27 cells under treatment with indicated different concentrations of elesclomol-Cu (ratio = 1:1) for 72 h ($n = 3$). $P = 0.00503$ (Sh1) and 0.00361 (Sh2).

Scale bar, 50 μm. **d** The growth survival rate in stable METTL16 knockdown AGS cells under treatment with indicated different concentrations of elesclomol-Cu (ratio=1:1) for 72 h ($n = 3$). $P = 0.00297$ (Sh1) and 0.00970 (Sh2). **e** Stable METTL16-knockdown HGC-27 cells were treated with indicated different concentrations of disulfiram-Cu (ratio = 1:1) for 72 h ($n = 3$). $P = 0.00990$ (Sh1) and 0.00577 (Sh2). **f** Tetrathiomolybdate (TTM) (1 μM) supplement reversed the recovery of growth inhibition induced by shMETTL16 with different elesclomol-Cu (1:1) concentrations in HGC-27 cells ($n = 3$). $P = 0.00198$ (TTM). **g**, **h** The growth survival rate of verified METTL16-KO clones (KO-1 and KO-4) under treatment with indicated different concentrations of elesclomol-Cu (**g**) and disulfiram-Cu (**h**) for 72 h ($n = 3$). $P = 0.00671$ (**g**) and 0.00189 (**h**). Statistical data presented in this figure show mean values ± SD of three times of independent experiments. Statistical significance was determined by Two-tailed $t$ test, $*P < 0.05$, $**P < 0.01$. Source data are provided as a Source data file.

were mainly associated with cancer pathways (Supplementary Fig. 3c). GO enrichment analysis showed top 15 pathways enriched by METTL16-modified genes, some of which were correlated with mitochondrial pathways (Fig. 3c). Furthermore, we identified 10 typical factors from cuproptosis-related pathway that were related to low m6A methylation levels as well as downregulated mRNA levels in METTL16-Sh group (Fig. 3d and Supplementary Data 1). qPCR analysis showed that the mRNA levels of *FDX1, MTF1, PDHB, ATP7A*, and *DLD* were decreased in METTL16-sh cells. *FDX1* exhibited the most significantly decreased mRNA level in METTL16-sh cells versus the control cells (Fig. 3e and Supplementary Fig. 3d). Furthermore, the significant

decreased protein level of FDX1 was detected in METTL16-Sh or -KO cells (Fig. 3f, g). Considering that FDX1 is the key regulators of copper ionophore–induced cell death, we presumed that METTL16 induced cuproptosis is FDX1-dependent.

Next, we evaluated whether METTL16 regulated cuproptosis by affecting the expression of FDX1. Given that protein lipoylation are the key element of cuproptosis[11], we assessed whether METTL16 knockdown affected protein lipoylation using a lipoic acid-specific antibody as a measure of DLAT lipoylation. The result showed that METTL16 knockdown resulted in reduced DLAT lipoylation, as measured by immunoblotting (Fig. 3h). Moreover, FDX1

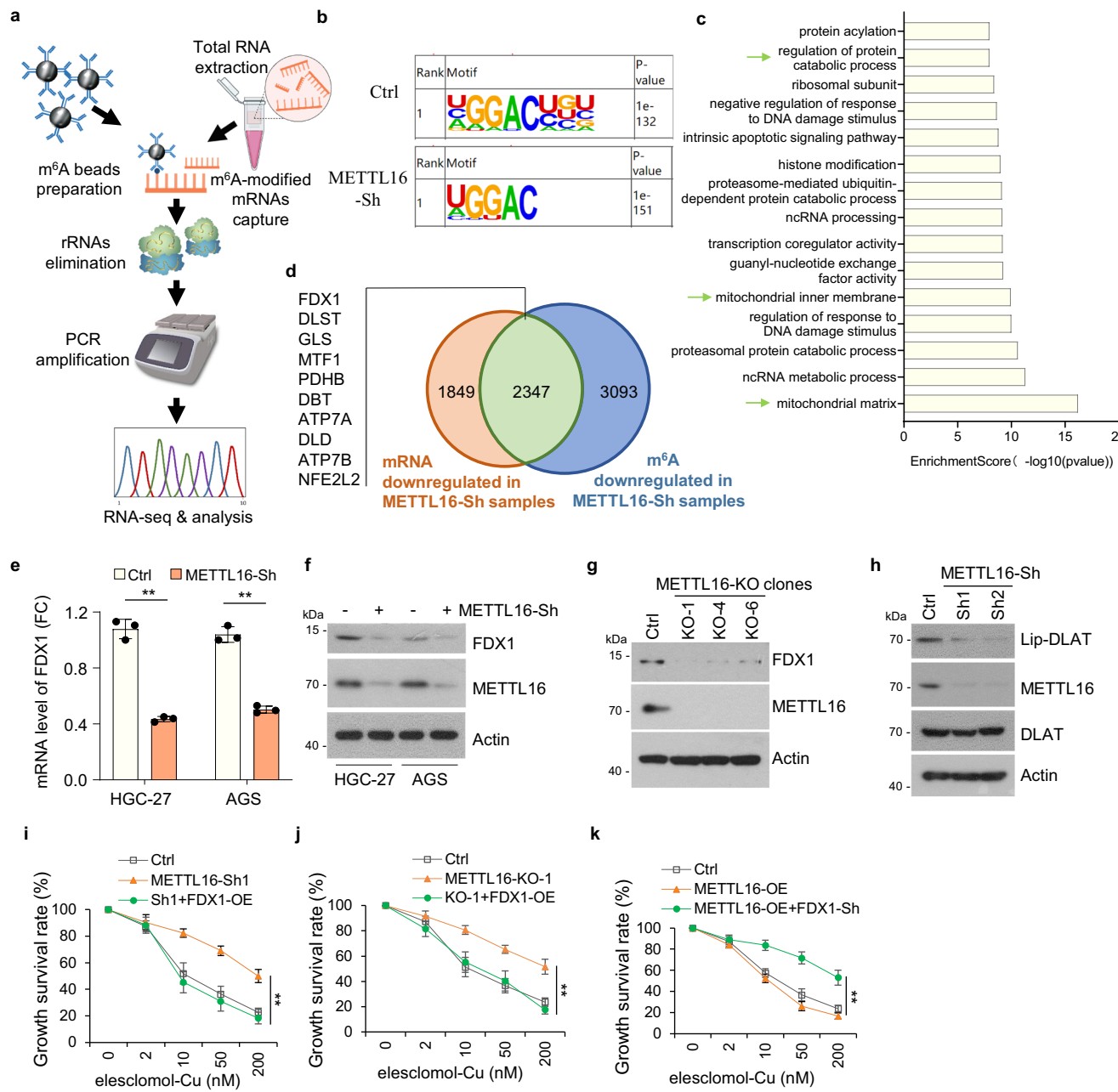

**Fig. 3 | METTL16 promotes cuproptosis by targeting FDX1. a–e** MeRIP-seq analysis of m⁶A regulation between METTL16 and FDX1 in HGC-27 cells. **a** Workflow for methylated RNA immunoprecipitation sequencing (MeRIP-seq). **b** Specific m⁶A motif analysis in stable Ctrl-Sh and METTL16-Sh cell lines. **c** Significant 15 GO terms of METTL16-related genes identified between Ctrl-Sh mRNAs and METTL16-Sh mRNAs containing m⁶A modifications. The green arrow indicated the specific terms that were related to cuproptosis. Enrichment Scores were calculated using chi-square test. **d** Venn plot showing the intersection of downregulated mRNAs and downregulated m⁶A in METTL16-Sh cells. The gene list showed the cuproptosis-related genes in the intersection. **e** qRT-PCR analysis of FDX1 in Ctrl and shMETTL16-Sh AGS and HGC-27 cells ($n = 3$). HGC-27: $P = 0.0023$. AGS: $P = 0.0064$. **f, g** Western blot analysis of METTL16 and FDX1 expressions in Ctrl and

METTL16-Sh cells (**f**) or METTL16-knockout (KO-1, KO-4, KO-6) (**g**) HGC-27 cells. **h** Western blot analysis of METTL16 and lipoylated DLAT expression in METTL16-knockdown and control HGC-27 cells. **i–k** Growth survival was analyzed by CCK8 assay. **i, j** FDX1-overexpression reversed the recovery of growth inhibition induced by METTL16-knockdown (**i**) or knockout (**j**) cell lines ($n = 3$). $P = 0.00668$ (Sh1) and $0.00984$ (KO-1). **k** FDX1-knockdown rescued the growth inhibition induced by METTL16-overexpression cells. Stable control or METTL16-overexpression cell lines transfected with or without FDX1 silencing plasmids were treated with indicated different concentrations of elesclomol-Cu (ratio = 1:1) for 72 h ($n = 3$). $P = 0.00617$. Statistical data presented in this figure show mean values ± SD of three times of independent experiments. Statistical significance was determined by Two-tailed $t$ test, *$P < 0.05$, **$P < 0.01$. Source data are provided as a Source data file.

overexpression reversed the antagonistic effect of cuproptosis in METTL16 knockdown and knockout cell lines after elesclomol/disulfiram-Cu treatment (Fig. 3i, j). Cuproptosis induced by METTL16-overexpression were inhibited by FDX1 knockdown (Fig. 3k). Collectively, these data suggested that METTL16 promotes cuproptosis by targeting FDX1.

## METTL16 promotes FDX1 accumulation via m⁶A modification on *FDX1* mRNA

To further explore how METTL16 regulated m⁶A modification and thus influenced *FDX1* mRNA levels, gene-specific MeRIP-qPCR assays were performed. We found that the m⁶A levels on *FDX1* mRNA is significantly decreased in METTL16 knockdown and knockout cells, and

significantly increased in cells transfected with METTL16-overexpression plasmid (Fig. 4a, b). The decay rate of *FDX1* mRNA was faster in METTL16-knockdown cells than in control cells when transcription was halted with actinomycin D (ActD) (Fig. 4c). It has been reported that METTL16 locates in cytoplasm to promote mRNA translation, and locates in the nucleus to deposit m⁶A into specific RNA targets[29]. Endogenous METTL16 located in the nucleus in GC tumor cells (Supplementary Fig. 3e). These results indicated that METTL16-mediated methylation led to *FDX1* mRNA stability. Integrative genomics viewer (IGV) analysis of enriched m⁶A peaks of *FDX1* showed a decreased m⁶A level in METTL16-sh cells (green peak) compared with control cells (red peak), indicating that METTL16 may promote m⁶A modification of *FDX1* in the CDS or 3′UTR regions (Fig. 4d). Therefore, part of CDS (selected region:110456920-110457047, and 110462353-110462468) as well as 3′UTR regions (selected region: 110462469-110464884) of *FDX1* were fused into the downstream of pGL3 luciferase reporter for constructing FDX1-WT luciferase reporter. With co-transfection of FDX1 luciferase reporter and different doses of METTL16-OE plasmids, we found that METTL16 significantly promoted the FDX1-luciferase activity (Fig. 4e).

To explore it furtherly, we predicted the possible m⁶A modification sites on *FDX1* mRNA using SRAMP, a sequence-based m⁶A modification site predictor. Integrating the prediction results and data in Fig. 4e, we identified two potential m⁶A modification sites with very high confidence: site 602 A in the CDS region and site 820 A in the UTR region of the *FDX1* transcript (Fig. 4f, g). The following MeRIP-qPCR analysis with specific primers proved that the 602 site on *FDX1* transcripts is the direct substrate of METTL16-mediated methylation (Fig. 4h). Additionally, we designed and constructed FDX1-602-Mut and FDX1-820-Mut luciferase reporters by replacing the specific adenosine (A) in m⁶A motif with thymine (T) based on FDX1-WT luciferase reporter (Fig. 4g). The results of the dual-luciferase assay indicated that METTL16 could not promote the luciferase activity of the reporter construct bearing *FDX1*-CDS/UTR with mutations at site 602 (Fig. 4i). Analysis of METTL16 and FDX1 expression in GEPIA database suggested that METTL16 and FDX1 were positively correlated in GC (Fig. 4j). In addition, immunohistochemical staining demonstrated that the expression of METTL16 was positively correlated with that of FDX1 in a tissue microarray containing 54 pairs of GC and adjacent normal tissues (Fig. 4k). Together, these results confirmed that METTL16-mediated m⁶A modification on *FDX1* mRNA is critical for *FDX1* mRNA stability.

## METTL16 is lactylated at K229 under copper stress

Next, we evaluated whether METTL16 responded to copper stress in GC. Upon copper treatment, METTL16 mRNA and protein levels were unchanged but FDX1 protein levels were increased dramatically (Fig. 5a, b). To verify whether upregulation of FDX1 expression was METTL16-dependent under copper stress, we measured FDX1 protein expression in METTL16-knockdown or -knockout cells in the presence or absence of copper. The results showed that METTL16 deficiency abolished the copper-induced upregulation of FDX1 (Fig. 5c, d). Since METTL16 mRNA and protein levels were unchanged under copper stress, we then measured changes in PTM of METTL16 using mass spectrometry. The analysis revealed eleven lactylation and two acetylation sites on METTL16 protein sequence (Fig. 5e). PTM levels of the six lactylation (Red) sites were significantly increased after lactate treatment, implying that these six sites could be regulated in tumor cells. We then confirmed the presence of a lactylation or acetylation response to copper stress. The results showed that copper treatment markedly increased lactylation rather than acetylation level of METTL16 (Fig. 5f). To identify the important lactylation site of METTL16 in copper-related metabolism, lactylation-defective mutants (K to R) of these six lactylation sites and K410 (as a representative site, of which the lactylation was unchanged under lactate stress) were

generated. We found that only the METTL16-K229R mutant showed reduced lactylation under copper stress (Fig. 5g). Subsequently, b-y ion matching diagram of METTL16-K229 and other lactylation sites were showed by LC-MS/MS analysis in Fig. 5e (Fig. 5h, Supplementary Fig. 4). To further confirm METTL16-K229 lactylation under copper stress, a METTL16 lacty-K229 antibody was generated, and its effectiveness and specificity were verified using the lacty-K229 peptide and cells expressing METTL16-WT, -K229R, or -K229E (Fig. 5i, j). Both copper and lactate treatments induced strong METTL16 lactylation at K229 (Fig. 5k, l), indicating that K229 is an essential lactylation site in copper-related metabolism. These results indicated that METTL16 is lactylated at K229 under copper stress.

Then, we analyzed potential impact of K229 lactylation on METTL16. Firstly, METTL16-K229 lactylation didn't affect the nuclear location of METTL16 (Supplementary Fig. 5a). Crystal structure of METTL16 complexed with MAT2A RNA hairpin (PDB 6DU5), showed that K229 formed a salt bridge with E226 and K163 from the auto-inhibition loop (pink) was inserted in the SAM binding site of METTL16 (Supplementary Fig. 5b). Our structure modelling indicated that R230 could stabilize the autoinhibited state by forming hydrogen bonds with K163 or Q162 of the loop. Lactylation of K229 will abolish its salt bridge with E226, which may lead to conformational changes in side-chains of E226 and allow the formation of salt bridge with R230 (Supplementary Fig. 5c). This would weaken the stability of the auto-inhibited state leading to elevated enzymatic activity of METTL16. Then, the downstream targets of METTL16, such as BCAT1, BCAT2, and GPX4[51] were selected to verified the effect of METTL16-K229 lactylation on the its methyltransferase activity. The result showed that METTL16-K229E promoted the methyltransferase activity (Supplementary Fig. 5d). FDX1 protein levels were further ensured in stable METTL16-rescued (METTL16-WT, -K229R, or -K229E) cell lines, in which endogenous METTL16 was replaced by wild-type (WT), lactylation-deficient (K229R), or lactylation-mimetic (K229E). The result showed that FDX1 protein levels were decreased in METTL16-knockdown cells and recovered after METTL16-WT or -K229E rescued cells, but not METTL16-K229R rescued cells (Fig. 5m). Together, METTL16-K229 lactylation is induced by copper stress and promotes the methyltransferase activity of METTL16.

## SIRT2 delactylates METTL16-K229 and inhibits the activity of METTL16

The above data confirmed that copper activated the lactylation of METTL16-K229, but the specific delactylases and lactyltransferases involved in tumor cells remained unclear. Theoretically, the higher the lactate content, the more lactic acid is available, increasing the lactylation of proteins. Thus, we examined the intracellular lactate in copper-treated GC cells using the lactic acid content detection kit and found no significant difference after copper treatment (Supplementary Fig. 6a). Then, we investigated whether copper mediated the regulation of lactyltransferases or delactylases on METTL16. The LC-MS/MS assay to identify the binding proteins of METTL16 was performed (Supplementary Data 2). Identified as a binding protein of METTL16, SIRT2 is a typical deacetylation enzyme[52] classified in the major delactylase and lactyltransferase groups (Fig. 6a). EIF3b was also identified as a binding protein of METTL16, consistently with previous research[29]. To investigate and verify the interaction between METTL16 and SIRT2, METTL16-SIRT2 binding was detected in HGC-27 cells (Fig. 6b). The direct METTL16-SIRT2 association was verified using a His-pull-down assay (Fig. 6c). METTL16-K229 lactylation was markedly inhibited by SIRT2 overexpression (Fig. 6d), and promoted by SIRT2 knockdown or treatment with the SIRT2 inhibitor- AGK2 (Fig. 6e). Moreover, copper-induced lactylation of METTL16-K229 was inhibited by SIRT2 overexpression, whereas increased by SIRT2 knockdown or AGK2 treatment (Fig. 6f, g). Collectively, these results indicated that SIRT2 interacts with and delactylates METTL16 at site K229.

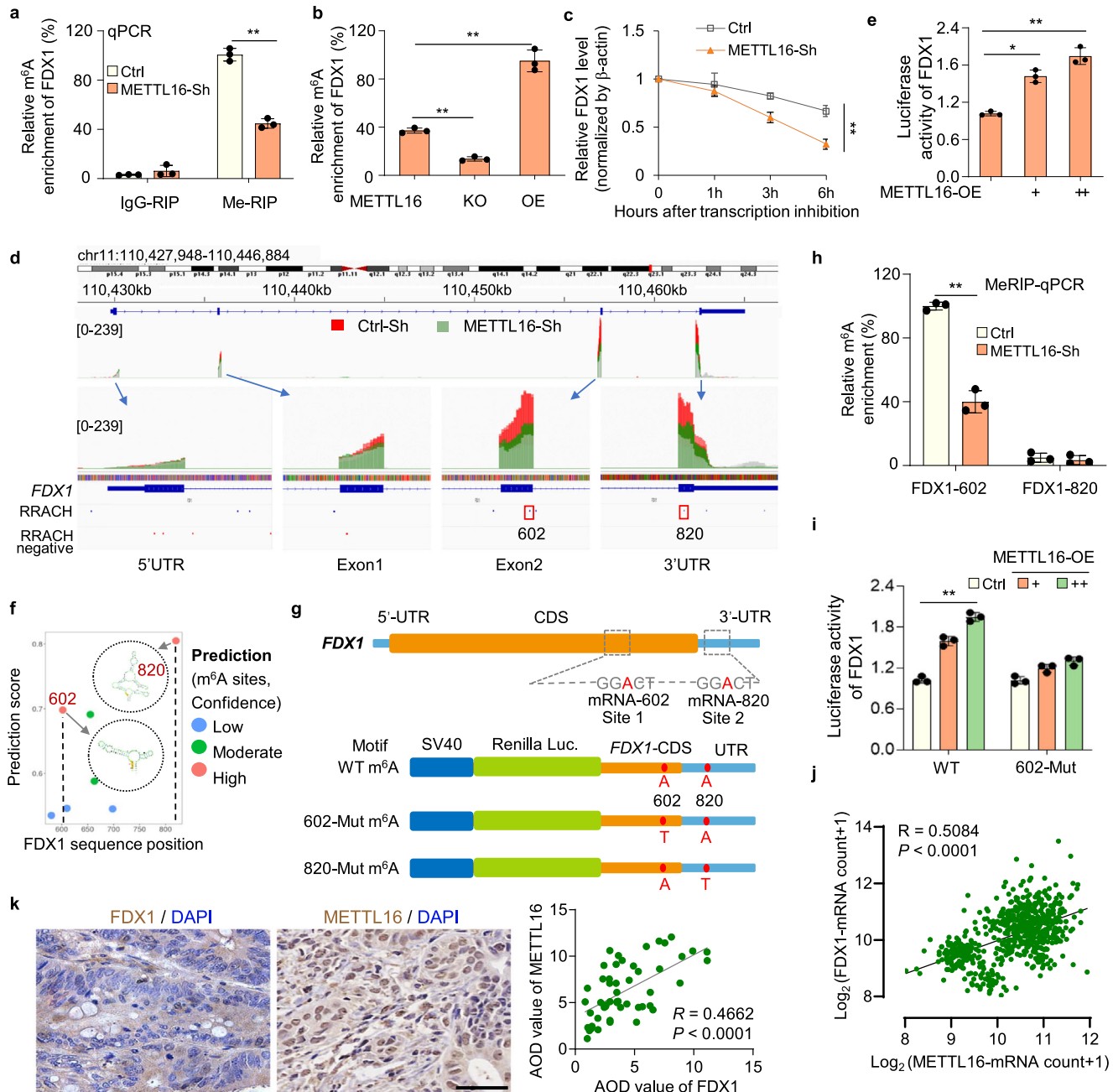

**Fig. 4 | METTL16 promotes FDX1 accumulation via m⁶A modification on *FDX1* mRNA. a** METTL16 knockdown significantly decreased the m⁶A levels on *FDX1* mRNA. Gene-specific Me-RIP or IgG (mock)-RIP qPCR assay analysis was performed in Ctrl and METTL16-Sh cell lines (n = 3). IgG-RIP: $P = 0.3481$. Me-RIP: $P = 0.0073$. **b** Bar plot showing the relative m⁶A enrichment of *FDX1* in METTL16-WT, KO, and OE cells (n = 3). $P = 8.35276E-05$. **c** qRT-PCR analysis of *FDX1* mRNA decay rate at the indicated times after actinomycin D (2 μg/ml) treatment in Ctrl and METTL16-Sh cell lines, with expression normalized to that of β-actin (n = 3). $P = 0.0063$. **d** IGV plots of m⁶A peaks in m⁶A immunoprecipitation sample (relative to input sample) at *FDX1* mRNA in METTL16-Sh cells (green peak) related to the control cells (red peak). The y-axis shows normalized reads coverage. **e** Luciferase assay analysis of the FDX1-luciferase activity after METTL16 overexpression in HGC-27 cells (n = 3). $P = 0.0044$. **f** Prediction of m⁶A modification sites. The secondary structure of high-confidence prediction sites is shown. **g** The structure chart showing the predicted m⁶A motifs in the CDS and 3'UTR of *FDX1* according to the result of MeRIP-seq. The

design and structure of luciferase reports were shown below. **h** Me-RIP-qPCR assay validating that FDX1-602 shows significant methylation activity compared to FDX1-820, using Ctrl and METTL16-Sh cell lines (n = 3). FDX1-602: $P = 0.0046$. **i** Relative luciferase activity of wildtype (WT) or mutation (Mut) (A-to-T mutation) of FDX1 luciferase reporter in Ctrl and METTL16-OE cell lines (n = 3). $P = 0.00051$. **j, k** Statistical analysis of Pearson correlation between METTL16 and FDX1. **j** Correlation between *METTL16* and *FDX1* mRNA expression in datasets from the TCGA and GTEx databases ($R = 0.5084$). $P < 0.0001$. **k** Immunohistochemical staining showing the cellular locations of METTL16 and FDX1 in tissue microassay (T16-425) with 54 pairs of GC tissues. Scale bar, 50 μm. Pearson correlation analysis of the average optical density (AOD) values between METTL16 and FDX1 was shown on the right ($R = 0.4662$). $P < 0.0001$. Statistical data presented in this figure show mean values ± SD of three times of independent experiments. Statistical significance was determined by Two-tailed *t* test, *$P < 0.05$, **$P < 0.01$. Source data are provided as a Source data file.

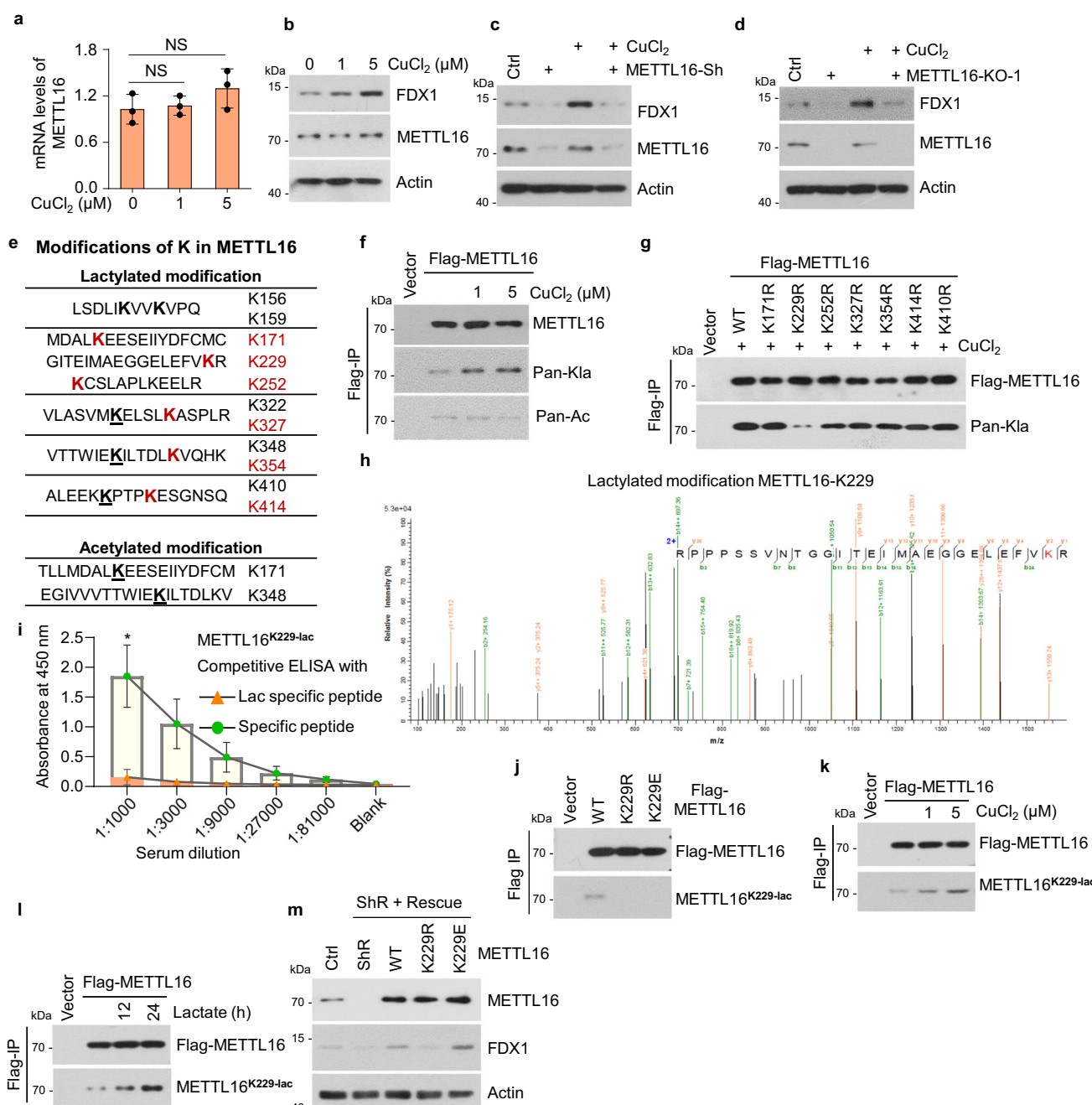

**Fig. 5 | METTL16 is lactylated at K229 under copper stress. a**, **b** The mRNA and protein levels of METTL16 under copper stress in HGC-27 cells. **a** qRT-PCR analysis of METTL16 mRNA levels under different CuCl$_2$ concentrations ($n = 3$). $P = 0.358$ (1 μM) and 0.1891 (5 μM). **b** Western blot analysis of METTL16 and FDX1 protein levels under different CuCl$_2$ concentrations. **c**, **d** Western blot analysis of FDX1 expression in METTL16-Sh (**c**) or METTL16-KO (**d**) HGC-27 cells in the presence or absence of CuCl$_2$ (1 or 5 μM) treatment. **e–h** The discovery and validation of the lactylated site of METTL16 in HGC-27 cells. **e** Mass spectrometry analysis of potential modifications of lysine (K) in METTL16. Potential lactylation modifications are shown above, and acetylation modifications are shown below. **f** HGC-27 cells transfected with Flag-METTL16 were treated with CuCl$_2$ (1 or 5 μM) for 12 h. Cell lysates were immunoprecipitated with Flag-M2 beads followed by western blot for lactylation and acetylation of METTL16 using anti-pan-lactyl and anti-pan-acetyl antibodies. **g** HGC-27 cells transfected with indicated METTL16 site mutations were treated with CuCl$_2$ followed by immunoprecipitation using Flag-M2 beads and western blot for lactylation of METTL16 using anti-pan-lactyl and Flag antibodies. **h** Mass spectra for METTL16 peptides lactylated at K229. **i**, **j** Validation of the

valence of the Lac-METTL16-K229 antibody in HGC-27 cells. (**i**) Competitive ELISA analysis of lactylation- or peptide-specific supplements in different serum dilutions ($n = 3$). $P = 0.02358$ (1:1000). **j** Characterization of METTL16-K229 lactylation antibody (METTL16$^{K229\text{-lac}}$). Cell lysates from 293T cells transfected with wildtype or mutant forms of Flag-METTL16 were immunoprecipitated with Flag-M2 beads followed by western blot using the antibody against METTL16$^{K229\text{-lac}}$ or Flag. **k** HGC-27 cells transfected with Flag-METTL16 were treated with CuCl$_2$ (1/ 5 μM) followed by immunoprecipitation using Flag-M2 beads and western blot for lactylation of METTL16 using anti-METTL16$^{K229\text{-lac}}$ antibody. **l** HGC-27 cells transfected with Flag-METTL16 were treated with Lactate (30 mM) for 12 h or 24 h. Cell lysates were immunoprecipitated using Flag-M2 beads and western blot for lactylation of METTL16 using the anti-METTL16$^{K229\text{-lac}}$ antibody (**m**) METTL16-WT/ -K229R/ -K229E were rescued in stable METTL16 knockdown cells and FDX1 protein level was detected using western blot. Statistical data presented in this figure show mean values ± SD of three times of independent experiments. Statistical significance was determined by Two-tailed $t$ test, *$P < 0.05$, **$P < 0.01$, NS not significant. Source data are provided as a Source data file.

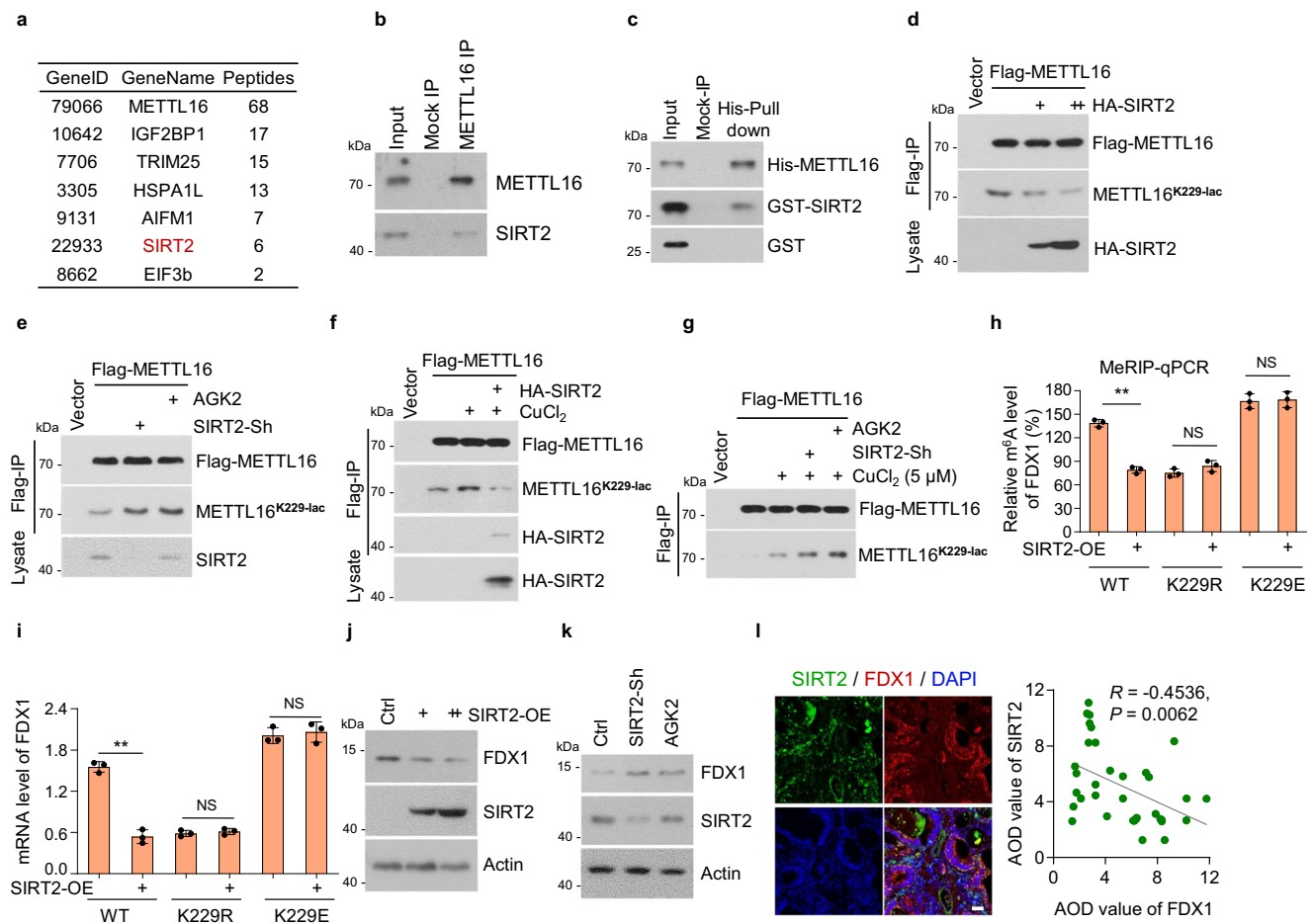

**Fig. 6 | SIRT2 delactylates METTL16-K229 and inhibits the activity of METTL16.**
**a** Identification of METTL16-associated proteins using liquid chromatography-tandem mass spectrometry (LC-MS/MS). **b, c** Western blot verification of the interaction between METTL16 and SIRT2 in HGC-27 cells. **b** Endogenous METTL16 in HGC-27 cells was precipitated using the anti-METTL16 antibody or IgG (mock IP). Co-precipitated SIRT2 was detected by western blot. **c** Recombinant His-tagged METTL16 was incubated with GST-SIRT2 or GST proteins at 4 °C for 4 h followed by His pull-down and western blot. **d, e** SIRT2 inhibited copper-induced METTL16-K229 lactylation. Cell lysates were immunoprecipitated using Flag-M2 beads and western blot for total or lactylation of METTL16 using anti-Flag or METTL16$^{K229-lac}$ antibody. **d** Flag-METTL16 and HA-SIRT2 plasmids were co-transfected into HGC27 cells followed by immunoprecipitation. **e** The Flag-METTL16 plasmid was transfected in control or SIRT2-knockdown (SIRT2-Sh) HGC-27 cells with or without AGK2 (10 μM) treatment for 24 h. **f** HGC27 cells co-transfected with Flag-METTL16 and HA-SIRT2 were treated with CuCl$_2$ (5 μM) for 12 h and followed by immunoprecipitation. **g** The flag-METTL16 plasmid was transfected in control or SIRT2-knockdown (SIRT2-Sh) HGC-27 cells with or without the treatment of AGK2 (10 μM,

24 h) or CuCl$_2$ (5 μM, 12 h). **h** SIRT2 was transfected in METTL16-WT/K229R/K229E rescued cells and the relative m6A level of *FDX1* was detected by MeRIP-qPCR assay (n = 3). P = 0.00371 (WT), 0.3224 (K229R), and 0.8714 (K229E). **i** SIRT2 was transfected in METTL16-WT/K229R/K229E rescued cells and *FDX1* mRNA level was detected by the qRT-PCR assay (n = 3). P = 0.00926 (WT), 0.4222 (K229R), and 0.5107 (K229E). **j** Different doses of SIRT2 overexpression plasmid were transfected in HGC-27 cells, and the protein level of FDX1 was analyzed by western blot using FDX1 antibody. **k** Western blot of FDX1 protein level in Ctrl or SIRT2-Sh HGC-27 cells with or without AGK2 treatment. **l** Immunofluorescent staining of tissue microarray (T16-425) containing 54 pairs of GC and adjacent normal tissues. Representative images showing the co-expression of SIRT2 and FDX1. Pearson correlation analysis of the average optical density (AOD) value between SIRT2 and FDX1 was measured by Image J (R = −0.4536). P = 0.0062. Scale bar, 50 μm. Statistical data presented in this figure show mean values ± SD of three times of independent experiments. Statistical significance was determined by Two-tailed t test, *P < 0.05, **P < 0.01, NS not significant. Source data are provided as a Source data file.

We further explored the regulation of METTL16 by SIRT2 in stable METTL16-rescued (METTL16-WT, -K229R, or -K229E) cell lines. SIRT2 overexpression inhibited *FDX1* m6A levels in METTL16-WT cells but not in METTL16-K229R or -K229E cells, indicating that SIRT2 inhibited METTL16 activity through METTL16-K229 lactylation (Fig. 6h, i). Furthermore, SIRT2 overexpression suppressed FDX1 protein levels (Fig. 6j), whereas SIRT2 knockdown and AGK2 treatment promoted FDX1 protein levels (Fig. 6k). The results of immunofluorescence staining (IF) demonstrated that the SIRT2 protein levels were negatively correlated with FDX1 protein levels in the GC tissue microarray (Fig. 6l). In addition, SIRT2, as a deacetylation enzyme, may also regulated METTL16 deacetylation. We found that total METTL16 acetylation had a partial reduction after SIRT2 over-expression (Supplementary Fig. 6b). To examine whether SIRT2-

mediated METTL16 deacetylation affects the methyltransferase activity of METTL16, cells with METTL16-Sh rescue of METTL16-WT, -K171R, -K171Q, -K348R, and -K348Q were generated and performed to detect m6A level. The results showed that acetylation sites had no effects on methyltransferase activity of METTL16 (Supplementary Fig. 6c and 6d). Moreover, there is no significant difference on increased mRNA level of *FDX1* after METTL16-WT, -K171R, -K171Q, -K348R, and -K348Q overexpression (Supplementary Fig. 6e, 6f). These data indicated that SIRT2-mediated METTL16 deacetylation did not affect the *FDX1* mRNA levels. There was no difference in METTL16 expression with or without SIRT2-knockdown or AGK2-treatment (Supplementary Fig. 6g and 6h). Taken together, these results indicated that SIRT2 negatively regulates METTL16 activity through delactylation at K229.

To further explore the regulation of SIRT2 on METTL16 under copper, we detected whether copper influences the interaction of SIRT2 and METTL16. Unexpectedly, the results showed that copper treatment slightly reduces the binding of SIRT2 and METTL16 (Supplementary Fig. 6i). Given that the lactylation of METTL16 increased under copper treatment, we have reasons to speculate that copper treatment is likely to enhance the binding of METTL16 and lactyltransferases. Therefore, ten potential lactyltransferases were selected, which has been reported or predicted, and were knockdown with specific siRNAs. These are as followed: KAT3B (EP300), CREBBP (CBP), ACSS2, AARS1, AARS2, KAT6A (MOZ), KAT8 (MOF), KAT5, NMT1, and NMT2. The results showed that just AARS1-Si and AARS2-Si decreased METTL16-K229 lactylation under copper treatment (Supplementary Fig. 7a). Furthermore, copper promotes the interaction of AARS1 or AARS2 with METTL16 significantly (Supplementary Fig. 7b). Overexpression of AARS1 or AARS2 promotes the lactylation of METTL16-K229 indeed after copper treatment (Supplementary Fig. 7c). Moreover, in vitro lactylation assay showed that AARS1/2 upregulates lactylation of METTL16-WT other than METTL16-K229R dramatically (Supplementary Fig. 7d and 7e). Moreover, synthesized unmodified peptide around K229 of METTL16 was lactylated at K229 by AARS1 or AARS2 (Supplementary Fig. 7f). These results indicated that AARS1 and AARS2 maybe lactyltransferases of METTL16. The promotion of METTL16 lactylation by copper could be due to the increased interaction between AARS1 or AARS2 with METTL16.

### SIRT2–METTL16–FDX1–cuproptosis axis support a promising therapeutic strategy

FDX1 is an essential mediator of cuproptosis, which is a potential treatment for cancer[53]. We showed that FDX1 protein level was increased after SIRT2-inhibition or copper-induced METTL16-K229 lactylation. Next, we examined the effect of METTL16-K229 lactylation on cuproptosis. The results showed that the decreased DLAT lipoylation mediated by METTL16 knockdown under copper stress was rescued in METTL16-WT/-K229E cells, but not in METTL16-K229R cells (Fig. 7a). Additionally, elesclomol treatment in the presence of copper induced cuproptosis in METTL16-WT or -K229E cells, but not in METTL16-K229R cells (Fig. 7b). Moreover, SIRT2-overexpression reversed the increase of DLAT lipoylation induced by METTL16-WT under copper stress, but not that induced by METT16-K229E, indicating that SIRT2 inhibited cuproptosis through delactylating METTL16-K229 (Fig. 7c). Consistently, AGK2 treatment promoted DLAT lipoylation in METTL16-WT cells, but not in METTL16-K229R cells under copper stress (Fig. 7d). In addition, AGK2 treatment promoted cuproptosis in the presence of elesclomol and copper (Fig. 7e).

We further evaluated whether METTL16 lactylation can enhance the therapeutic response of xenograft tumors in mice receiving elesclomol treatment. METTL16-WT, -K229R, -K229E cells were subcutaneously injected into nude mice above their left and right hind legs. Tumor growth was markedly inhibited in mice receiving elesclomol treatment in the METTL16-WT and -K229E groups but not in the METTL16-K229R group, as observed by a reduction in tumor volume and weight (Fig. 7f–h).

The FDX1 staining in tumor sections showed higher in METTL16-WT and -K229E groups than METTL16-K229R group in tumor xenografts (Fig. 7i). These results demonstrated that the lactylation status of METTL16 at K229 played a crucial role in determining cuproptosis. Additionally, combined treatment of elesclomol and AGK2 were used in tumor xenografts. Analysis of tumor sections showed that AGK2 sensitized elesclomol treatment in vivo, as observed by a reduction of tumor volume and weight (Fig. 7j–l). These results demonstrated that the combination of elesclomol and AGK2 is promising therapy in gastric cancer, especially in malignant tumors—mucinous adenocarcinomas (higher copper content).

## Discussion

Altered copper homeostasis can promote tumor growth, invasiveness, or even confer chemotherapy resistance. However, excessively intracellular copper accumulation leads to a unique type of cell death termed cuproptosis[11]. In this study, we found that copper content is significantly higher in gastric cancer than in normal gastric tissues. Moreover, high copper content is positively related with the TNM staging and worse OS and DFS of GC patients, especially in mucinous adenocarcinomas that are difficult to treated with a more advanced stage and worse OS and DFS than non-mucinous adenocarcinoma[39]. Hence, how to utilize high content of copper in malignant tumors, such as mucinous adenocarcinomas to trigger cuproptosis will be developed as a promising strategy for cancer therapy, and which is worth further investigation. Furthermore, we demonstrated that the combination of elesclomol and SIRT2 inhibitor AGK2 is a promising therapeutic strategy, and tackles the research bottleneck of cuproptosis on tumor therapy. Given that there is no typical cell line for mucinous adenocarcinoma to validate the effectiveness of cuproptosis on cancer therapy, patient-derived xenograft (PDX) models of mucinous adenocarcinomas are constructing and will be used in our future work.

Cuproptosis induced by copper overload is a specific cell death pathway that differs from apoptosis, pyroptosis, and necroptosis[11]. How the essential proteins of cuproptosis were activated, modified, and operated in the disruption of copper homeostasis remains unclear[54]. In this study, we found that FDX1, responsible for reducing $Cu^{2+}$ to a more toxic form $Cu^{1+}$, is dramatically increased under high level copper circumstances. Further research revealed that high copper content promotes METTL16-K229 lactylation, and then lactylated METTL16 upregulates *FDX1* mRNA and protein levels via $m^6A$-modification on *FDX1* mRNA, which ultimately induces DLAT lipolylation and cuproptosis in GC. Consequently, clinical data from GC tissue microarray proved the correlation between METTL16 and FDX1. The copper–lactylated METTL16–FDX1–cuproptosis axis is a key regulatory process in copper-related metabolism, and fills the gap in cuproptosis manipulation.

Abnormal glycolipid metabolism is an typical characteristics of tumors, and tumor cells produce large amounts of lactate through glycolysis even under aerobic conditions[55]. Lactate is taken up by tumor cells and transported to mitochondria for oxidation to provide energy[31], while acting as a form of PTM to regulate histone lactylated modifications[30]. Histone lactylated modification is involved in multiple pathological processes[30,32,33]. Whether there are numerous lactylated modifications on non-histone proteins in tumor progress remain unclear[38]. In this study, we found that non-histone protein METTL16 was lactylated at site K229 under copper stress screening from six lactylation sites that were identified by LC-MS/MS analysis. METTL16-K229 lactylation accelerated its role in DLAT lipoylation and cuproptosis via $m^6A$-modification on *FDX1* mRNA. Moreover, delactylase SIRT2, identified as a binding protein of METTL16 by LC-MS/MS, could reduce METTL16-K229 lactylation, and inhibited the function of METTL16 on cuproptosis. These results revealed a positive feedback loop in which high lactate levels produced by glycolysis promote METTL16 lactylation, which in turn regulates the lipoylation of TCA-related proteins, consequently inducing cuproptosis in the presence of high concentrations of copper in tumor cells. The lactylation of non-histone protein METTL16 is significant to understand the regulatory mechanism between glucose- and copper-related metabolisms in tumors.

Lactate functions as interlinkage between glycolysis and oxidative phosphorylation[56]. Whatever in aerobic or anaerobic conditions, high level of glycolysis is present in almost all tumor types comparing to normal tissues[57]. As a result, lactate was produced, thus providing sufficient substrate for the lactylation of proteins. In addition, lactate can be used to support oxidative phosphorylation[56]. Interestingly,

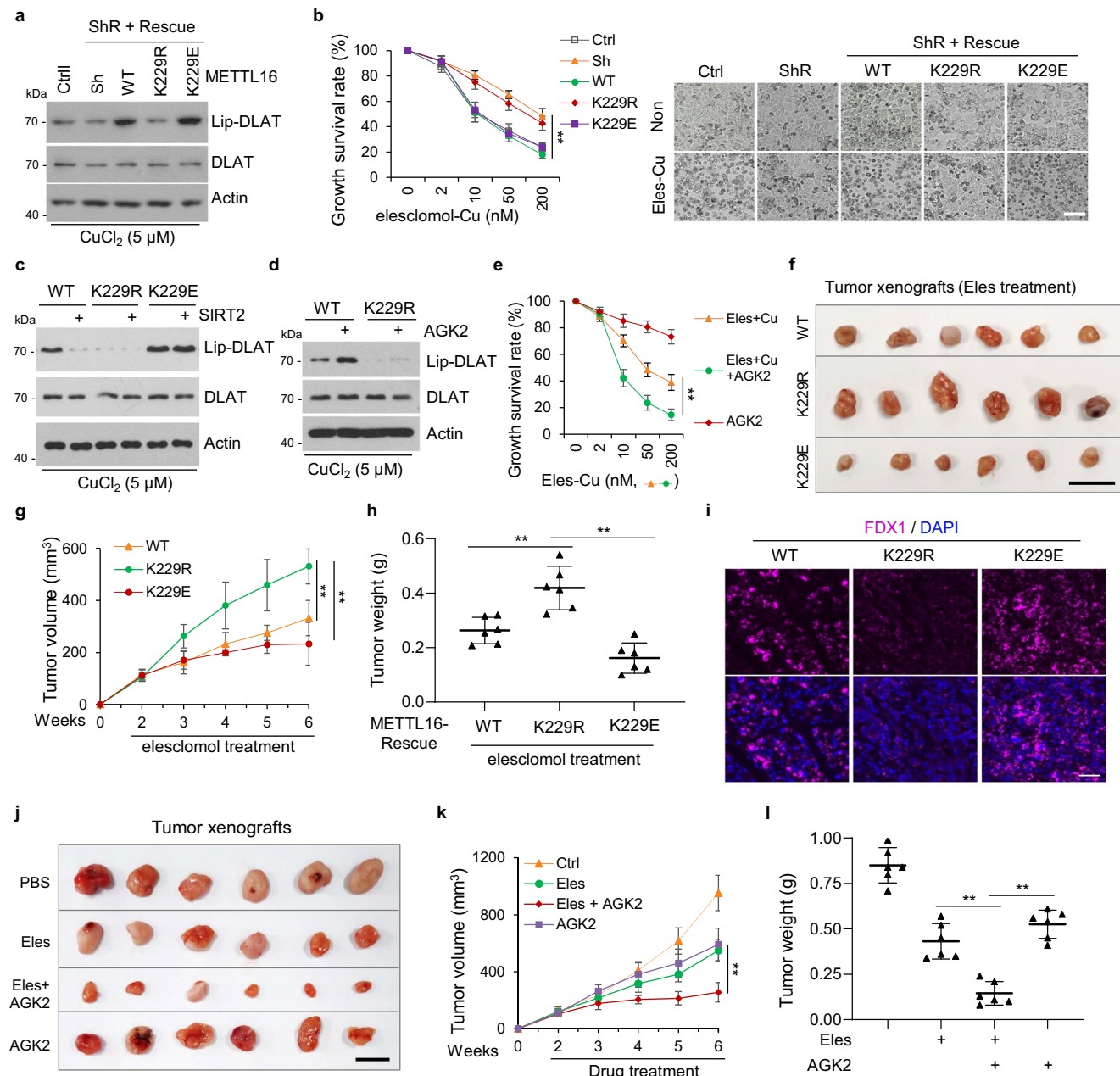

**Fig. 7 | SIRT2–METTL16–FDX1–cuproptosis axis support a promising therapeutic strategy. a** Western blot analysis of lipoylated DLAT (lip-DLAT) expression in METTL16-knockdown and -rescue (WT, K229R, K229E) HGC-27 cell lines. **b** METTL16-knockdown and WT/K229R/K229E rescued HGC-27 cell lines were treated with different concentrations of elesclomol-Cu (1:1 ratio). The growth survival in these cells was detected by CCK8 assay (left) and was observed by microscope (right) (*n* = 3). *P* = 0.00987 (WT-Sh), 0.00176 (K229R-Sh), and 0.00333 (K229E-Sh). Scale bar, 50 µm. **c** SIRT2 was transfected in METTL16-WT, -K229R, or -K229E rescued HGC-27 cells, and the lip-DLAT was detected by western blot. **d** METTL16-WT and -K229R rescued HGC-27 cells were treated with different concentrations of elesclomol-Cu (1:1 ratio) in the presence or absence of AGK2 (10 µM) and the lip-DLAT, DLAT, and Actin was detected by western blot. **e** HGC-27 cells were treated with different concentrations of elesclomol-Cu (1:1 ratio) in the presence or absence of AGK2 and the growth survival rate was detected by CCK8 assay

(*n* = 3). *P* = 0.00244. **f**–**i** Xenograft experiment with METTL16-WT, -K229R, or -K229E rescued HGC-27 cells (*n* = 6 in each group). Tumors were collected and photographed (**f**) (scale bar, 1 cm). The growth curve (**g**) and tumor weights (**h**) were measured with indicated treatment. *P* values: *P* = 0.00579 and 7.84236E−05 (**g**). *P* values: *P* = 0.0052 and 0.00094 (**h**). **i** Representative immunohistochemical images showing FDX1 staining in METTL16-WT, -K229R, or -K229E rescued xenograft tumors (scale bar, 50 µm). **j**–**l** Xenograft experiment with PBS-, elesclomol-, elesclomol +AGK2- and AGK2-treated HGC-27 cells. Tumors were collected and photographed (**j**) (*n* = 6 in each group). The growth curve (**k**) and tumor weights (**l**) were measured with or without elesclomol and AGK2 treatment. *P* values: *P* = 0.00130 (**g**), *P* values: *P* = 0.00153 and 0.00027 (**l**). Statistical data presented in this figure show mean values ± SD of three times of independent experiments. Statistical significance was determined by Two-tailed *t* test, **P* < 0.05, ***P* < 0.01. Source data are provided as a Source data file.

---

oxidative phosphorylation is upregulated in certain cancers, including leukemias, lymphomas, pancreatic ductal adenocarcinoma, and endometrial carcinoma, and can occur even in the face of active glycolysis[58]. The decreases in mtDNA content have been observed in gastric cancers, showing reduced oxidative capacity[59]. MSI/MSS sig-

high gastric cancer is characterized by oxidative phosphorylation[60]. This suggests that some gastric cancers with high level of oxidative phosphorylation may be suitable to be treated through cuproptosis.

There are mainly two sets of m6A RNA methyltransferases with verified mRNA targets in humans: METTL3/14/WTAP complex and

METTL16. Although identified targets of METTL16 are limited until now, the roles of METTL16 as a potential target in many pathologic processes are emerging. There is a large similarity in protein structures between these two enzymes[61], but some differences still exist, suggesting that they have unique responsibilities in cells. In our study, we found that METTL16 was significantly elevated in gastric cancer tissues, while METTL3 and METTL14 had no significant difference (Supplementary Fig. 2a). Further study showed that METTL16 is the methyltransferase of FDX1 by regulating the stability of *FDX1* mRNA. In addition, we also analyzed the mRNA level of *FDX1* is unchanged in METTL3 or METTL14 knockdown cells (Supplementary Fig. 2b). The above results prove that METTL16 mediated m$^6$A modification is the key m$^6$A modification for FDX1 stability.

Given that there is no direct evidence indicating that AARS1/2 is a lactyltransferase, we compared the lactylation modification group with other known aminoacylation modification groups and found that the lactylation modification group has a similar modification structure to the modification structure of the alanine modification group. Therefore, we speculate that AARS1 and AARS2, which have been shown to promote alanine modification, may also have a role in promoting lactylation modification[62–64]. Our study showed that AARS1/2 directly regulate the lactylation modification of METTL16, suggesting that AARS1/2 maybe lactyltransferase for METTL16. In addition, we analyzed the protein structure of AARS1/2 and found that both have metal sulfur cluster binding sites in their amino acid sequences. Therefore, we showed that when cells are exposed to copper ions, copper ions may bind to the metal sulfur cluster binding region of AARS1/2 protein, causing a conformational change in AARS1/2 protein, enhancing its binding to METTL16, and thereby leading to the enhanced lactylation of METTL16. However, how AARS1/2 function as lactyltransferases and whether there are other factors involved in supporting or regulating enzymatic activities of AARS1/2 remain to be further explored.

Collectively, these results indicate that METTL16 acts as a driving force for cuproptosis. The lactylation of METTL16 at K229 upregulates FDX1 protein expression via m$^6$A modification on *FDX1* mRNA and serves as a pathway for triggering cuproptosis under copper stress. Additionally, the study findings demonstrate that METTL16 is a central hub in coordinating glucose and copper metabolism in GC. Moreover, the results indicate that METTL16 lactylation provide a potential target for evaluating the effectiveness of copper ionophore drugs. Given that gastric tumors (especially malignant tumor types) have higher copper and lactate concentrations than normal tissues, treatment with copper ionophores to bind copper ions and SIRT2-specific inhibitor may serve as a feasible and available strategy for gastric cancer therapy.

## Methods

### Cells and culturing conditions
AGS, HGC-27, or 293T cells (Culture Collection of the Chinese Academy of Sciences, Shanghai, China) were cultured in Roswell Park Memorial Institute (RPMI) 1640 medium (Bioagrio) or Dulbecco's Modified Eagle Medium (DMEM) medium supplemented with 10% fetal bovine serum (FBS, Bioagrio) and 1% penicillin and streptomycin (NCM Biotech). AGS and HGC-27 cells were subjected to cell-line authentication. All cell lines were cultured in a humidified incubator at 37 °C under at an atmosphere containing 5% CO$_2$. Cells were treated with different concentrations of elesclomol, disulfiram, and TTM (MCE, USA) for in vitro researches.

### Tissue specimens and patient information
Three paired groups of GC and adjacent normal tissues were used in the present study. The first group included 48 pairs of fresh-frozen GC and adjacent normal tissue samples obtained from patients who underwent gastrectomy at Shanghai General Hospital (Shanghai, China) between October 2016 and May 2021. All samples were transferred to liquid nitrogen and stored at −80 °C until protein extraction.

The second group included 57 pairs of GC and adjacent normal tissues obtained between 2013 and 2014 from Shanghai General Hospital[62]. The samples were fixed with formalin soon after surgery and embedded in paraffin to construct a tissue microarray (TMA). Three samples were lost due to producing an error, and thus we subsequently only had 54 pairs of samples. The third group included 44 plasma samples obtained from patients who underwent gastrectomy at Shanghai General Hospital (Shanghai, China) between December 2015 and October 2022. Plasma was isolated from the blood cellular component by primary centrifugation and stored at −80 °C till usage. All samples were collected with approval from the Ethics Committee of the Shanghai General Hospital (Research Ethics Approval Code: 2022SQ123), and informed consent was obtained from all patients.

### Mouse xenograft tumor experiment
Animal experiments were performed according to the guidelines of the National Institutes of Health and approved by the Animal Care Committee of Shanghai General Hospital. Briefly, we performed two groups of xenograft tumor experiments. The first group includes 18 BALB/C nude mice aged among 4–6 weeks. Stably transfected METTL16-WT/K229R/K229E AGS cells were injected into the upper back of the nude mice (6 mice per group, $1 \times 10^7$ cells, 100 μL per mouse). The second group includes 24 BALB/C with the same conditions presented above. AGK2 (MCE, USA) was dissolved in 100% DMSO at 10 mM stock concentration, while elesclomol (MCE, USA) was diluted into 800 μl of DMSO to reach a final concentration of 156.05 mM. Both of compounds were diluted into working solution at 10% concentration for better dissolution (the ingredients of the cosolvent contained 40% PEG300, 5% Tween-80 and 45% saline). When tumors reached a volume of 100 mm$^3$, mice were intraperitoneally injected with the dissolved compounds (100 μl per time), and the dose frequency was 3 times per week for 3 weeks for AGK2, while 4 times for 3 weeks for elesclomol. The weights and volumes of tumors were measured and calculated after collection.

### m$^6$A dot blot assay
Total RNA was isolated from GC cells that were treated with different concentrations of Cu$^{2+}$ (0, 1, and 5 μM Cu$^{2+}$), diluted to 50 and 10 ng/μL and denatured at 95 °C for 3 min. The samples were deposited on nitrocellulose filter membranes (Millipore, Billerica, MA, USA) and cross-linked by UV light twice, each for 5 min, followed by incubation with anti-m$^6$A antibody (2 μg/ml; Abclonal, A19841, Wuhan, China) overnight at 4 °C. The spots were visualized using an imaging system after incubation with the secondary antibodies.

### Antibody production and purification
The site-specific antibody of lactylated METTL16-K229 was purchased from the Shanghai HuiOu Biotechnology Co., Ltd. The process of production and purification was operated by the company. In brief, the antibody was raised by immunizing rabbits with the synthetic peptide C-EGGELEFVK(lactyl)RIIHDS coupled with keyhole limpet hemocyanin (KLH). Anti-serum was collected after four immunization doses. The valence of the antibody was detected using competitive ELISA[63] with the synthetic peptide (C-EGGELEFVK(lactyl)RIIHDS) and an unmodified peptide (C-EGGELEFVKRIIHDS).

### RNA extraction and quantitative real-time polymerase reaction (qRT-PCR)
The total RNA was extracted from GC cells and tissues using EasyZol Reagent (NovaBio, Shanghai, China). The protein concentration was measured using a Nanodrop 2000 spectrophotometer (Thermo Fisher Scientific, Waltham, MA, USA). Total RNA was reverse transcribed using HiScript III All-in-one RT SuperMix Perfect for qPCR (Vazyme, Shanghai, China) and diluted 5-fold. cDNA amplification was performed using Taq Pro Universal SYBR qPCR Master Mix (Vazyme,

Shanghai, China) with the QuantStudio™ 6 Flex Real-Time PCR System (Thermo Fisher Scientific, USA) using the following cycling conditions: 1 cycle of 95 °C for 30 s; 40 cycles of 95 °C for 10 s, 60 °C for 30 s; 1 cycle of 95 °C for 15 s, 60 °C for 60 s and following 95 °C for 15 s. β-actin was used as the internal control, and each sample was analyzed in triplicated. The results were analyzed using the $2^{-\Delta\Delta CT}$ method and gene expression was reported relative to that of the internal control[64].

## Luciferase reporter assay

Part of CDS (selected region:110456920-110457047, and 110462353-110462468) as well as 3'UTR regions (selected region: 110462469-110464884) of *FDX1* were cloned into pGL3-control vectors (Genewiz, USA) which was comprised of firefly luciferase (F-luc). For mutant 602 reporter plasmid, the predicted adenosine (A) in m6A motif were replaced by thr (T), respectively. Pre-treated AGS cells were seeded into 24-well plate followed by co-transfection of 0.5 μg of wild-type or mutated FDX1 reporter plasmids and 25 ng pRL-TK plasmids (renilla luciferase reporter vector) using jetPRIME Polyplus kit. After 24–36 h, cells were harvested to access the luciferase activity using Dual-Glo Luciferase system (Promega, USA) with the normalization to pRL-TK. Each group was conducted in triplicate.

## Western Blot

GC cells or tissues lysates were extracted using RIPA lysis buffer (Biosharp, Hefei, China) containing 1% PMSF solution (Beyotime Biotech, Shanghai, China) and 1% protease inhibitor cocktail (MedChemexpress, Monmouth Junction, NJ, USA) at 4 °C for 20 min. The protein concentration was quantified using the bicinchoninic acid (BCA) assay kit (Beyotime Biotech, China) and samples were diluted to the same protein concentration. Equal amounts of protein were separated by 10% SDS-PAGE electrophoresis (Epizyme Biotech, Shanghai, China) and transferred to an Immobilon-P PVDF membrane (Millipore, USA). After blocking with NcmBlot blocking buffer (NCM Biotech, Suzhou, China), the membrane was incubated overnight with primary antibodies, including anti-β-actin, (Servicebio, Wuhan, China, 1:2000), anti-Klac (PTM Bio, Hangzhou, China, 1:1000), anti-METTL16 (ABclonal Technology, Wuhan, China, 1:2000), anti-FDX1 (Proteintech, USA, 1:1000), anti-Lipoic Acid (Abcam, UK, 1:1000), anti-DLAT (Cell Signaling Technology, USA, 1:1000), anti-Ac (Cell Signaling Technology, USA, 1:500), anti-Flag (Sigma-Aldrich, USA, 1:3000), anti-GST (Cell Signaling Technology, USA, 1:1000), anti-SIRT2 (Cell Signaling Technology, USA) overnight. Subsequently, the membranes were incubated with secondary antibodies for 1 h at room temperature and visualized using NcmECL Ultra (NCM Biotech, Suzhou, China) with the Tanon 4600 Automatic Chemiluminescence/Fluorescence Image Analysis System (Tanon, Shanghai, China).

## His-pull-down assay

Purified recombinant His-METTL16 or GST-SIRT2 proteins were expressed in *E.coli* and purified with Ni-NTA beads (Qiagen, Dusseldorf, Germany) or GE28401748 GSTrap™ 4B (GE, USA), and then washed with wash buffer (tris-buffered saline (TBS) containing 40 mM imidazole). GST or GST-SIRT2 protein was incubated with Ni-NTA beads with or without His-METTL16, following incubation at 4 °C overnight. The beads were pelleted, washed five times, and resuspended in 150 μL SDS loading buffer, boiled and analyzed by Western blotting.

## Point mutation construction

A total of 26 pairs of PCR primers were designed and synthesized by Sangon Biotech (Shanghai, China) according to the prediction of the HPLC/MS/MS analysis. Point mutation plasmids were synthesized after general PCR using the METTL16 plasmid as a template with 2 × Phanta Max Master Mix (Vazyme, Nanjing, China). The products were treated with the restriction enzyme DpnI (New England Biolabs, Beijing, China)

to eliminate the remaining plasmids. After undergoing transformation, selection, and transfection in HGC-27 cells, the successfully constructed point mutation plasmids were used in further experiments.

## shRNA/siRNA design and construction

Specific shRNA primers and PLKO.1-puro lentiviral vectors were designed and purchased from Sangon Biotech. The synthesized oligonucleotides were annealed and ligated using BstYI/EcoRI (New England Biolabs) to the sites of PLKO.1-puro to produce PLKO.1-GFP-shMETTL16. Agarose gel electrophoresis was used to separate the specific proteins using a DNA purification kit (BioTeke, Beijing, China). Subsequently, the enzyme-digested fragments were ligated using the DNA Ligation kit (Takara Bio, Dalian, China). Fast-T1 competent cells (Vazyme) were used to transform the ligation products. The amplified bacterial colonies were cultured and subjected to Sanger sequencing. Lactyltransferase siRNAs were designed and produced (AZENTA, Suzhou, China) for verifying the relative lactyltransferase of METTL16. SiRNAs were transfected into GC cells using Lipofectamine 2000 (Invitrogen, CA, USA) according to the manufacturer's instruction. The sequences of siRNAs were listed below: EP300, 5'-CAATTCCGAGACATCTTGAGAdTdT; CBP, CCCGATAACTTTGTGATGT-TTdTdT; ACSS2, 5'-GCTTCTGTTCTGGGTCTGAATdTdT; AARS1, CGATGTCCAGA-AACGAGTGTTdTdT; AARS2, 5'-CCATCATACCTTCTTTGAAATdTdT; KAT6A, 5'-CCGCTGTCACAGTGTAGTATGdTdT; KAT8, 5'-CGAAATTGATGCCTGGTATTTdTdT; KAT5, 5'-TCGAATTGTTTGGGCACTGATdTdT; NMT1, 5'-CGGAAATTGGTTG-GGTTCATTdTdT; NMT2, 5'-CGAAGT-GCTCAAGGAGTTATAdTdT. METTL16-Sh1 was generated with 5'-CCCTTGA-GACTCAACTATATT. METTL16-Sh2 was generated with 5'-ATGGCTGG-TATTTCCTCGCAA. METTL3-Sh was generated with 5'-GCCAAGGA ACA-ATCCATTGTT. METTL14-Sh was generated with 5'-CCATGTACTTACAAGCCGATA.

## Copper microplate assay

The Copper Microplate Assay Kit was purchased from Absin Bioscience (Shanghai, China). The GC tissue lysates extracted from 48 GC patients were used as the liquid samples. The experiment was performed according to the user manual.

## Immunoprecipitation (IP)

Immunoprecipitation was performed as previously described[65]. Briefly, cultured cells were lysed on the ice for 30 min using an IP lysis solution containing 1% protease inhibitor cocktail (MCE, USA). After centrifugation for 10 min, a 100 μL aliquot containing the total cellular proteins was incubated with 90 μL anti-FLAG M2 magnetic beads (Sigma-Aldrich, St. Louis, MO, USA) overnight at 4 °C. The precipitate was washed three times with lysis buffer and then boiled with SDS-PAGE loading buffer (NCM Biotech). The supernatant was extracted for immunoblotting using the appropriate antibodies.

## Immunofluorescence staining

GC cells transfected with point mutation plasmids were washed with phosphate-buffered saline (PBS), fixed in 4% paraformaldehyde at room temperature for 15 min, permeabilized with 0.5% Triton X-100 for 20 min and blocked with 3% BSA (PBS) for 30 min. Then the samples were incubated at 4 °C overnight with the primary antibody METTL16 (Abclonal Technology, Wuhan, China) diluted in 3% BSA (PBS) overnight. Subsequently, the cells were washed three times with PBS and incubated with fluorescein-labeled secondary antibodies diluted in PBS for 1 h at room temperature. The cells were stained and sealed with DAPI Fluoromount-G (Yeasen, Shanghai, China), and observed using a confocal microscope (Leica, Wetzlar, Germany).

## MeRIP-qPCR

The m6A modifications on *FDX1* mRNA were determined using the Ribo MeRIP m6A Transcriptome Profiling Kit (RiboBio, Canton, China).

Briefly, about 50–100 g total RNA was extracted and fragmented using RNA fragmentation buffer, and one-tenth of the fragmented RNA was preserved as the input group. Subsequently, 25 μL A/G magnetic beads were prewashed and mixed with 5 μg of anti-m⁶A antibody (supplied by the kit). The prepared MeRIP reaction buffer was added to the beads, and the mixture was incubated at 4 °C for 2 h followed by washing three times. The conjugate was eluted and purified using the Magen Hipure Serum/Plasma miRNA Kit (Magen, Canton, China) prior to qPCR analysis as described above. Specific primers were designed for MeRIP-qPCR analysis according to the motif-dependent m⁶A site predictor SRAMP (http://www.cuilab.cn/sramp) (*FDX1*-F: CCCTGGCT TGTTCAACCTGT; *FDX1*-R: CCCAACCGTGATCTGTCTGTTA). (*FDX1*-merip-602-F1: TGCAATCACTGATGAGGAGAA; *FDX1*-merip-602-R1: TGTTTCAGGCA-CTCGAACAG and *FDX1*-merip-820-F2: GGCAA-GACCTCCTGAACTAGAA; *FDX1*-merip-820-R2: TGCAAGGTGAAT-TAAAATAGTAAAGC). Relative enrichment of m⁶A was normalized to the input: %Input =1/10 × 2Ct [IP]−Ct [input].

### Methylated RNA immunoprecipitation sequencing (MeRIP-seq)
Methylated RNA immunoprecipitation sequencing was performed and the results were analyzed by Shanghai Genechem Co.,Ltd (Shanghai, China). In brief, three samples of m⁶A-modified mRNAs were extracted, captured and purified as described above, then underwent cDNA synthesis and PCR amplification for further RNA-seq analysis. Raw data were imported into the analysis process: *HISAT2* was used for mapping the unique reads, *exomePeak* was used for Peak Calling and RNA methylation differential analysis, and *HOMER* was used for searching motifs.

### RNA immunoprecipitation (RIP) assays
The binding between METTL16 and FDX1 was assessed using the RNA-Binding Protein Immunoprecipitation Kit (Magna, Shanghai, China). Briefly, A/G magnetic beads were incubated at 4 °C overnight with 5 μg of METTL16 antibody and subjected to immunoprecipitation. Then, the RNA was extracted and purified using the phenol-chloroform-ethanol method. The relative interaction between METTL16 and FDX1 was determined by qPCR later, as described above.

### RNA stability assays
GC cells were seeded into 6-well plates and treated with actinomycin D (5 μg/mL, Sigma-Aldrich, USA) at 1, 3, and 6 h, while DMSO (Sangon, Shanghai, China) was added to the negative control group at 0 h. Total RNA was isolated using the phenol-chloroform-ethanol method and then analyzed using qPCR. The mRNA half-live time was calculated according to the linear regression analysis[66].

### CRISPR/Cas9 knockout cell lines
The single guide RNAs (sgRNAs) targeting human METTL16, 5′-TAGGATGCCAGATATGCAAAGTC-3′ and 5′-AAACGACTTTGCATATCT GGCAT-3′, were designed with CRISPR Designer (http://crispr.mit.edu/) and cloned into plenty-sgRNA (puromycin resistant) and then were introduced into inducible Cas9 cell lines by lentivirus transduction. Cas 9 was induced using 1 μg/mL doxycycline hyclate for 3 days. To generate stable cell lines, we used doxycycline to knockout the non-essential genes for several days. After single-cell sorting, cells were expanded without doxycycline and clones were screened by immunofluorescence and Western blotting.

### In vitro lactylation assay
The purified AARS1 or AARS2 (5 μM) and METTL16 (10 μM) or unmodified peptide (ELEFVK(229)RIIHD, 10 μM, synthesized by Gl Biochem (Shanghai) Co., Ltd) were incubated in reaction buffer (100 mM HEPES, 100 mM MgCl₂, 10 mM KCl, and 0.1 mM lactyl-CoA (PH = 7, Shanghai Nafu Biotechnology Co., Ltd)) at 37 °C for 2 h. The reaction products were analyzed by Western blotting with anti-lactylation.

### Bioinformatic analysis
A total of 443 human tumor samples and 174 normal human gastric tissues were obtained from the TCGA (http://gdc-portal.nci.nih. gov) and GTEx (http://www.gtexportal.org) datasets that were available in public repositories. Gene expressions were analyzed using the R package 'limma' and 'DESeq2'. ClusterProfiler package was used to perform gene ontology annotation and gene enrichment analysis in R studio. The average optical density (AOD) values of IFC results were measured by Image J software according to the formula: AOD = Integrated density value/ fluorescent area, at a certain threshold.

### Ethical statement
We have complied with all relevant ethical regulations for animal testing and research. The research protocol was approved by the Animal Experiment Ethics Committee of Shanghai General Hospital.

### Statistics and reproducibility
Statistical analysis was carried out using two-tailed Student's *t* test, and the results are expressed as the mean ± standard deviation (SD) of three independent experiments. *P* values are indicated in the Figure legends. Microscopy images shown are representative of at least 5 fields from 3 independent experiments. Western Blot images are representative of three independent experiments. All biological and biochemical experiments were performed with appropriate internal negative and/or positive controls as indicated. All statistical analysis was performed and visualized using GraphPad Prism 9.4 software (GraphPad Software, La Jolla, CA, USA).

### Reporting summary
Further information on research design is available in the Nature Portfolio Reporting Summary linked to this article.

## Data availability
The original MeRIP-seq data generated in this study have been deposited in the GEO database under accession code: GSE224890. Other data generated in this study are provided in the Supplementary Information/Source Data file. Source data are provided with this paper.

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

## Acknowledgements

This work was supported by the National Natural Science Foundation of China (32270749, 82072662, 31970720, and 32070770), Shanghai Three-year Action Plan to Promote Clinical Skills and Clinical Innovation in Municipal Hospitals (SHDC2020CR4022), Shanghai Municipal Education Commission—Gaofeng Clinical Medicine Grant Support (the second round, 20191425).

## Author contributions

L.S., Y.Z., S.S., T.F., Z.C., T.Z., Y.L., and G.F. performed experiments. B.Y., Y.Z., S.S., P.Z., Z.L., and G.F. interpreted data and wrote the manuscript. Z.Q., G.F., and C.H. supervised the study and reviewed the manuscript.

## Competing interests

The authors declare no competing interests.
