## [Peer Review File · Nature Communications]

Lactylation of METTL16 promotes cuproptosis via m6A-modification on FDX1 mRNA in gastric cancerEditorial Note: Parts of this Peer Review File have been redacted as indicated to remove third-party material where no permission to publish could be obtained.

REVIEWER COMMENTS

Reviewer #1 (Remarks to the Author):

In this work Sun et al., show that Cu levels are high in some GC tumors correlating with poor patient outcome. They further show that METTL16 directly regulates FDX1 mRNA methylation increasing the levels of FDX1 protein translation. Cu promotes METTL16 activity by lactylation at lysine 229 a process that can be reversed by SIRT2. Ultimately the authors suggest that combining SIRT2 inhibition with ES can result in synergistic effects in a xenograft tumor model. This paper provides a new mechanism of FDX1 translation regulation with direct insights to specific METLL16 PTM that is regulated by SIRT2 that might be relevant in GC . This could have an impact on research conducted on mRNA m6A methylation and on mechanisms of FDX1 regulated copper induced cell death in specific cancers.

Overall, there is much elegant work showing the METLL16 regulations of FDX1 mRNA and how this process might be further regulated by SIRT2 de lactylation of METTL16. However, the way the manuscript is currently structured makes it very hard to follow and undermines the overall ability of the reader to appreciate the data and authors interpretation. Beyond the need to possibly restructure the outline there is also a need to provide better data describing the in vivo data experiments in the end. Tumor size at beginning of study and growth curves are required to enable proper interpretation of this experiment. Also, the experimental protocols including formulation, mode of administration of each of the compounds and precise protocol should be better described. Additional specific points are listed below.

Specific points:

1. I found the order of the manuscript somewhat confusing, and some restructuring might help.

Specific points:

- In figure 2, the authors show that HGC-27 Cells treated with CuCl₂ showed higher levels of M6A RNA methylation levels. Then they show that in TCGA there is an increase in specific methyltransferases in particular – METTL7B, METTL8, METTL16, METTL27. To go from cell line treated with Cu to tumors and METTL levels is very non intuitive and confusing.

- Did the authors measure the levels of m6A RNA methylation in the GC tumors analyzed for Cu levels? If there is increased m6A RNA methylation in these tumors this would be more intuitive to see why one would look at levels of METTL16 (as done in figure 2).

- The data in panel 3E alone is not convincing by itself. Should be combined with the data in Fig. 4 showing much more convincing regulation of FDX1 mRNA by METLL16. or additional data needs to be provided that quantifies or elaborates on the change in FDX1 mRNA modification that is described in 3E.

- Consider combining the rescue experiments (of ES-Cu and disulfiram) in Fig. 2 and 3 in one figure that will follow the description of METTL16 regulation of FDX1 mRNA. This will make it much more intuitive to understand the motivation of these experiments and interpretation. Otherwise, in KD of METTL16 resistance to Es-Cu, Dis-Cu where reduction in lip-DLAT levels is shown (Fig. 2) it is not clear why FDX1 levels are not shown.

-

2. In Fig. 3D, the list of genes that are affected should be better presented. Who are the genes that are most dramatically altered? Is the list presented in the figure the most significant? Or were they picked based on relevance to pathway? this needs to be clarified. Why isn't the whole list provided as supplementary data ? As there are more than 2000 genes changing this is important to clarify. Also did the authors validate the protein levels of any of the other genes in the list? This is a very important point in the manuscript and should be presented in a more quantified and analyzed manner with the data in the supplementary.

3. The way the mouse model experiments are currently described in figure 7 is insufficient. Growth curves need to be provided; the starting point of the experiment (tumor size before treatment) should be provided. Also, what was the formulation and method of administration of ES ? what was the protocol? All these need to be provided to enable proper data analysis.

4. The authors mentioned that high lactose levels induce the lactylation of METTL16 , however this

was not shown. Does increased copper induce an increase in intracellular lactate? How does Cu induce promote lactylation of METLL16 ?

Small points:

- Some data can go into supplementary such as : METLL16 KO clones in figure 2I
- Fig 3C what is the x -axis?
- Fig 3B there is a typo in METTL16
- In Fig 1A if the means are presented it would help
- Fig2F, 2G labeling on the figure not consistent with previous panels (sh1).
- More consistent coloring of graphs in Fig 3I-K would help (between the different panels).
- In 4C , need to elaborate what exactly is "FDX1 luciferase reporter" otherwise the reader needs to go to methods to learn it's the 3'UTR of FDX1 fused to luciferase, this should be explained in results and maybe a schematic in the figure itself would help. Moreover, all the data related to figure 4C-F is very poorly described in results and figure. It would benefit the reader if the authors used a diagram of what they did. Schematic of luc with the different fusions and also better explain in the text, what was the experiment , what was the result and what was the conclusions.

Reviewer #2 (Remarks to the Author):

Summary:

Sun et al. have found that copper content is increased in malignant gastric cancers, possibly inducing cuproptosis. They go on to find increased METTL16 activity in GC causes FDX1 mRNA stabilisation through m6A and this increased METTL16 activity is dependent on lactylation of METTL16. The results show that this mechanism can be exploited to improve therapy and can promote cuproptosis in gastric tumors. Overall the study identifies an interesting novel mechanism and very nicely outlines the molecular mechanism.

Critique:

The manuscript is well written, and the results follow a clear line of thought. The results are convincing, and the methods are well presented. The authors could discuss why METLL16 dependent m6A is the key m6A for FDX1 stability, why METTL16 is lactylated and how metabolic changes are involved. These might also be addressed in future studies.

Overall, I feel that the identified mechanism is relevant and of great relevance for the community.

Comments:

ABSTRACT: the sentence "Furthermore, we confirmed that copper stress promotes METTL16 lactylation at site K229, which could be delactylated by SIRT2, leading to cuproptosis." is a bit misleading – it is not directly clear whether SIRT2 delactylation leads to cuproptosis or METTL16 lactylation.

Fig 1E-G: Correlations are rather weak, meaning that the sentence "These results indicated that copper is an essential cofactor for biological functions and suggested its involvement in the progression and development of GC" might be a bit overstated

Fig 3: m6A is deposited by a larger number of different METTL proteins – several of which being expressed in GC. The changes shown indicate that METTL16 is not the only m6A transferase involved in FDX1 methylation. Why did the authors not repeat the MeRIP with the METTL16 KO lines and OE lines? Also, any indication of specificity of METTL16?

Fig3E: The effects on METTL16 shRNA on FDX1 m6A profiles are rather modest – how do the replicates look?

Reviewer #3 (Remarks to the Author):

Although cuproptosis has been proposed as a novel form of cell death, which is caused by excessively intracellular copper accumulation, the mechanisms of cuproptosis in gastric cancers remain unknown. This manuscript investigated the function of METTL16 in cuproptosis. The authors found copper content is high in gastric cancer and related to tumor progression. They showed that METTL16 is a driving force of cuproptosis via the m6A methylation on FDX1 mRNA. In addition, they found METTL16 is lactylated at site K229, which could be delactylated under copper stress, triggering cuproptosis. Further, they evaluated the therapeutic impact of SIRT2 inhibitor-AGK2 and copper ionophore drug-Elesclomol on gastric cancer cell lines in xenograft mouse models. They proposed the SIRT2-METTL16-FDX1-cuproptosis pathway as a promising therapeutic strategy in gastric cancers. However, there are several comments that need to be addressed.

Major concerns:

1. METTL16 is expressed in nucleus for methyltransferase activity. However, in this manuscript, METTL16 is located mainly in cytoplasm (see Figure 4H), how to be sure the specificity for METTL16? In Figure 5A-B and 2A-B, copper did not change the expression of METTL16, but increased the m6A level in gastric cancer cells. but there was an increased expression of METTL16 and higher copper content in gastric cancers. How to explain this discrepancy? If this is due to METTL16-K229-lac, what's the level of METTL16-K229-lac in gastric cancers? Further, histone lactylation is occurred in cell nucleus. Where does METTL16-K229-lac occur in the gastric cancers?
2. Although there is one reference showed that SIRT2 is highly expressed in gastric cancers, the expression of SIRT2 in gastric cancer is undetectable or at every low level in The Human Protein Database. It would be interesting to see in your panel, whether the expression of SIRT2 is increased or no change in gastric cancers? If there is no difference, the biological significance of AGK2 application in xenograft model need to be addressed carefully.
3. In Figure 5E, the authors claimed there are two acetylation sites on METTL16 protein sequence. SIRT2 is a typical deacetylation enzyme. What's the deacetylated function of SIRT2 on METTL16 in gastric cancers? Whether the deacetylation is more important than METTL16-lac in coprotopsis? Acetylated modification is a modulator of gene expression, it would be essential to know whether SIRT2 overexpression or knockdown or AGK2 treatment affects the expression of METTL16 in gastric cancer?

Minor questions:

1. In Figure 1I-J, the copper content was positively correlated with invasive and metastatic gastric cancer, as well as mucinous adenocarcinomas. Does the copper content can be considered as a prognostic factor?
2. For the dot blot for m6A assay in figure 2A, HPLC-MS would be recommended for quantification the m6A level in different concentration of copper-treated cells.
3. In Figure 3B, the m6A motif was not the classic RRACH, especially in METTL16-sh cells. There were nearly 7000 m6A-modified mRNAs in METTL16-knockdown cells, why A disappeared in m6A motif?
4. In Figure 3C, GO analysis of genes encoded by RNAs containing m6A modification in control cells was shown. What about GO analysis of those genes whose encoded by RNAs containing decreased m6A methylation after METTL16 knockdown? Furthermore, the number of enriched genes in GO analysis is too few to be significant.
5. In Figure 3E, how to explain there were so many peaks in FDX1 intron from the IGV?
6. The authors did not provide the information on primer sequences for m6A-IP-qPCR in figure 4A assays.
7. In Figure 4B, m6A regulates mRNA stability and translation in various cancers. Does METTL16 regulate FDX1 mRNA translation via m6A modification?
8. Given the region between amino acids 40-289 of METTL16 protein is responsible for methyltransferase activity, and METTL16 lactylation site K229 was located in this region, it would be important to see whether the mutated METTL16 lactylation site influenced its methyltransferase activity.

Reviewer #4 (Remarks to the Author):

In this manuscript, Sun et al, reported a novel mechanism underlying regulation of cuproptosis by METTL16 in gastric cancer. The authors found that copper is enriched in gastric cancer. Copper induces lactylation of METTL16, which in turn increases its methyltransferase activity. One of METTL16's substrates is FDX1, a key regulator of cuproptosis. METTL16 stabilizes the mRNA of FDX1 by increasing m6A modification, thus sensitizing gastric cancer cells to cuproptosis. For translational purposes, the authors used a combination of the copper ionophore drug- Elesclomol and AGK2, a SIRT2-specific inhibitor, to induce cuproptosis in gastric tumors, revealing the significance of METTL16 lactylation on regulating cuproptosis in tumors. The manuscript is generally well organized and clearly written. However, there are several major questions that need to be answered to strengthen their claims.

Major points:

There is lack of evidence or not sufficiently demonstrated for:

1. It is not clear whether METTL16 is the only methyltransferase that responds to copper and lactate. Please also refer to Minor point 3.
2. How is lactylation induced in copper enriched gastric cancers? Does copper directly induce lactate production? Why? Please also refer to Minor point 9.
3. Does lactylation directly enhance METTL16's methyltransferase activity? What is the mechanism?
4. Cuproptosis has been reported to rely on oxidative phosphorylation metabolism, rather than glycolysis. This is likely inconsistent with the intent to treat lactate-rich gastric cancer by inducing cuproptosis.

Minor points:

1. Fig.1, is there any explanation as to why gastric tumors are rich in copper?
2. Fig.2a, there is lack of a control showing an equal amount of RNA is loaded. Besides, LC/MS is suggested to accurately measure the difference in m6A levels.
3. Fig.2c, does shRNA-METTL3/14 affect sensitivity of cells to copper ionophore?
4. Fig.2h should be moved into Fig.3.
5. Fig.3a-d, is global m6A level changed by shRNA-METTL16?
6. Fig.4a, there is hardly any change in m6A shown in Fig.3e, while there is about 60% decrease of m6A shown in Fig.4a. Please explain this inconsistency.
7. Fig.4c, a description of FDX1-luciferase is needed in the main paper.
8. Fig.5, are the levels of lactylated METTL16 and m6A enriched in gastric tumors compared to normal tissue?
9. Fig.5f, does Cu²⁺ treatment result in intracellular lactate accumulation? How?
10. Fig.6-7, Sirt2 inhibitor has been reported to have anti-tumor effects, which may not be related to its de-lactylation activity. Alternatively, can LDH inhibitor compromise the effect of METTL16-mediate Cuproptosis? Or can inhibitors of oxidative phosphorylation sensitize the cells to METTL16-mediate Cuproptosis?

Jan 23, 2023

Re: Revision to *Nature communications*

Dear Reviewers:

It's our great pleasure to submit our revised manuscript entitled "Lactylation of METTL16 promotes cuproptosis via m⁶A-modification on *FDXI* mRNA in gastric cancer" (NCOMMS-22-41195) for publication in *Nature Communications*. First of all, we would like to thank the reviewers for the thoughtful comments and constructive suggestions, which helped to improve the quality of this manuscript. Based on the comments, we have incorporated changes into the manuscripts, as detailed below. Please also see below for a point-by-point response to the reviews' comments. We look forward to working with you and the reviewers to move this manuscript closer to the publication in *Nature Communications*.

Arrangement of Figures

Original	In Revision
Fig. 1A	Fig. 1A (revised) Fig. 2A (new)
Fig. 2A	Fig. 2B (relocated and revised)
Fig. 2F	Fig. 2F (revised)
Fig. 2G	Removed
Fig. 2H	Fig. 3H (relocated)
Fig. 2I	Supplemental Fig. S2C (relocated)
Fig. 3B-3D	Fig. 3B-3D (revised)
Fig. 3E	Fig. 4E (relocated and revised)
Fig. 3F-3H	Fig. 3E-3G (revised) Fig. 4B (new)
Fig. 4B-4D	Fig. 4C, 4D, and 4F (relocated) Fig. 4G (new)
Fig. 4E-4I	Fig. 4H-4K (relocated and 4K revised)

	Fig. 6A-C (new)
Fig. 6A-G and 6I-M	Fig. 6D-J and 6K-O (relocated)
Fig. 6H	Fig. 5M (relocated)
Fig. 7G and 7K	Fig. 7G and 7K (revised)
	Supplemental Fig. 2B (new)
Supplemental Fig. 2A-I	Supplemental Fig. S2D-L (relocated)
Supplemental Fig. 2D	Supplemental Fig. S2F (relocated and revised)
	Supplemental Fig. 2D-F (new)
	Supplemental Fig. S5 (new)
	Supplemental Fig. S6 (new)

REVIEWER COMMENTS

Responses to critiques (We used blue font to answer the questions):

Reviewer #1 (Remarks to the Author):

In this work Sun et al., show that Cu levels are high in some GC tumors correlating with poor patient outcome. They further show that METTL16 directly regulates FDX1 mRNA methylation increasing the levels of FDX1 protein translation. Cu promotes METTL16 activity by lactylation at lysine 229 a process that can be reversed by SIRT2. Ultimately the authors suggest that combining SIRT2 inhibition with ES can result in synergistic effects in a xenograft tumor model. This paper provides a new mechanism of FDX1 translation regulation with direct insights to specific METLL16 PTM that is regulated by SIRT2 that might be relevant in GC. This could have an impact on research conducted on mRNA m⁶A methylation and on mechanisms of FDX1 regulated copper induced cell death in specific cancers.

Overall, there is much elegant work showing the METLL16 regulations of FDX1 mRNA and how this process might be further regulated by SIRT2 delactylation of METTL16.

However, the way the manuscript is currently structured makes it very hard to follow and undermines the overall ability of the reader to appreciate the data and authors interpretation.

Beyond the need to possibly restructure the outline there is also a need to provide better data describing the in vivo data experiments in the end. Tumor size at beginning of study and growth curves are required to enable proper interpretation of this experiment.

Also, the experimental protocols including formulation, mode of administrative of each of the compounds and precise protocol should be better described.

Response to the comments of Reviewer #1

Many thanks for your kind and constructive comments and suggestions on our manuscript. We have studied the comments carefully and accordingly revised the manuscript seriously and thoroughly.

1. We have restructured the outline in the revised manuscript.
2. Tumor sizes in the initial stage and the growth curves were recorded during the experiments which were not presented in the primary manuscript. Therefore, we revised the data of Fig. 7G and 7K in revised manuscript.
3. We are sorry that we may have not expressed it clearly. We have re-write the portion of “Mouse xenograft tumor experiment” in reply to the supplement for “formulation, mode of administrative of each of the compounds and precise protocol” (Vide Page 20, lines 2-16).

Additional specific points are listed below.

Specific points:

I found the order of the manuscript somewhat confusing, and some restructuring might help.

Specific points:

- In figure 2, the authors show that HGC-27 Cells treated with CuCl₂ showed higher levels of m⁶A RNA methylation levels. Then they show that in TCGA there is an increase in specific methyltransferases in particular – METTL7B, METTL8, METTL16, METTL27. To go from cell line treated with Cu to tumors and METTL levels is very non intuitive and confusing.

Did the authors measure the levels of m⁶A RNA methylation in the GC tumors analyzed for Cu levels? If there is increased m⁶A RNA methylation in these tumors this would be more intuitive to see why one would look at levels of METTL16 (as done in figure 2).

Response: Thanks to the constructive comments. Indeed, we neglected to measure the levels of m⁶A RNA methylation in GC tumors. Thus, we further detected the m⁶A levels using dot blot and found that the m⁶A level was positive correlated with the Cu content in these GC tissues (Fig. 2A). In addition, it has been reported that total m⁶A levels were significantly increased in GC groups compared with benign gastric disease and healthy control patients [1], which is consistent with our study. Lastly, we rearranged the structure as shown on Figure 2 in revised manuscript.

The data in panel 3E alone is not convincing by itself. Should be combined with the data in Fig. 4 showing much more convincing regulation of FDX1 mRNA by METTL16. or additional data needs to be provided that quantifies or elaborates on the change in FDX1 mRNA modification that is described in 3E.

Response: Thanks for your constructive comments. The data in panel 3E alone couldn't lead to the regulation of FDX1 mRNA by METTL16, which should be combined with Figure 4. Considering that, we adjust the Fig. 3E as Fig. 4E. More intuitively, we changed a merged IGV picture showing the m⁶A abundances in METTL16 knockdown cells related to the control cells. In addition, we conducted another 2 groups of MeRIP-seq in Ctrl-Sh and METTL16-Sh GC cells, and the result further proved the conclusion that METTL16 regulates FDX1 mRNA modification. The original data were presented in Supplementary table 1.

Consider combining the rescue experiments (of ES-Cu and disulfiram) in Fig. 2 and 3 in one figure that will follow the description of METTL16 regulation of FDX1 mRNA. This will make it much more intuitive to understand the motivation of these experiments and interpretation. Otherwise, in KD of METTL16 resistance to Es-Cu, Dis-Cu where reduction in lip-DLAT levels is shown (Fig. 2) it is not clear why FDX1 levels are not shown.

Response: Thanks for your constructive comments. The rescue experiment in Fig. 2 were combined in Fig. 3. In additions, Fig. 2H in which METTL16 downregulates Lip-DLAT levels

was moved as Fig. 3H. Because FDX1 levels has been showed in Fig. 3F, it didn't show in Fig.3H. These make the study more intuitive to understand. Thanks again.

2. In Fig. 3D, the list of genes that are affected should be better presented. Who are the genes that are most dramatically altered? Is the list presented in the figure the most significant? Or were they picked based on relevance to pathway? this needs to be clarified. Why isn't the whole list provided as supplementary data? As there are more than 2000 genes changing this is important to clarify. Also did the authors validate the protein levels of any of the other genes in the list?

This is a very important point in the manuscript and should be presented in a more quantified and analyzed manner with the data in the supplementary.

Response: Thanks for your constructive comments.

1. We have showed that METTL16 plays a crucial role in cuproptosis (Fig. 2). To enlarge and confirm the enriched genes in GO analysis, we have repeated 2 groups of MeRIP-seq in Ctrl-Sh and METTL16-Sh GC cells. Genes with downregulated methylation levels as well as changed mRNA levels after METTL16 knockdown were chosen to perform GO analysis. The results showed that METLL16 involved in cuproptosis related pathways (new Fig. 3C). Then we identified 10 typical factors from cuproptosis-related pathway that were related to downregulated m⁶A methylation levels as well as downregulated mRNA levels in METTL16-sh group (Fig. 3D). qPCR analysis showed that the mRNA levels of *FDX1*, *MTF1*, *PDHB*, *ATP7A*, and *DLD* were decreased in METTL16-sh cells. However, FDX1 exhibited the most significantly decreased mRNA level in METTL16-sh cells versus the control cells (Fig. 3E and Supplementary Fig. S3D). Furthermore, the significant decreased protein level of FDX1 was detected in METTL16-Sh or -KO cells (Fig. 3F and 3G). Considering that FDX1 is the key regulators of copper ionophore-induced cell death, we presumed that METTL16-induced cuproptosis is FDX1-dependent.

2. We feel sorry that we neglect to provide the complete gene list as a supplementary file, which was attached below in new Supplementary table 1.

3. From the gene list, we choose MMP14 (downregulated gene) and PDCD6-AHRR (upregulated gene) and detected the protein expressions via western blot analysis. The results as follow showed that knockdown of METTL16 lead to MMP14 decreased while PDCD6-AHRR increased.

3. The way the mouse model experiments are currently described in figure 7 is insufficient. Growth curves need to be provided; the starting point of the experiment (tumor size before treatment) should be provided. Also, what was the formulation and method of administration of ES? what was the protocol? All these need to be provided to enable proper data analysis.

Response: Thanks for your suggestions. It is our negligence that we didn't provide the growth curves and tumor sizes before treatment. Tumor sizes in the initial stage and the growth curves were recorded during the experiments which were not presented in the primary manuscript. Therefore, we revised the data of Fig. 7G and 7K in revised manuscript. The specific protocol regarding the formulation and method of administration of AGK2 and Elesclomol was provided in methods (Vide Page 20, lines 2-16):

“AGK2 (MCE, USA) was dissolved in 100% DMSO at 10 mM stock concentration, while Elesclomol (MCE, USA) was diluted into 800 μ l of DMSO to reach a final concentration of 156.05 mM. Both of the compounds were diluted into a working solution at 10% concentration for better dissolution (the ingredients of the cosolvent contained 40% PEG300, 5% Tween-80 and 45% saline). When tumors reached a volume of 100 mm³, mice were intraperitoneally injected with the dissolved compounds (100 μ l per time), and the dose frequency was 3 times per week for 3 weeks for AGK2, while 4 times for 3 weeks for Elesclomol. The weights and volumes of tumors were measured and calculated after collection.”

This specific protocol could lead to a proper data analysis.

The authors mentioned that high lactose levels induce the lactylation of METTL16, however this was not shown. Does increased copper induce an increase in intracellular lactate? How does Cu induce promote lactylation of METLL16?

Response: Thanks to your vital and constructive comments. We found that high lactate levels induced the lactylation of METTL16-K229 in Fig. 5L (Page 11, line 17-19): “Both copper and lactate treatments induced strong METTL16 lactylation at K229 (Fig. 5K, 5L),”.

1. Theoretically, the higher the lactate content, the more lactic acid is available, increasing the lactylation of proteins. Thus, we examined the intracellular lactate in copper-treated GC cells using the lactic acid content detection kit (Solarbio, Beijing, China) and found no significant change between different groups with different copper concentrations (Supplementary Fig. 6A).

2. Then, we investigated whether copper mediated the regulation of lactoyltransferases on METTL16. Thus, the LC-MS/MS assay to identify the binding proteins of METTL16 was performed (Supplementary table 2). However, there is no binding lactoyltransferases, and it may result from the low abundance of proteins involved or insufficient bonding strength. Therefore, ten potential lactoyltransferases were selected, which has been reported or predicted, and were knockdown with specific siRNAs. These are as followed: are KAT3B (EP300),

CREBBP (CBP), ACSS2, AARS1, AARS2, KAT6A (MOZ), KAT8 (MOF), KAT5, NMT1, and NMT2. The results showed that just AARS1-si and AARS2-si decreased METTL16-K229 lactylation under copper treatment (Fig. 6A). Furthermore, copper promotes the interaction of AARS1 / AARS2 to METTL16 significantly (Fig. 6B). Overexpression of AARS1 and AARS2 promote the lactylation of METTL16-K229 indeed after copper treatment, indicating that AARS1 and AARS2 are both lactoyltransferases of METTL16 (Fig. 6C). Overall, we confirmed that copper promotes METTL16 lactylation is due to the increased interaction between AARS1 and AARS2 with METTL16.

Small points:

Some data can go into supplementary such as: METLL16 KO clones in figure 2I

Response: Thanks for your suggestion. We have moved METLL16 KO clones to supplementary data (The data of Fig. 2I were moved to Supplementary Fig. 2C).

Fig 3C what is the x -axis?

Response: Thanks for your question and we are sorry for ignoring the label of the x-axis in Fig. 3C. To enlarge and confirm the enriched genes in GO analysis, we have repeated 2 groups of MeRIP-seq in Ctrl-Sh and METTL16-Sh GC cells. Genes with downregulated methylation levels as well as changed mRNA levels after METTL16 knockdown were chosen to perform GO analysis (new Fig. 3C). The label of the x-axis is “Enrichment Score (-log₁₀ (p value))”.

Fig 3B there is a typo in METTL16

Response: Thanks very much for pointing out this mistake. We have corrected it in the figure.

In Fig 1A if the means are presented it would help

Response: Thanks for your suggestion. we have added means in Figure 1A.

Fig2F, 2G labeling on the figure not consistent with previous panels (sh1).

Response: Thanks for pointing out the problem. We have revised Fig2F and 2G labeling on the figure.

More consistent coloring of graphs in Fig 3I-K would help (between the different panels).

Response: Thanks for your suggestion! We have changed the consistent coloring of graphs in Fig. 3I-K.

- In 4C, need to elaborate what exactly is “FDX1 luciferase reporter” otherwise the reader needs to go to methods to learn it’s the 3’UTR of FDX1 fused to luciferase, this should be explained

in results and maybe a schematic in the figure itself would help. Moreover, all the data related to figure 4C-F is very poorly described in results and figure. It would benefit the reader if the authors used a diagram of what they did. Schematic of luc with the different fusions and also better explain in the text, what was the experiment, what was the result and what was the conclusions.

Response: Thank you for your vital suggestion.

1. We are sorry for the inadequate description of FDX1-luciferase. To construct FDX1-luciferase, CDS and 3'UTR regions of FDX1 were constructed and fused into the downstream of pmirGLO dual-luciferase vector (Promega, E1330), while FDX1-602-Mut or FDX1-820-Mut were constructed by replacing adenosines (A) in m⁶A motif with thymine (T). For dual-luciferase reporter assay, cells seeded in 24-well plates were co-transfected with wild-type or mutant FDX1-luc and METTL16. After 48 h post transfection, the activities of firefly luciferase and Renilla luciferase in each well were determined by a Dual-Luciferase Reporter Assay System (Promega, E1910) according to the manufacturer's protocol. In addition, we have added the schematic into Fig. 4G to make it intuitional.

2. We are also sorry for all the data related to Fig. 4C-F is very poorly described in results and figure. We have elaborated the results in revised manuscript (Page 9-10).

Reviewer #2 (Remarks to the Author):

Summary:

Sun et al. have found that copper content is increased in malignant gastric cancers, possibly inducing cuproptosis. They go on to find increased METTL16 activity in GC causes FDX1 mRNA stabilisation through m6A and this increased METTL16 activity is dependent on lactylation of METTL16. The results show that this mechanism can be exploited to improve therapy and can promote cuproptosis in gastric tumors. Overall the study identifies an interesting novel mechanism and very nicely outlines the molecular mechanism.

Critique:

The manuscript is well written, and the results follow a clear line of thought. The results are convincing, and the methods are well presented.

The authors could discuss why METLL16 dependent m6A is the key m6A for FDX1 stability, why METTL16 is lactylated and how metabolic changes are involved. These might also be addressed in future studies.

Overall, I feel that the identified mechanism is relevant and of great relevance for the community.

Response to the comments of Reviewer #2

Response: Thanks for your constructive comments, and we would like to discuss the questions above in the Discussion part:

There are mainly two sets of m⁶A RNA methyltransferases with verified mRNA targets in humans: METTL3/14/WTAP complex and METTL16. Although identified targets of METTL16 are limited until now, the roles of METTL16 as a potential target in many pathologic processes are emerging. There is a large similarity in protein structures between these two enzymes [2], but some differences still remain, suggesting that they have unique responsibilities in cells. In our study, we found that METTL16 was significantly elevated in gastric cancer tissues, while METTL3 and METTL14 had no significant difference (Supplementary Fig. 2A). Further study showed that METTL16 is the methyltransferase of FDX1 by regulating the stability of FDX1-mRNA. In addition, we also found that the mRNA level of FDX1 is unchanged in METTL3 or METTL14 knockdown cells (Supplementary Fig. 2B). there no significant difference (slight change) between Ctrl-Sh and METTL3- or METTL14-knockdown cells in the sensitivity of cells to copper ionophore (The Figs of below data). The above results prove that METTL16 mediated m⁶A modification is the key m⁶A modification for FDX1 stability.

Fig. The effects of METTL3 and METTL14 in regulating cuproptosis. (A) qRT-PCR verification of METTL3-Sh and METTL14-Sh efficiencies. (B) Growth survival rate of Ctrl, METTL3-Sh, and METTL14-Sh cells after treatment with indicated Elesclomol-Cu concentrations. Statistical data presented in this figure show mean \pm SD from $n \geq 3$ independent experiments, * $P < 0.05$, ** $P < 0.01$, *** $P < 0.001$, n.s. = No Significance, Student's t-test.

Lactate functions as interlinkage between glycolysis and oxidative phosphorylation. Whatever in aerobic or anaerobic conditions, high level of glycolysis is present in almost all tumor types comparing to normal tissues [3]. As a result, lactate was produced, thus providing sufficient substrate for the lactylation of proteins. In addition, lactate can be used to support oxidative phosphorylation [4]. Interestingly, oxidative phosphorylation is upregulated in certain cancers, including leukemias, lymphomas, pancreatic ductal adenocarcinoma, and endometrial carcinoma, and can occur even in the face of active glycolysis [5]. The decreases in mtDNA content have been observed in gastric cancers, showing reduced oxidative capacity [6]. MSI/MSS sig-high gastric cancer is characterized by oxidative phosphorylation [7]. This suggests that some gastric cancers

with high level of oxidative phosphorylation may be suitable to be treated through cuproptosis.

Comments:

ABSTRACT: the sentence “Furthermore, we confirmed that copper stress promotes METTL16 lactylation at site K229, which could be delactylated by SIRT2, leading to cuproptosis.” is a bit misleading – it is not directly clear whether SIRT2 delactylation leads to cuproptosis or METTL16 lactylation.

Response: Thank you for your suggestion. We are sorry for the misleading expression. We reconstruct the sentence and replace it with “Furthermore, we confirmed that copper stress promotes METTL16 lactylation at site K229 followed by cuproptosis. The process of METTL16 lactylation was inhibited by SIRT2”.

Fig 1E-G: Correlations are rather weak, meaning that the sentence “These results indicated that copper is an essential cofactor for biological functions and suggested its involvement in the progression and development of GC” might be a bit overstated

Response: Thanks for the thoughtful comment. We rewrote the sentence as “These results indicated that the high content of copper is involved in the progression and development of GC.

Fig 3: m⁶A is deposited by a larger number of different METTL proteins – several of which being expressed in GC. The changes shown indicate that METTL16 is not the only m⁶A transferase involved in FDX1 methylation. Why did the authors not repeat the MeRIP with the METTL16 KO lines and OE lines? Also, any indication of specificity of METTL16?

Response: Thanks to your constructive comments. To clarify it, we repeat the MeRIP-qPCR in METTL16 KO and OE lines and found that m⁶A levels on *FDXI* mRNA is significantly decreased in METTL16 knockout cells, and significantly increased in cells transfected with METTL16-overexpression plasmid (Fig. 4B). There are mainly two sets of m⁶A RNA methyltransferases with verified mRNA targets in humans: METTL3/14/WTAP complex and METTL16. We found that the mRNA level of FDX1 is unchanged in METTL3 or METTL14 knockdown cells (Supplementary Fig. 2B). In addition, there is no significant difference (slight change) between Ctrl-Sh and METTL3-Sh or METTL14-Sh cells in the sensitivity of cells to copper ionophore (The Figs of below data). The results indicated that METTL16 is the primary methyltransferase responsible for FDX1 methylation.

Fig. The effects of METTL3 and METTL14 in regulating cuproptosis. (A) qRT-PCR verification of METTL3-Sh and METTL14-Sh efficiencies. **(B)** Growth survival rate of Ctrl, METTL3-Sh, and METTL14-Sh cells after treatment with indicated Elesclomol-Cu concentrations. Statistical data presented in this figure show mean \pm SD from $n \geq 3$ independent experiments, * $P < 0.05$, ** $P < 0.01$, *** $P < 0.001$, n.s. = No Significance, Student's t-test.

Fig3E: The effects on METTL16 shRNA on FDX1 m⁶A profiles are rather modest – how do the replicates look?

Response: Thank you for your suggestion. More intuitively, we changed a merged IGV picture showing the m⁶A abundances in METTL16 knockdown cells related to the control cells. And the picture showed that CDS and 3'UTR region of m⁶A levels of *FDX1* mRNA were inhibited significantly in stable METTL16-sh cell lines (Fig. 4E). In addition, we conducted another 2 groups of MeRIP-seq in Ctrl and METTL16-Sh GC cells, and the result further proved the conclusion that METTL16 regulates FDX1 mRNA modification. The original data were presented in Supplementary table 1.

Reviewer #3 (Remarks to the Author):

Although cuproptosis has been proposed as a novel form of cell death, which is caused by excessively intracellular copper accumulation, the mechanisms of cuproptosis in gastric cancers remain unknown. This manuscript investigated the function of METTL16 in cuproptosis. The authors found copper content is high in gastric cancer and related to tumor progression. They showed that METTL16 is a driving force of cuproptosis via the m⁶A methylation on FDX1 mRNA. In addition, they found METTL16 is lactylated at site K229, which could be delactylated under copper stress, triggering cuproptosis. Further, they evaluated the therapeutic impact of SIRT2 inhibitor-AGK2 and copper ionophore drug-Elesclomol on gastric cancer cell lines in xenograft mouse models. They proposed the SIRT2-METTL16-FDX1-cuproptosis pathway as a promising therapeutic strategy in gastric cancers. However, there are several comments that need be addressed.

Major concerns:

1. METTL16 is expressed in nucleus for methyltransferase activity. However, in this manuscript, METTL16 is located mainly in cytoplasm (see Figure 4H), how to be sure the

specificity for METTL16? In Figure 5A-B and 2A-B, copper did not change the expression of METTL16, but increased the m6A level in gastric cancer cells. but there was an increased expression of METTL16 and higher copper content in gastric cancers. How to explain this discrepancy? If this is due to METTL16-K229-lac, what's the level of METTL16-K229-lac in gastric cancers? Further, histone lactylation is occurred in cell nucleus. Where does METTL16-K229-lac occur in the gastric cancers?

Response to the comments of Reviewer #3

Response: Thank you for your thoughtful comments. We agree that this is an important area which requires further research.

1. METTL16 is situated mostly in the nucleus for methyltransferase activity. We furtherly checked the location of METTL16 carefully in GC tissues. Interestingly, METTL16 locates mostly in the nucleus in tumor tissues (Red box, in Fig. A of below data), but has a strong staining in the circular structure of the stomach (Green box, in Fig. A of below data). The circular structure is the glands of the stomach. Columnar epithelium of gastric glands consists of foveolar cells, parietal cells, chief cells, and, enteroendocrine cells, which play an essential role in digestion and preventing the stomach from being corroded by secreting mucus, gastric acid, intrinsic factor, pepsinogen, gastric lipase, cholecystokinin and so on. That implies that METTL16 may function in other aspects. The strong staining in the glands had misled us about the main METTL16 in GC tissues in old fig4H. To verify our idea, the anti-METTL16 antibody were used to stain endogenous METTL16 in gastric cancer HGC-27 cells and confirmed the nuclear location of METTL16 (Fig. B of below data / Supplementary Fig. 3E). In addition, overexpression of Flag-METTL16 -WT, -K229R, and -K229E in HGC-27 cells also showed nuclear localization, indicated METTL16-K229 Lactylation does not affect the location of METTL16 (Fig. C of below data / Supplementary Fig. 5A). Lastly, we revised the Fig. 4H using DAB staining that better represented the location of nucleus or cytoplasm, and recalculated the statistics (in new Fig. 4H).

Fig. The cellular localization of METTL16 and mutants. (A) Immunohistochemical staining showing the cellular localization of METTL16 in GC tissues. The red frame showed the nucleus localization of METTL16 in the parenchyma. The green frame showed the strong staining of METTL16 in the gastric glands. (B) Immunofluorescent staining showing the endogenous localization of METTL16 in HGC-27 cells (scale bar, 50 μ m). (C) Immunofluorescent staining showing the nucleus localization of METTL16-WT, -K229R, -K229E in Flag-METTL16-OE and GFP co-transfected cells (scale bar, 10 μ m).

2. We would detect the level of METTL16-K229-lac in gastric cancers, but it is limited by the quality of current METTL16-K229-lac antibody. A clear single strip would appear only after undergoing Flag-METTL16 immunoprecipitation, while jumbly strips emerge without taking immunoprecipitation. It is depressing that there were still jumbly strips using METTL16-K229-lac antibody in WB assay, even though we purified it. Considering the facts mentioned above, we cannot detect the level of METTL16-K229-lac in GC tissues at the moment. Thus, we can't utilize the antibody to make correlation analysis between METTL16-K229-lac and the copper concentration in GC tissues. We will try to optimize and purify the anti-K229-lac antibody, and establish the correlation in GC tissues in future study.

3. The immunoprecipitation of overexpression of Flag-METTL16 -WT, -K229R, and -K229E showed that METTL16-K229 Lactylation does not affect the nuclear location of METTL16 (Fig. C of above data). In the study, there is no data showed the correlation analysis between METTL16 expression and higher copper content in GCs, and it will be explored in the future.

2. Although there is one reference showed that SIRT2 is highly expressed in gastric cancers, the expression of SIRT2 in gastric cancer is undetectable or at every low level in The Human Protein Database. It would be interesting to see in your panel, whether the expression of SIRT2

is increased or no change in gastric cancers? If there is no difference, the biological significance of AGK2 application in xenograft model need to be addressed carefully.

Response: Thank you for your thoughtful comments. We furtherly detected SIRT2 expression in the tissue microarray of GC patients, and found that SIRT2 expression could be detected in GCs, especially in poor differentiated GCs using fresh surgical gastric cancer tissues (Fig. A of below data), which maybe as a supplement of databases like Human Protein Atlas and TCGA. In addition, endogenic SIRT2 expression could be detected in several GC cell lines, such as AGS, GT-38, SNU-1, and GES cells, some of which show higher SIRT2 expression than GES (Gastric epithelial cells) (Fig. B of below data). In all, SIRT2 still existed in some GCs, and the application of AGK2 is suitable for these GCs. In the study, activators or inhibitors of other proteins might also be important to promote new intervention strategies for cancer treatment.

Fig. SIRT2 expression in GC tissues and GC cells. (A) Immunofluorescent staining showing SIRT2 expression in well differentiated, moderately differentiated, and poorly differentiated tissues in the GC TMA images. The red frames showed the amplified images of typical staining regions (right, 30x microscopy) from the global staining regions (left, 7x microscopy). Scale bar, 50 μ m. **(B)** Western blot analysis of SIRT2 expression in AGS, GT-38, GES and SNU-1 GC cells.

3. In Figure 5E, the authors claimed there are two acetylation sites on METTL16 protein sequence. SIRT2 is a typical deacetylation enzyme. What's the deacetylated function of SIRT2 on METTL16 in gastric cancers? Whether the deacetylation is more important than METTL16-lac in coprotopsis? Acetylated modification is a modulator of gene expression, it would be essential to know whether SIRT2 overexpression or knockdown or AGK2 treatment affects the expression of METTL16 in gastric cancer?

Response: Thanks for your suggestion. This is an important area requiring further research. SIRT2, as a deacetylation enzyme, may also regulated METTL16 deacetylation. The results showed that total METTL16 acetylation had a partial reduction after SIRT2 overexpression (Supplementary Fig. 6D). In the study, we found METTL16 acetylation sites K171 and K348 by LC-MS/MS analysis, and further constructed their acetylation-defective or -activated mutants (K to R or Q) and verified their expression with Flag-tag (Supplementary Fig. 6E). There is no significant difference on increased mRNA level of FDX1 after METTL16-WT, -K171R, -K171Q, -K348R, and -K348Q overexpression (Supplementary Fig. 6F), indicating that SIRT2-mediated METTL16 deacetylation did not affect the FDX1 mRNA levels. In addition, the mRNA and protein levels of METTL16 did not change after SIRT2-knockdown or AGK2-treatment (Supplementary Fig. 6G and 6H).

Minor questions:

1. In Figure 1I-J, the copper content was positively correlated with invasive and metastatic gastric cancer, as well as mucinous adenocarcinomas. Does the copper content can be considered as a prognostic factor?

Response: Thanks for your suggestion. The copper content is positively correlated with invasive indicators like lymph node metastasis and vascular invasion in invasive and metastatic gastric cancer. It is also negatively correlated with the overall survival and disease-free survival of GC patients. However, the copper content has no significant relevance with invasion or metastasis in common GC tissues (Supplementary Figure 1A-D). Thus, only by increasing the quantities and types of GC, we can provide accurate and stable foundation for whether and for which kind of gastric cancer could the copper content be the prognostic factor in our future study.

2. For the dot blot for m6A assay in figure 2A, HPLC-MS would be recommended for quantification the m6A level in different concentration of copper-treated cells.

Response: Thank you for your constructive comment. HPLC/MS is needed for accurately measuring the difference in m⁶A levels. We have treated the GC cells with different copper concentrations at 0 μ M, 1 μ M, and 5 μ M, and sent them for HPLC-MS analysis in LC. bio (Shanghai, <http://www.lc-bio.com/>). However, the process of HPLC-MS is delayed by the massive COVID epidemic across China. We promise to offer the results once we receive the analysis report. Below information is our contracts with the company about the HPLC-MS analysis.

[REDACTED]

3. In Figure 3B, the m6A motif was not the classic RRACH, especially in METTL16-sh cells. There were nearly 7000 m6A-modified mRNAs in METTL16-knockdown cells, why A disappeared in m6A motif?

Response: Thanks for your interesting question. We are very sorry that this typical motif was omitted due to our negligence. After carefully analysis of the sequencing data from our sequence data and the other two replicates, we found that there were several motifs of METTL16, including classic motif-GGAC and non-classical motif-GACA. Indeed, the classic motif-GGAC was still in the first place in both the control and METTL16-sh cells. And we have updated the motif sequence in the revised manuscript.

4. In Figure 3C, GO analysis of genes encoded by RNAs containing m6A modification in control cells was shown. What about GO analysis of those genes whose encoded by RNAs containing decreased m6A methylation after METTL16 knockdown? Furthermore, the number of enriched genes in GO analysis is too few to be significant.

Response: Thank you for your constructive suggestion. To enlarge and confirm the enriched genes in GO analysis, we have repeated 2 groups of MeRIP-seq in Ctrl-Sh and METTL16-Sh GC cells. Genes with downregulated methylation levels as well as changed mRNA levels after METTL16 knockdown were chosen to perform GO analysis. The results showed that METLL16 involved in cuproptosis related pathways (new Fig. 3C).

5. In Figure 3E, how to explain there were so many peaks in FDX1 intron from the IGV?

Response: Thank you for your comment. We admit that there were so many peaks in FDX1 intron in the old figure, and we consider them as non-specific peaks enriched in the process of MeRIP-seq, due to the low levels of exonic peaks enriched by METTL16 immunoprecipitation. To promote the quality of enrichment, we repeat the experiment in LC-SCIENCE using two pairs of new METTL16-Ctrl and METTL16-Sh samples, and re-analyze the results. The new results showed few peaks in FDX1 intron since significant enrichment in the 3'UTR and CDS

regions (Fig. 4E).

6. The authors did not provide the information on primer sequences for m⁶A-IP-qPCR in figure 4A assays.

Response: Thank you for your suggestion. We feel sorry for neglecting that. In the Methods part, we added the primer sequences: F: 5'-CCCTGGCTTGTTCAACCTGT, and R: 5'-CCC-AACCGTGATCTGTCTGTTA. The PCR product using the primers contains the site 602 in *FDX1* CDS (page 26, 19-20).

7. In Figure 4B, m⁶A regulates mRNA stability and translation in various cancers. Does METTL16 regulate *FDX1* mRNA translation via m⁶A modification?

Response: Thank you for your suggestion. The mRNA translation occurs in cytoplasm, and it has been reported that METTL16 can relocate in cytoplasm and promote mRNA translation [8]. In the study, we found that endogenous METTL16 and exogenous Flag-METTL16-WT, -K229R and -K229R are all located in the nucleus in GC tumor cells (Supplementary Fig. 3E and Supplementary Fig. 5A). In the cell nucleus, METTL16 functions as an m⁶A writer to deposit m⁶A into specific RNA targets. In addition, the result showed that the decay rate of *FDX1* mRNA was faster in METTL16-knockdown cells than in control cells when transcription was halted with actinomycin D (ActD) (Fig. 4C). Thus, we confirmed that METTL16 regulate *FDX1* mRNA stability via m⁶A modification.

8. Given the region between amino acids 40-289 of METTL16 protein is responsible for methyltransferase activity, and METTL16 lactylation site K229 was located in this region, it would be important to see whether the mutated METTL16 lactylation site influenced its methyltransferase activity.

Response: Thank you for your constructive suggestion. We agree that this is an important area that requires further research. Thus, we analyzed potential impact of K229 lactylation on METTL16. Crystal structure of METTL16 complexed with MAT2A RNA hairpin (PDB 6DU5), showed that K229 formed a salt bridge with E226 and K163 from the autoinhibition loop (pink) was inserted in the SAM binding site of METTL16 (Supplementary Fig. 5B). Our structure modelling indicated that R230 could stabilize the autoinhibited state by forming hydrogen bonds with K163 or Q162 of the loop. Lactylation of K229 will abolish its salt bridge with E226, which may lead to conformational changes in sidechains of E226 and allow new salt bridge formation with R230 (Supplementary Fig. 5C). This would weaken the stability of the autoinhibited state leading to elevated enzymatic activity of METTL16. The structure is shown in cartoon diagram with METTL16 colored in green and RNA in orange. The side chains of

K163, E226, K229, and R230 are shown in sticks with nitrogen atoms in blue and oxygen in red.

Then, the downstream targets of METTL16, such as BCAT1, BCAT2 [9], and GPX4 [10] were selected to verify the effect of METTL16-K229 lactylation on its methyltransferase activity. The result showed that METTL16-K229E promoted the methyltransferase activity (Supplementary Fig. 5D).

Reviewer #4 (Remarks to the Author):

In this manuscript, Sun et al, reported a novel mechanism underlying regulation of cuproptosis by METTL16 in gastric cancer. The authors found that copper is enriched in gastric cancer. Copper induces lactylation of METTL16, which in turn increases its methyltransferase activity. One of METTL16's substrates is FDX1, a key regulator of cuproptosis. METTL16 stabilizes the mRNA of FDX1 by increasing m⁶A modification, thus sensitizing gastric cancer cells to cuproptosis. For translational purposes, the authors used a combination of the copper ionophore drug– Elesclomol and AGK2, a SIRT2-specific inhibitor, to induce cuproptosis in gastric tumors, revealing the significance of METTL16 lactylation on regulating cuproptosis in tumors. The manuscript is generally well organized and clearly written. However, there are several major questions that need to be answered to strengthen their claims.

Major points:

There is lack of evidence or not sufficiently demonstrated for:

1. It is not clear whether METTL16 is the only methyltransferase that responds to copper and lactate. Please also refer to Minor point 3. shRNA-METTL3/14 affect sensitivity of cells to copper ionophore

Response: Thank you for your thoughtful comments. There are mainly two sets of m⁶A RNA methyltransferases with verified mRNA targets in humans: METTL3/14/WTAP complex and METTL16. Indeed, METTL3 has been recently reported to function as methyltransferase in high lactate circumstance. Lactate accumulated in tumor microenvironment could lead to METTL3 upregulation and lactylation [11]. The lactylation-driven METTL3-mediated RNA m⁶A modification promoting immunosuppressive capacity of Tumor-infiltrating myeloid cells [11]. However, the roles of METTL3 or METTL14 in copper stress are not elucidated yet. To explore whether METTL3 or METTL14 involved in cuproptosis, shMETTL3 and shMETTL14 cells were constructed and treated with Elesclomol and copper in a 1:1 ratio, then cell viability was detected by CCK8. The results showed that there no significant difference (slight change) between Ctrl-Sh and METTL3-Sh or METTL14-Sh cells in the sensitivity of cells to copper ionophore (The Figs of below data). In addition, the mRNA level of FDX1 is

unchanged in METTL3 or METTL14 knockdown cells (Supplementary Fig. 2B). The results indicated that METTL16 is the primary methyltransferase responsible for FDX1 methylation.

Fig. The effects of METTL3 and METTL14 in regulating cuproptosis. (A) qRT-PCR verification of METTL3-Sh and METTL14-Sh efficiencies. (B) Growth survival rate of Ctrl, METTL3-Sh, and METTL14-Sh cells after treatment with indicated Elesclomol-Cu concentrations. Statistical data presented in this figure show mean \pm SD from $n \geq 3$ independent experiments, * $P < 0.05$, ** $P < 0.01$, *** $P < 0.001$, n.s. = No Significance, Student's t-test.

2. How is lactylation induced in copper enriched gastric cancers? Does copper directly induce lactate production? Why? Please also refer to Minor point 9. does Cu^{2+} treatment result in intracellular lactate accumulation? How?

Response: Thanks to your vital and constructive comments.

1. Theoretically, the higher the lactate content, the more lactic acid is available, increasing the lactylation of proteins. Thus, we examined the intracellular lactate in copper-treated GC cells using the lactic acid content detection kit (Solarbio, Beijing, China) and found no significant change between different groups with different copper concentrations (Supplementary Fig. 6A).

2. Then, we investigated whether copper mediated the regulation of lactoyltransferases on METTL16. Thus, the LC-MS/MS assay to identify the binding proteins of METTL16 was performed (Supplementary table 2). However, there is no binding lactoyltransferases, and it may result from the low abundance of proteins involved or insufficient bonding strength. Therefore, ten potential lactoyltransferases were selected, which has been reported or predicted, and were knockdown with specific siRNAs. These are as followed: are KAT3B (EP300), CREBBP (CBP), ACSS2, AARS1, AARS2, KAT6A (MOZ), KAT8 (MOF), KAT5, NMT1, and NMT2. The results showed that just AARS1-si and AARS2-si decreased METTL16-K229 lactylation under copper treatment (Fig. 6A). Furthermore, copper promotes the interaction of AARS1 / AARS2 to METTL16 significantly (Fig. 6B). Overexpression of AARS1 and AARS2 promote the lactylation of METTL16-K229 indeedly after copper treatment, indicating that AARS1 and AARS2 are both lactoyltransferases of METTL16 (Fig. 6C). Overall, we confirmed that copper promotes METTL16 lactylation is due to the increased interaction between AARS1 and AARS2 with METTL16.

3. There is no activator of AARS1 and AARS2, and the interaction of SIRT2 and METTL16 just had a part decrease after copper treatment (Supplementary Fig. 6C). Thus, SIRT2 inhibitor

AGK2 is the suitable chemical drug to furtherly complete the related mechanism research in the study.

3. Does lactylation directly enhance METTL16's methyltransferase activity? What is the mechanism?

Response: Thank you for your constructive suggestion. We agree that this is an important area that requires further research. Thus, we analyzed potential impact of K229 lactylation on METTL16. Crystal structure of METTL16 complexed with MAT2A RNA hairpin (PDB 6DU5), showed that K229 formed a salt bridge with E226 and K163 from the autoinhibition loop (pink) was inserted in the SAM binding site of METTL16 (Supplementary Fig. 5B). Our structure modelling indicated that R230 could stabilize the autoinhibited state by forming hydrogen bonds with K163 or Q162 of the loop. Lactylation of K229 will abolish its salt bridge with E226, which may lead to conformational changes in sidechains of E226 and allow new salt bridge formation with R230 (Supplementary Fig. 5C). This would weaken the stability of the autoinhibited state leading to elevated enzymatic activity of METTL16. The structure is shown in cartoon diagram with METTL16 colored in green and RNA in orange. The side chains of K163, E226, K229, and R230 are shown in sticks with nitrogen atoms in blue and oxygen in red.

Then, the downstream targets of METTL16, such as BCAT1, BCAT2 [9], and GPX4 [10] were selected to verified the effect of METTL16-K229 lactylation on the its methyltransferase activity. The result showed that METTL16-K229E promoted the methyltransferase activity (Supplementary Fig. 5D).

4. Cuproptosis has been reported to rely on oxidative phosphorylation metabolism, rather than glycolysis. This is likely inconsistent with the intent to treat lactate-rich gastric cancer by inducing cuproptosis.

Response: Thank you for your question.

1. Whatever in aerobic or anaerobic conditions, high level of glycolysis is present in almost all tumor types comparing to normal tissues [3]. As a result, lactate was produced, thus providing sufficient substrate for the lactylation of proteins. In addition, lactate can be used to support oxidative phosphorylation [4]. Lactate functions as interlinkage between glycolysis and oxidative phosphorylation.

2. Interestingly, oxidative phosphorylation is upregulated in certain cancers, including leukemias, lymphomas, pancreatic ductal adenocarcinoma, and endometrial carcinoma, and can occur even in the face of active glycolysis [5].

3. Although the decreases in mtDNA content have been observed in gastric cancers, showing reduced oxidative capacity [6], MSI/MSS sig-high gastric cancer is characterized by oxidative phosphorylation [7], indicated that metabolic states are distinct in different gastric cancer types. Consistently, we detected gastric cancer tissues of different patients and found that HIF1a varied

significantly, indicating the status of oxidative phosphorylation in gastric cancer is different (Figs. of below data). This suggests that some gastric cancers with high level of oxidative phosphorylation may be suitable to be treated through cuproptosis.

4. In this study, we found that lactated METTL16 participated in copper death by regulating FDX1, and the lactation of METTL16 mediated by copper ions was the key reason of cuproptosis in gastric cancer cells. Overall, cuproptosis induction has become a promising therapeutic strategy for GC.

Figs. Western blot analysis of HIF1a expression in different GC. C: Cancer tissues, N: Normal tissues.

Minor points:

1. Fig.1, is there any explanation as to why gastric tumors are rich in copper?

Response: Thank you for your suggestion. We are sorry that there is no research explaining the enrichment of copper in gastric tumors. It may be relevant to the higher expression of the copper channel proteins, such as copper influx transporter 1 encoded by SLC31A1. TCGA and GTEx analysis demonstrates higher expression of SLC31A1 in gastric cancer compared with gastric mucosa (<http://gepia2.cancer-pku.cn/#analysis>), which may partly explain the enrichment of copper in gastric cancer. In addition, copper is mainly absorbed through the stomach and upper small intestine, and the SLC31A1 expression is important for copper absorption in gastric mucosa.

2. Fig.2a, there is lack of a control showing an equal amount of RNA is loaded. Besides, LC/MS is suggested to accurately measure the difference in m⁶A levels.

Response: Thank you for your constructive comment. We have added the control showing an equal amount of RNA using the original samples in the revised Fig. 2B. In addition, HPLC/MS is needed for accurately measuring the difference in m⁶A modification levels. We have treated the GC cells with different copper concentrations at 0 μM, 1 μM, and 5 μM, and sent them for HPLC-MS analysis in LC. bio (Shanghai, <http://www.lc-bio.com/>). However, the process of HPLC-MS is delayed by the massive COVID epidemic across China. We promise to offer the results once we receive the analysis report. Below information is our contracts with the company about the HPLC-MS analysis.

[REDACTED]

3. Fig.2c, does shRNA-METTL3/14 affect sensitivity of cells to copper ionophore?

Response: Thank you for your suggestion. The results showed that there no significant difference (slight change) between Ctrl-Sh and METTL3-Sh or METTL14-Sh cells in the sensitivity of cells to copper ionophore (The Figs of above data).

4. Fig.2h should be moved into Fig.3.

Response: Thank you for your suggestion. we have adjusted our study as per the comments that the Fig. 2H were moved as Fig. 3G.

5. Fig.3a-d, is global m⁶A level changed by shRNA-METTL16?

Response: Thank you for your suggestion. The global m⁶A level is down-regulated in METTL16-Sh samples, as shown in new Supplementary Fig. 3A.

6. Fig.4a, there is hardly any change in m⁶A shown in Fig.3e, while there is about 60% decrease of m⁶A shown in Fig.4a. Please explain this inconsistency.

Response: Thank you for your question. Firstly, we changed a merged IGV picture showing the m⁶A abundances in Ctrl-Sh and METTL16-Sh samples to intuitively show the distinction of total m⁶A modification on *FDX1* mRNA between Ctrl-Sh and METTL16-Sh samples (new Fig. 4E). In addition, we checked the primers sequences: F, 5'-CCCTGGCTTGTTCAACCTGT, and R, 5'-CCCAACCGTGATCTGTCTGTTA, in Fig. 4A. The PCR product using the primers is 127 bp, just targeted some sections of *FDX1* mRNA, and contained the key site 602 in the CDS region, which had been proved by many experiments as the important m⁶A modified site of METTL16 specific target. There might lead to about 60% decrease of m⁶A in Fig.4A.

7. Fig.4c, a description of *FDX1*-luciferase is needed in the main paper.

Response: Thank you for your vital suggestion. We are sorry for the inadequate description of *FDX1*-luciferase. To construct *FDX1*-luciferase, CDS and 3'UTR regions of *FDX1* were

constructed and fused into the downstream of pmirGLO Dual-luciferase vector (Promega, E1330), while FDX1-602-Mut was constructed by replacing adenosines (A) in m⁶A motif with thymine (T). For dual-luciferase reporter assay, cells seeded in 24-well plates were co-transfected with wild-type or mutant FDX1-luc and METTL16. After 48h post transfection, the activities of firefly luciferase and Renilla luciferase in each well were determined by a Dual-Luciferase Reporter Assay System (Promega, E1910) according to the manufacturer's protocol. In addition, we have added the schematic into Fig. 4G to make it intuitional.

8. Fig.5, are the levels of lactylated METTL16 and m6A enriched in gastric tumors compared to normal tissue?

Response: Thank you for your suggestion. We would detect the level of METTL16-K229-lac in gastric cancers, but it is limited by the quality of current METTL16-K229-lac antibody. A clear single strip would appear only after undergoing Flag-METTL16 immunoprecipitation, while jumbly strips emerge without taking immunoprecipitation. It is depressing that there were still jumbly strips using METTL16-K229-lac antibody in WB assay, even though we purified it. Considering the facts mentioned above, we cannot detect the level of METTL16-K229-lac in GC tissues at the moment. Thus, we can't utilize the antibody to make correlation analysis between METTL16-K229-lac and the copper concentration in GC tissues. We will try to optimize and purify the anti-K229-lac antibody, and establish the correlation in GC tissues in future study.

9. Fig.5f, does Cu²⁺ treatment result in intracellular lactate accumulation? How?

Response: Thank you for your comments. Theoretically, the higher the lactate content, the more lactic acid is available, increasing the lactylation of proteins. Thus, we examined the intracellular lactate in copper-treated GC cells using the lactic acid content detection kit (Solarbio, Beijing, China) and found no significant change between different groups with different copper concentrations (Supplementary Fig. 6A).

Then, we investigated whether copper mediated the regulation of lactoyltransferases on METTL16. Thus, the LC-MS/MS assay to identify the binding proteins of METTL16 was performed (Supplementary table 2). However, there is no binding lactoyltransferases, and it may result from the low abundance of proteins involved or insufficient bonding strength. Therefore, ten potential lactoyltransferases were selected, which has been reported or predicted, and were knockdown with specific siRNAs. These are as followed: are KAT3B (EP300), CREBBP (CBP), ACSS2, AARS1, AARS2, KAT6A (MOZ), KAT8 (MOF), KAT5, NMT1, and NMT2. The results showed that just AARS1-si and AARS2-si decreased METTL16-K229 lactylation under copper treatment (Fig. 6A). Furthermore, copper promotes the interaction of AARS1 / AARS2 to METTL16 significantly (Fig. 6B). Overexpression of AARS1 and AARS2

promote the lactylation of METTL16-K229 after copper treatment indeedly, indicating that AARS1 and AARS2 are both the lactoyltransferases of METTL16 (Fig. 6C). Overall, we confirmed that copper promoting METTL16 lactylation is due to the increased interaction between AARS1 and AARS2 with METTL16.

10. Fig.6-7, Sirt2 inhibitor has been reported to have anti-tumor effects, which may not be related to its de-lactylation activity. Alternatively, can LDH inhibitor compromise the effect of METTL16-mediate Cuproptosis? Or can inhibitors of oxidative phosphorylation sensitize the cells to METTL16-mediate Cuproptosis?

Response: Thank you for your suggestion. As shown in Supplementary Fig. 6B, LDH inhibitor compromised the effect of METTL16-mediated cuproptosis. The decrease of lactate concentration might inhibit METTL16 lactylation, ultimately inhibited cuproptosis. It is reported that the inhibitor of oxidative phosphorylation NAC didn't affect cuproptosis (A slight inhibition at high concentration of NAC) [12]. The possible reasons are: 1. Partly increase of lactate induced by the inhibition of oxidative phosphorylation just has a little influence on METTL16 lactylation in tumor tissues, which contain high concentration of lactate; 2. Cuproptosis is dependent on mitochondria respiration, which is one of the important processes of oxidative phosphorylation [12]. When oxidative phosphorylation was inhibited, cuproptosis was impossible to be activated, even though METTL16 was lactylation.

** See Nature Portfolio's author and referees' website at www.nature.com/authors for information about policies, services and author benefits.

Print Email

tracking system home | privacy policy | cookie policy | manage cookies

Reference:

1. Ge, L., et al., *Level of N6-Methyladenosine in Peripheral Blood RNA: A Novel Predictive Biomarker for Gastric Cancer*. Clin Chem, 2020. **66**(2): p. 342-351.
2. Doxtader, K.A., et al., *Structural Basis for Regulation of METTL16, an S-Adenosylmethionine Homeostasis Factor*. Mol Cell, 2018. **71**(6): p. 1001-1011 e4.
3. Liberti, M.V. and J.W. Locasale, *The Warburg Effect: How Does it Benefit Cancer Cells?* Trends Biochem Sci, 2016. **41**(3): p. 211-218.
4. Noe, J.T., et al., *Lactate supports a metabolic-epigenetic link in macrophage polarization*. Sci Adv, 2021. **7**(46): p. eabi8602.
5. Ashton, T.M., et al., *Oxidative Phosphorylation as an Emerging Target in Cancer Therapy*. Clin Cancer Res, 2018. **24**(11): p. 2482-2490.

6. Zhang, G., et al., *Variable copy number of mitochondrial DNA (mtDNA) predicts worse prognosis in advanced gastric cancer patients*. *Diagn Pathol*, 2013. **8**: p. 173.
7. Li, Y., et al., *Author Correction: Proteomic characterization of gastric cancer response to chemotherapy and targeted therapy reveals potential therapeutic strategies*. *Nat Commun*, 2022. **13**(1): p. 6749.
8. Su, R., et al., *METTL16 exerts an m(6)A-independent function to facilitate translation and tumorigenesis*. *Nat Cell Biol*, 2022. **24**(2): p. 205-216.
9. Han, L., et al., *METTL16 drives leukemogenesis and leukemia stem cell self-renewal by reprogramming BCAA metabolism*. *Cell Stem Cell*, 2023. **30**(1): p. 52-68 e13.
10. Ye, F., J. Wu, and F. Zhang, *METTL16 epigenetically enhances GPX4 expression via m6A modification to promote breast cancer progression by inhibiting ferroptosis*. *Biochem Biophys Res Commun*, 2023. **638**: p. 1-6.
11. Xiong, J., et al., *Lactylation-driven METTL3-mediated RNA m(6)A modification promotes immunosuppression of tumor-infiltrating myeloid cells*. *Mol Cell*, 2022. **82**(9): p. 1660-1677 e10.
12. Tsvetkov, P., et al., *Copper induces cell death by targeting lipoylated TCA cycle proteins*. *Science*, 2022. **375**(6586): p. 1254-1261.

REVIEWER COMMENTS

Reviewer #1 (Remarks to the Author):

I have found the authors addressed all of my comments in an adequate manner and I find the manuscript in its current form suitable for publication in Nature Communications.

I have some minor comments that I think would benefit the authors and the readers (but there are recommendations and not requirements)

1. One caveat of the work is that copper was measured using an enzymatic indirect assay. Ideally one would explore this using ICP-MS. However, at this point I think it would be unreasonable to ask for these experiments to be done, but I do think that it should be at least mentioned as a limitation. Also, from my understanding of the assay there is no way to distinguish between Cu²⁺ and Cu⁺, so throughout the manuscript, in the copper measurements it should be labeled as total Cu and not Cu²⁺ as it is currently labeled (if I understood correctly).

2. Do not think the drug names (i.e. elesclomol and disulfiram) should be capitalized.

3. The paragraph explaining Fig 4E and 4D would benefit some restructuring. First, 4E is discussed before 4D. Second, what is the luciferase reporter used in Fig. 4D? it just mentions FDX1 luciferase activity. Third, Figure 4E which is an important panel in explaining the logic of FDX1 mRNA modification would benefit from better visual representation and figure legend description. It is not intuitive to understand what the red and green peaks? What is the y-axis scale? Why would it suggest that these are the regions to be modified as mentioned in the text? Etc., this should be textually clarified in main text, figure legend and possibly on the figure panel itself. Also, would benefit elaborating more on the 602 mutation as it is in the CDS region, where does it fall in the FDX1 protein etc.,.

Reviewer #2 (Remarks to the Author):

The revised version of the manuscript by Sun et al. has been improved and addresses my former comments. I also feel the updated structure makes the manuscript easier to follow. I have no additional comments.

Reviewer #3 (Remarks to the Author):

The authors didn't address all the major concerns and most of minor questions appropriately and completely, for instance:

1, The major concern 1 "In Figure 5A-B and 2A-B, copper did not change the expression of METTL16, but increased the m6A level in gastric cancer cells. But there was an increased expression of METTL16 and higher copper content in gastric cancers. How to explain this discrepancy?"

2, Major concern 2, No information on SIRT2 in control samples.

3, Major concern 3, Unclearly explain on the effects of acetylation sites on methyltransferase activity of METTL16.

4, Minor questions 3-5, the responses were not convinced. Combined with m6A motif results, the m6A-seq analysis was doubtful.

5, The revised contents were not highlighted in the new manuscript and figures. 6, The text in page 8, paragraph 1 didn't match with the Fig. 3D and Fig. S3B.

Reviewer #4 (Remarks to the Author):

The authors have addressed most of the previous concerns by presenting additional experiments. However, the mechanism by which copper induces METTL16 lactylation remains unclear. The authors still need to provide references to support the assertion that AARS1/AARS2 are lactoyltransferases and show experimental evidence of AARS directly transferring lactylation to METTL16 in vitro. Furthermore, an explanation of copper's role in enhancing the interaction between AARS and METTL16 would strengthen the authors' findings.

April 27, 2023

Re: Revision to *Nature communications*

Dear editor and reviewers:

It's our great pleasure to submit our revised manuscript entitled "Lactylation of METTL16 promotes cuproptosis via m⁶A-modification on *FDXI* mRNA in gastric cancer" (NCOMMS-22-41195B) for publication in *Nature Communications*. Firstly, we greatly appreciate the thoughtful comments and highly professional constructive suggestions from the reviewers, which helped us to improve the quality of this manuscript. Based on the comments, we have incorporated changes into the manuscripts, as detailed below. Please also see below for a point-by-point response to the reviews' comments. The revised content was highlighted. We look forward to working with you and the reviewers to move this manuscript closer to the publication in *Nature Communications*.

Arrangement of Figures

Original	In Revision
Fig. 4D	Fig. 4E (relocated)
Fig. 4E	Fig. 4D (relocated)
Fig. 6A	Supplementary Fig. 7A (relocated)
Fig. 6B	Supplementary Fig. 7B (relocated)
Fig. 6C	Supplementary Fig. 7C(relocated)
Fig. 6D-O	Fig. 6A-L (relocated)
Supplementary Fig. S6C	Supplementary Fig. 6I (relocated)
Supplementary Fig. S6D	Supplementary Fig. 6B (relocated)
	Supplementary Fig. 6C (new)
	Supplementary Fig. 6D (new)
	Supplementary Fig. 7D(new)
	Supplementary Fig. 7E (new)
	Supplementary Fig. 7F (new)

REVIEWER COMMENTS

Responses to critiques (We used blue font to answer the questions):

Reviewer #1 (Remarks to the Author):

I have found the authors addressed all of my comments in an adequate manner and I find the manuscript in its current form suitable for publication in Nature Communications.

I have some minor comments that I think would benefit the authors and the readers (but there are recommendations and not requirements)

1. One caveat of the work is that copper was measured using an enzymatic indirect assay. Ideally one would explore this using ICP-MS. However, at this point I think it would be unreasonable to ask for these experiments to be done, but I do think that it should be at least mentioned as a limitation. Also, from my understanding of the assay there is no way to distinguish between Cu^{2+} and Cu^+ , so throughout the manuscript, in the copper measurements it should be labeled as total Cu and not Cu^{2+} as it is currently labeled (if I understood correctly).

Response: Thank you very much for your constructive suggestion. Indeed, the direct measurement of copper ions should be done using the ICP-MS method rather than the enzymatic indirect method. Due to our limited experimental conditions, we are not capable of performing ICP-MS on all samples at this stage. The principle of the copper microplate assay kit (abs580140) is that Cu^{2+} dissociated from ceruloplasmin and albumin can be reduced to Cu^+ by ascorbic, then Cu^+ reacts with complexing agent 3,5-dibromo-PAESA to produce blue complex, whose absorbance can be detected at 600nm wavelength. In fact, the copper measured by the enzymatic indirect method cannot distinguish between Cu^{2+} and Cu^+ present in the sample itself. Therefore, this measurement is for the total copper ion. We have re-labeled all Cu^{2+} in the entire manuscript as copper ions (Figure.1 and Supplementary Fig. 1). We will perform accurate analysis of copper ions using ICP-MS when the experimental condition is available.

2. Do not think the drug names (i.e elesclomol and disulfiram) should be capitalized.

Response: Thank you for your suggestion. We have made changes to the capitalization of drug names in the manuscript.

3. The paragraph explaining Fig 4E and 4D would benefit some restructuring. First, 4E is discussed before 4D. Second, what is the luciferase reporter used in Fig. 4D? it just mentions

FDX1 luciferase activity. Third, Figure 4E which is an important panel in explaining the logic of FDX1 mRNA modification would benefit from better visual representation and figure legend description. It is not intuitive to understand what the red and green peaks? What is the y-axis scale? Why would it suggest that these are the regions to be modified as mentioned in the text? Etc., this should be textually clarified in main text, figure legend and possibly on the figure panel itself.

Also, would benefit elaborating more on the 602 mutation as it is in the CDS region, where does it fall in the FDX1 protein etc.,.

Response: Thank you very much for your constructive suggestion.

1st, we are deeply sorry for attaching Fig.4D and Fig.4E in reverse order, and correct it in the revised manuscript.

2nd, the luciferase reporter gene mentioned in Figure 4D is FDX-WT-luc, which corresponds to the FDX1-WT-luc shown in Figure 4G. It is part of the CDS (Selected region:110456920-110457047, and 110462353-110462468) as well as the 3'UTR regions (Selected region: 110462469-110464884) of FDX1 were cloned into pGL3-control vectors (Genewiz, USA) which was comprised of firefly luciferase (F-luc). We apologize for not providing a detailed description in the original manuscript, and have added this information to the revised version.

3th, the red peak represents the sequencing m⁶A abundance of IP sample of control group, and the green peak represents the sequencing m⁶A abundance of IP sample of METTL16-Sh group. In the third and fourth peaks, the red peak is significantly higher than the green peak, suggesting that the differentially peak regions contain m⁶A modification sites mediated by METTL16. In order to visualize the m⁶A modifications present in the two groups of samples more intuitively, we performed IGV analysis on the IP and input of each group. The results are shown below, where the y-axis represents the normalized reads coverage (0-239). And we have made supplements and modifications in both the figure and the legends. We have revised the manuscript accordingly.

Fig. Coverage plots of m⁶A peaks in the *FDX1* genes in METTL16-Sh cells related to the control cells.

4th, the 602 position in the *FDX1*-CDS region corresponds to position 143 of the *FDX1* amino acid sequence, which located within the domain of 2Fe-2S ferredoxin-type and is crucial for the function of *FDX1*.

Reviewer #2 (Remarks to the Author):

The revised version of the manuscript by Sun et al. has been improved and addresses my former comments. I also feel the updated structure makes the manuscript easier to follow. I have no additional comments.

Response: Thank you for your recognition of our article.

Reviewer #3 (Remarks to the Author):

The authors didn't address all the major concerns and most of minor questions appropriately and completely, for instance:

1, The major concern 1 "In Figure 5A-B and 2A-B, copper did not change the expression of METTL16, but increased the m⁶A level in gastric cancer cells. But there was an increased expression of METTL16 and higher copper content in gastric cancers. How to explain this discrepancy?"

Response: Thank you very much for your constructive comments. We apologize for not explaining this issue clearly in our previous discussion. As shown in Figure. 5A-B and 2A-B, copper did not change the METTL16 protein level, but increased the m⁶A level, which may have led to the positive correlation between copper content and total m⁶A levels in GC tissues (Fig. 2A). The expression of METTL16 and copper content are both upregulated in gastric cancer comparing with adjacent non-tumor tissue. However, the correlation between the METTL16

protein level and copper content could not be established. Moreover, there are many inconsistencies in different gastric cancers: (I) copper content is associated with tumor malignancy (Fig. 1E-F), but the expression of METTL16 is not related to the degree of tumor malignancy (Supplementary Fig. 2G); (II) copper content is positively correlated with gastric cancer metastasis and malignant gastric cancer (Fig1. H-J), but the expression of METTL16 is not related to gastric cancer metastasis or malignant gastric cancer (Supplementary Fig. 2H-J); (III) Higher copper content is associated with poorer patient prognosis (Fig. 1C-D), but the expression of METTL16 is negatively correlated with gastric cancer prognosis (Supplementary Fig. 2K-L).

The reason for the increased expression of METTL16 in gastric cancer tissues compared to adjacent non-tumor tissue remains to be elucidated in our future work, which will be a very meaningful insight for using METTL16 as a therapeutic target. Again, we gratefully acknowledge your insightful guidance.

2, Major concern 2, No information on SIRT2 in control samples.

Response: Thank you for your question. In the last revision, we apologize for not including corresponding fluorescence images of SIRT2 in adjacent normal tissues, which have been added in the figure below. It was found that, at the overall level of gastric cancer, the expression of SIRT2 was lower in cancer tissues than that in adjacent normal tissues, but for some poorly differentiated gastric cancers, SIRT2 expression was upregulated in cancer tissues, and the application of AGK2 is suitable for these GCs.

Fig. SIRT2 expression in GC tissues. Immunofluorescent staining showing SIRT2 expression in paired adjacent normal tissues and well differentiated, moderately differentiated, or poorly differentiated tumor tissues in the GC TMA images. The red frames showed the amplified images of typical staining regions (right, 30x microscopy) from the global staining regions (left, 7x microscopy). Scale bar, 50 μ m.

3, Major concern 3, Unclearly explain on the effects of acetylation sites on methyltransferase activity of METTL16.

Response: Thank you very much for your thoughtful suggestions. We have demonstrated that the acetylation site mutations of METTL16 have no effect on the FDX1 expression (Supplementary Fig.6E-F). To further demonstrate whether the acetylation site of METTL16 affects its methyltransferase activity, we generated stable cell lines with METTL16 knockdown and rescue of different acetylation site mutations (-WT/ -K171R/ -K171Q/ -K348R/ -K348Q) and treated these cells with copper ions. Dot blot experiments were then performed to detect the m⁶A level in these cells. The results showed that the above mutations did not affect the m⁶A level in the cells (new Supplementary Fig. 6C and 6D). This indicates that the acetylation site of METTL16 does not affect its methyltransferase activity.

4, Minor questions 3-5, the responses were not convinced. Combined with m⁶A motif results, the m⁶A-seq analysis was doubtful.

Response: Above all, we sincerely appreciate your giving us another chance to explain your doubts. As shown in below, I will illustrate them accordingly in sequence:

1. Regarding to the minor question 3 that why the motif we identified is not the classical m⁶A motif. We apologize for placing the wrong motif in our initial manuscript. We mistakenly used all mRNA sequences (whether m⁶A modification occurs or not) for motif analysis (the left in the below figure). We then conducted motif analysis on mRNAs with m⁶A modification (the right in the below figure). To display the identified motifs clearly, we listed all possible motifs in decreasing order of *p*-values in the following figure. As shown, the top hit in both the Control and METTL16-Sh groups is the classic motif- GGAC. The sequencing results showed that the m⁶A level of FDX1 decreased in METTL16 knockdown cells, and the potential m⁶A sites located in the CDS and 3'UTR regions, with the motif region being the classical GGAC motif (see supplementary table-1 and uploaded GEO database (accession code: GSE224890)). We apologize for carelessly listing the motif of the left-hand side image in old Figure 3B of the first version of manuscript.

In addition, we added new sequencing data from another two repeated samples in the previous revision. The motif with the highest *p*-value in this result was also the classical GGAC motif (Fig. 3B). We also found that the m⁶A level of FDX1 decreased in METTL16 knockdown cells, and the potential m⁶A sites were in the CDS and 3'UTR regions (see supplementary table-1 and uploaded GEO database (accession code: GSE224890)). Therefore, we corrected the motif in the revised manuscript. We apologize again for our negligence and carelessness.

2. In minor question 4, you asked about the GO analysis of genes with decreased m⁶A in METTL16-KD cells and the reason why the number of genes in the GO analysis was too few to be significant. In our previous manuscript's Fig. 3C, the genes in the GO analysis were those with decreased m⁶A and mRNA levels, which means the genes were related to copper death and the TCA cycle shown in the chart of Fig. 3D. The number of genes was indeed too few. From the results showed in the Fig. 1 and Fig. 2, we demonstrated that METTL16 may be involved in the regulation of copper death. To directly prove the mechanism by which METTL16 regulates copper death, we performed METTL16-mediated m⁶A sequencing and found that there were copper death and TCA-related genes among the genes which are not only with m⁶A caused by METTL16 knockdown, but also the decreased mRNA levels (old Figure. 3D). We performed GO analysis to validate the pathway enrichment of these genes (old Fig. 3C), of which its quantity was not sufficient. After much deliberation, we admitted the layout of our previous statement might lead to ambiguity of the sequence of the Figures. To prove our conclusion, we performed sequencing on another two repeated samples. We performed GO analysis on all genes with decreased m⁶A and mRNA levels caused by METTL16 knockdown (Fig. 3C) and found copper death and TCA-related genes among them (Fig. 3D). These results

indicate that METTL16 directly participates in the regulation of m⁶A modification of copper death-related genes and leads to changes in mRNA.

3. In minor question 5, you asked why there were many peaks in the FDX1 intron region as shown in the old Fig. 3E IGV image. We reanalyzed the first sequencing results and after adjusting the data range, we observed that the peaks in the intron region were largely eliminated (as shown in the figure below). Of course, there were still a few peaks present in the intron region, which may indicate the presence of m⁶A modifications in pre-mRNA, consistent with previous reports [1-3].

However, as shown in the figure below, the m⁶A level of FDX1 decreased in METTL16-Sh cells, including positions 602 and 820. Furthermore, the m⁶A peak intensity in the intron region is relatively lower than other regions (5'UTR, 3'UTR, and CDS regions), so we did not focus on m⁶A modifications in the intron region, which should be further investigated in future work. To verify our conclusion, we performed sequencing on another two repeated samples. Although there were some differences in the IGV images of FDX1 due to different batches and IP technology from different companies, the conclusion remained the same: the m⁶A level of FDX1 decreased in METTL16-Sh cells, and potential m⁶A sites were found in the CDS and 3'UTR regions, including positions 602 and 820.

In summary, both sequencing results confirmed that METTL16 directly regulates the m⁶A level and mRNA level of the copper-death-related gene FDX1, and that potential m⁶A sites of FDX1 located in the CDS and 3'UTR regions. Combining the prediction results, we conclude that the potential m⁶A sites of FDX1 are A602 and A820. We apologize for not displaying the results clearly which result in confusion in our previous revision. Once again, sincere thanks for your professional guidance and help in improving our manuscript, which has enabled us to gain a deeper understanding of the field.

5, The revised contents were not highlighted in the new manuscript and figures.

Response: Thank you for your suggestion. We have highlighted the revised content as requested, firstly and secondly.

6, The text in page 8, paragraph 1 didn't match with the Fig. 3D and Fig. S3B.

Response: Thank you for your thorough review and for correcting our mistakes. We apologize for the error in the description of Fig. 3D. We have corrected it to "Furthermore, we identified 10 typical factors from the cuproptosis-related pathway that were related to low m⁶A methylation levels as well as downregulated mRNA levels in METTL16-Sh group (Fig. 3D)." (lines 10-12, page 8). In addition, we have revised the description of Fig. S3B to show that the m⁶A peaks were mainly enriched in CDS and 3'UTR regions in both METTL16-Ctrl and METTL16-Sh cells (lines 5-7, page 8).

Reviewer #4 (Remarks to the Author):

The authors have addressed most of the previous concerns by presenting additional experiments. However, the mechanism by which copper induces METTL16 lactylation remains unclear. The authors still need to provide references to support the assertion that AARS1/AARS2 are lactoyltransferases and show experimental evidence of AARS directly transferring lactylation to METTL16 in vitro. Furthermore, an explanation of copper's role in enhancing the interaction between AARS and METTL16 would strengthen the authors' findings.

Response: Thank you for your constructive comments and suggestions. The confirmed lactyltransferase enzymes is rare, so to expand the search scope, we have also included all other possible proteins. Our results showed that copper treatment enhances the binding of METTL16 to AARS1/AARS2 and promotes lactylation modification (Supplementary Fig.7B-C). Although there is no direct evidence indicating that AARS1/2 is a lactyltransferase, we compared the lactylation modification group with other known aminoacylation modification groups and found that the lactylation modification group has an extremely similar modification structure to the modification structure of the alanine modification group (below data) [4-6]. Therefore, we speculate that AARS1 and AARS2, which have been shown to promote alanine modification, may also have a role in promoting lactylation modification.

Fig. Comparison between acetyl modification group and propionyl modification group.

Our study found that knocking down AARS1/2 significantly reduces the lactylation level of METTL16 (Supplementary Fig. 7A). However, we did not investigate whether AARS1/AARS2 is a direct lactyltransferase of METTL16. Therefore, we conducted additional experiments as follow:

1st, we used western blot to detect lactylation in AARS1/2-OE cell lysate and found that overexpression of AARS1/2 upregulates the overall lactylation level (below figure).

Fig. Overexpression of AARS1 or AARS2 upregulates the overall lactylation level. HCG27 cells were overexpressed with AARS1 or AARS2 for 36 h. Whole cell lysate was blotted with pan-Kla antibody by western blotting.

2nd, METTL16/AARS1/2 was overexpressed and purified in 293T cells separately, then conducted *in vitro* lactylation assay, and the results showed that AARS1/2 can promote lactylation of METTL16 dramatically (Supplementary Fig. 7D).

3th, we purified METTL16-WT and METTL16-K229R from *E. coli* separately and incubated with AARS1/2 purified from 293T cells and conducted *in vitro* lactylation experiments, the results showed that AARS1/2 significantly promote lactylation of METTL16-WT other than METTL16-K229R *in vitro* (Supplementary Fig. 7E).

4th, to further investigate whether AARS1/2 directly regulates the lactylation of METTL16, we synthesized an unmodified peptide around K229 of METTL16 (ELEFVK(229)RIIHD) and incubated it with purified AARS1/2 from 293T. LC-MC detection revealed that METTL16

peptide was lactylated at K (229) by AARS1 or AARS2 (Supplementary Fig. 7F). The above results suggest that AARS1/2 directly regulate the lactylation modification of METTL16.

To explain the reason why copper ions enhance the binding of METTL16 to AARS1/2 and enhance lactylation, we analyzed the protein structure of AARS1/2 and found that both have metal sulfur cluster binding sites in their amino acid sequences. Therefore, we speculate that when cells are exposed to copper ions, copper ions may bind to the metal sulfur cluster binding region of AARS1/2 protein, causing a conformational change in AARS1/2 protein, enhancing its binding to METTL16, and thereby inhibiting the binding of SIRT2 to METTL16, leading to the enhanced lactylation of METTL16. We have also added the above content to our discussion and will continue to explore it in future work.

Thank you very much for your constructive and professional suggestion again.

References

1. Timcheva, K., et al., *Chromatin-associated YTHDC1 coordinates heat-induced reprogramming of gene expression*. Cell Rep, 2022. **41**(11): p. 111784.
2. Wei, G., et al., *Acute depletion of METTL3 implicates N (6)-methyladenosine in alternative intron/exon inclusion in the nascent transcriptome*. Genome Res, 2021. **31**(8): p. 1395-1408.
3. Li, W., et al., *Mapping the m1A, m5C, m6A and m7G methylation atlas in zebrafish brain under hypoxic conditions by MeRIP-seq*. BMC Genomics, 2022. **23**(1): p. 105.
4. Kwon, N.H., P.L. Fox, and S. Kim, *Aminoacyl-tRNA synthetases as therapeutic targets*. Nat Rev Drug Discov, 2019. **18**(8): p. 629-650.
5. Khan, K., V. Gogonea, and P.L. Fox, *Aminoacyl-tRNA synthetases of the multi-tRNA synthetase complex and their role in tumorigenesis*. Transl Oncol, 2022. **19**: p. 101392.
6. Almalki, A., et al., *Mutation of the human mitochondrial phenylalanine-tRNA synthetase causes infantile-onset epilepsy and cytochrome c oxidase deficiency*. Biochim Biophys Acta, 2014. **1842**(1): p. 56-64.

REVIEWER COMMENTS

Reviewer #3 (Remarks to the Author):

The authors have addressed most of the concerns, I have no further comments for the work.

Reviewer #4 (Remarks to the Author):

The authors suggest that AARS1/2 could potentially catalyze the lactylation reaction directly, considering the structural similarities between L-lactate and alanine, a natural substrate of AARS1/2. However, the choice to use Lac-CoA instead of lactate+ATP for their in vitro lactyl-transferase assay raises concerns regarding the similarity to the standard aminoacylation reaction procedure.

Fig. S7D, it is highly probable that the observed lactyl-transferase activity is attributed to other enzymes co-purified with AARS1/2 during the flag-affinity purification process in HEK293T cells. The authors have not provided evidence regarding the purity of AARS1/2, which raises concerns about potential contamination. A preferable approach would be to use recombinant proteins.

Fig. S7F, it is not clear by their description that "we synthesized an unmodified peptide around K229 of METTL16 (ELEFVK(229)RIIHD) and incubated it with purified AARS1/2 from 293T". Additional clarification is needed to understand how the experiment was conducted.

July 24, 2023

Re: Revision to *Nature communications*

Dear reviewers:

We greatly appreciate the helpful comments and highly professional suggestions from the reviewers, which have significantly improved the quality of this manuscript, as well as in guiding our research in the future. In response to the comments, we have revised our manuscripts accordingly, as detailed below. Please refer to the point-by-point response to the reviewers' comments below. The revised content was highlighted.

Thank you again for your valuable comments and suggestions. We look forward to hearing from you soon.

REVIEWER COMMENTS

Responses to critiques (We used blue font to answer the questions):

Reviewer #3 (Remarks to the Author):

The authors have addressed most of the concerns, I have no further comments for the work.

Response: Thank you for your recognition and support of our work.

Reviewer #4 (Remarks to the Author):

We deeply acknowledge your carefulness and conscientiousness. Your insightful suggestions are so professional and valuable for revising and improving our paper. According to your suggestions, we have made the following revisions on this manuscript.

The authors suggest that AARS1/2 could potentially catalyze the lactylation reaction directly, considering the structural similarities between L-lactate and alanine, a natural substrate of AARS1/2. However, the choice to use Lac-CoA instead of lactate+ATP for their in vitro lactyl-transferase assay raises concerns regarding the similarity to the standard aminoacylation reaction procedure.

Response: Thank you for your constructive comments.

As you known, the specific reaction processes of aminoacylation and lactylation are different

from each other. The reaction process of aminoacylation involved the initial binding of an amino acid to a specific aminoacyl-tRNA synthetase, and thus forming an aminoacyl-AMP intermediate. Then, this intermediate was bounded to the corresponding tRNA or protein, and, under the catalysis of aminoacyl-tRNA synthetase, was transformed into aminoacyl-tRNA or aminoacyl-lysine (He et al., 2018; Ibba & Soll, 2001; Kwon, Fox, & Kim, 2019).

The enzymatic dependent acylation is mediated by the acyltransferase (writer), which transfer the acyl groups from the “donors” of acyl-CoA to the side-chain of lysine residues (Shang, Liu et al. 2022). The lactylation is produced after a two-step process. First, lactic acid was converted to lactyl-CoA by the action of coenzyme A transferase. Next, acted as a donor, lactyl-CoA was transferred by lactylation enzymes (writer) to the target protein. Lactyltransferase, such as P300 or CBP, was responsible for transferring lactyl groups to target proteins. Therefore, lactyl coenzyme A (lactyl-CoA) was used as the donor of *in vitro* lactylation assay (Zhang, Tang et al. 2019).

Similarly, it was found that AARS1/2 may act as lactyltransferase (writer), which are responsible for catalyzing the transfer of lactyl groups to target proteins. Therefore, in our *in vitro* lactylation assay, we used lactyl-CoA as the donor instead of lactic acid.

Fig. S7D, it is highly probable that the observed lactyl-transferase activity is attributed to other enzymes co-purified with AARS1/2 during the flag-affinity purification process in HEK293T cells. The authors have not provided evidence regarding the purity of AARS1/2, which raises concerns about potential contamination. A preferable approach would be to use recombinant proteins.

Response: Thank you for your constructive review. We feel sorry that we didn't provide the evidence regarding the purity of AARS1/2 in our previous revision. To address this, we performed coomassie brilliant blue staining (Below data) and mass spectrometry analysis (Related Manuscript File- Table 1 and Table 2, at the end of the document) for the purified samples of AARS1/2 from 293T cells. This coomassie staining method allows us to visually assess the purity of AARS1/2. As shown in the figure below, besides the non-specific bands, the most prominent band is AARS1/2.

Fig. Coomassie brilliant blue staining on the purified samples of AARS/2 from 293T cells.

To rule out the possibility of the presence of other potential lactyltransferases in the samples, a direct mass spectrometry analysis was performed (Related Manuscript File- Table 1 and Table 2). The results indicated that there were no other known lactyltransferases present in the samples. The research on new lactyltransferases is ongoing, and there may be new lactyltransferases waiting to be discovered. How AARS1/2 function as lactoyltransferases and whether there are other factors involved in supporting enzymatic activities of AARS1/2 remain to be further explored. We have added the above content to our discussion section and will continue to explore it in future work.

In addition, we attempted to express and purify AARS1/2 from *E.coli*. Unfortunately, neither AARS1 nor AARS2 was successfully expressed from *E.coli*, possibly due to inclusion bodies formation or other unknown reasons. We are really apologized that recombinant AARS1/2 proteins are expressed failure and we will overcome this obstacle in our future work. Although we cannot exclude an indirect effect by AARS1/2 in METTL16 lactylation, together with the *in vitro* lactylation assay (Fig S7D-F), these data suggest that AARS1/2 are potential lactylation writers to promote lactylation of METTL16.

Thank you once again for your professional suggestions. With your help, we have deepened our understanding of this field and enhanced the quality of our work.

Fig. S7F, it is not clear by their description that “we synthesized an unmodified peptide around K229 of METTL16 (ELEFVK(229)RIIHD) and incubated it with purified AARS1/2 from 293T”. Additional clarification is needed to understand how the experiment was conducted.

Response: Thank you for your suggestion. We apologized for the unclear description, and we revised as below to clarify the experiment: To investigate the potential interaction between

METTL16 and AARS1/AARS2, a peptide of METTL16 (ELEFVK(229)RIIHD) was synthesized by GI Biochem (Shanghai) Co., Ltd. Flag-AARS1 or AARS2 was precipitated from Flag-AARS1 or AARS2 overexpressing 293T cells using FLAG-M2 beads and eluted with Flag peptide (100 µg/ml, Sigma-aldrich). The peptide of METTL16 incubated with AARS1, AARS2, or vector in reaction buffer (100 mM HEPES, 100 mM MgCl₂, 10 mM KCl, and 0.1 mM lactyl-CoA, PH = 7, Shanghai Nafu Biotechnology Co., Ltd) at 37 °C for 2h. The reaction products were performed to LC-MS detection. LC-MS/MS analysis revealed that the peptide of METTL16 remained unmodified when incubated with vector, while the lysine site (K229) was obviously modified by lactylation when incubated with AARS1 and AARS2. And we have added the above description for Fig.S7F to our manuscript.

Reference

He, X., et al. (2018). "Sensing and Transmitting Intracellular Amino Acid Signals through Reversible Lysine Aminoacylations." Cell Metabolism 27, 1–16.

Ibba, M. and D. Soll (2001). "The renaissance of aminoacyl-tRNA synthesis." EMBO Rep 2(5): 382-387.

Kwon, N. H., et al. (2019). "Aminoacyl-tRNA synthetases as therapeutic targets." Nat Rev Drug Discov 18(8): 629-650.

Shang, S., J. Liu and F. Hua (2022). "Protein acylation: mechanisms, biological functions and therapeutic targets." Signal Transduct Target Ther 7(1): 396.

Zhang, D., et al. (2019). "Metabolic regulation of gene expression by histone lactylation." Nature 574(7779): 575-580.

Related Manuscript File- Table 1

Fasta headers	GeneName	Number of proteins	Peptides	Razor + unique peptides	Unique peptides	Sequence coverage [%]	Mol. weight [kDa]	Score	Intensity	iBAQ
A0A6Q8	AARS1 AARS	18	48	48	48	64.8	108.29	323.31	1.27E+10	2E+08
A0A2R8	SIK1	3	1	1	1	12.4	17.802	0.11231	0	0
A0A2X0	PLEKHG3	4	1	1	1	1	134.41	0.10111	0	0
U3KPR6	C5orf63	6	1	1	1	20	6.2514	0.61983	0	0
A0A2R8	SMCHD1	3	1	1	1	5.2	35.015	0.00979	0	0
C9J302	C4orf51	1	1	1	1	4.5	23.001	0.10214	0	0
F8VZJ2	NACA	10	1	1	1	10.3	15.016	0.08895	0	0
J3QRY4	PSMD11	3	1	1	1	6.4	21.109	0.35199	0	0
A0A1W2	EIF3F	6	1	1	1	6.4	16.804	0.86874	4616300	2E+06
O14513	NCKAP5 ERIF	1	1	1	1	0.5	208.53	0.04685	0	0
O14744	PRMT5 HRM	15	29	29	29	62.5	72.683	142.31	5.78E+09	2E+08
A0A2R8	TPP1	11	1	1	1	14.1	10.535	0.29372	5515000	1E+06
A0A024	hCG_203941	2	1	1	1	1.3	301.64	0.02943	82839000	579290
O15050	TRANK1 KIAA	2	1	1	1	0.4	336.22	-2	0	0
G5E9S7	ARPC2 hCG_	5	1	1	1	23.7	4.1457	0.29288	2084700	521170
A0A8V8	ACTN4	59	6	6	6	9.3	99.811	17.244	54053000	1E+06
O60287	URB1 C21orf	1	1	1	1	1.1	254.39	0.14442	0	0
O60361	NME2P1	8	1	1	1	12.4	15.529	0.01368	3172900	317290
K7EL96	PLIN3	4	1	1	1	8.7	17.962	0.80464	15979000	1E+06
A0A8V8	WDR1	13	1	1	1	4.2	36.35	1.4802	6640700	510830
H0YH95	PPFIA2	7	1	1	1	4.2	32.763	1.377	0	0
A0A804	FLNB	6	9	7	7	3.4	274.02	3.0604	101400000	729520
H0YIC4	CS	7	1	1	1	8.9	14.123	0.95124	10003000	1E+06
Q7Z497	SF3B1	2	1	1	1	2	87.408	0.37994	2347900	61787
C9JIR6	PPM1B	15	8	8	8	31.5	41.83	37.521	197410000	1E+07
B2R644	DDAH1 HEL-	2	1	1	1	3.9	31.121	0.03376	2345700	137980
J3KSF8	EPN2	14	1	1	1	50	3.6841	0.00785	0	0
O95365	ZBTB7A FBI1	1	1	1	1	2.9	61.438	0.08101	220600000	1E+07
O43698	ZFS-6	18	1	1	1	8.1	11.514	0.1252	0	0
F5GZQ4	LDHA	8	1	1	1	20.8	7.8752	4.3527	56266000	1E+07
V9HWF4	HEL-S-68p	6	7	7	7	24	44.614	36.926	167520000	6E+06
P01859	IGHG2	6	1	1	1	2.8	35.9	0.39834	17321000	1E+06
Q5I6Y5	LMNA	16	8	8	8	18.8	55.043	3.4445	117210000	4E+06
P03952	KLKB1 KLK3	3	1	1	1	1.4	71.369	0.10712	0	0
V9HWN	HEL-S-87p A	13	6	6	6	18.7	39.42	2.2108	74771000	3E+06
P04083	ANXA1 ANX1	26	2	2	2	11.6	38.714	0.11863	13500000	675000
Q0QET7	GAPDH	8	5	5	5	30.4	24.62	2.1758	54936000	5E+06
P05141	SLC25A5 AAC	8	2	2	1	6.4	32.852	1.2051	21206000	1E+06
F8VQY6	RPLP0	15	1	1	1	15.5	15.813	0.20391	12091000	1E+06
E9PGX9	SSB	5	1	1	1	11.7	13.909	0.01452	4696100	782690
Q0QEN7	ATP5B	6	5	5	5	14.2	48.113	5.5551	66684000	3E+06
P06733	ENO1 ENO1L	22	12	12	10	39.6	47.168	28.21	574730000	3E+07
A0A7I2Y	NPM1	18	4	4	4	19.2	35.713	5.8798	98023000	8E+06
A0A7P0	P4HB	33	6	6	6	14.7	62.031	3.6643	63830000	2E+06
F8WD96	CTSD	13	1	1	1	4.7	30	4.3189	10167000	847250
H0YMW	ANXA2	28	9	9	9	31.1	41.899	314.62	317180000	1E+07
P07437	TUBB TUBB5	57	12	12	3	36.7	49.67	145.25	332150000	2E+07
Q8TBA7	HSP90AA1	16	5	1	1	9.4	73.826	0.23878	6720900	231760
P07910	HNRNPC HN	29	3	3	3	9.8	33.67	2.0189	24865000	1E+06
B4DMA2		7	8	8	3	15.7	79.194	4.0707	116120000	4E+06
A0A6Q8	MME	4	1	1	1	2.8	79.915	0.79777	0	0
V9HWE1	HEL113 VIM	43	21	21	16	44.6	53.651	20.945	1.491E+09	4E+07
D6RBE9	ANXA5	4	2	2	2	10.5	24.698	4.7672	37239000	2E+06
A0A024	LOC388524 F	11	2	2	2	12.6	22.014	1.2606	51191000	5E+06

A0A087>GSTP1	5	1	1	1	29	7.4493	0.01153	2627700	656920
P09651 HNRNPA1 HI	23	2	2	2	7.3	38.746	4.4206	8920300	495570
P0CG39 POTEJ	1	4	1	1	5.5	117.39	0.7601	6001900	115420
J3QSA3 UBB	39	1	1	1	37.2	4.8535	1.6419	62501000	3E+07
A1L407 HIST1H1T	10	1	1	1	5.3	22.033	0.52086	18661000	2E+06
P10809 HSPD1 HSP6	21	6	6	6	12.7	61.054	4.5308	93022000	3E+06
A0A7P0` HSPA5	5	4	3	3	8.3	69.01	2.3532	29816000	1E+06
P11142 HSPA8 HSC7	26	7	6	5	16.6	70.897	10.727	128610000	4E+06
A0A8Q3 G6PD	25	5	5	5	9.3	66.183	3.1459	41927000	1E+06
H0YB86 PABPC1	20	1	1	1	8.5	17.985	0.44977	6598900	599900
Q6I9V5 SLC25A6 hCG	2	2	1	1	6.4	32.866	0.0289	3321100	166050
A0A024FRNH1 hCG_1	10	3	3	3	9	49.416	3.2251	32821000	1E+06
H7C4N2 PLS3	6	1	1	1	6.3	21.69	3.8034	0	0
P13929 ENO3	6	3	1	1	12.9	46.986	0.05097	0	0
A0A8V8` PKM	29	14	14	14	29.8	65.805	52.558	511660000	1E+07
A0A7P0` HSP90B1	14	1	1	1	3	53.079	0.03916	5664600	217870
J3QLE5 SNRPN	11	1	1	1	4.7	17.546	3.6364	4586800	458680
Q53SB5 tmp_locus_2	10	6	1	1	9.1	53.535	0.18191	0	0
A0A0U1 DDX17	17	1	1	1	21.4	5.9768	1.3494	6293700	2E+06
A8MUD` RPL7	4	1	1	1	5.3	24.433	0.99027	11307000	1E+06
B3KM80 NCL hCG_33	15	9	9	9	22.4	58.554	51.666	214300000	8E+06
H0Y6D8 TBP	6	1	1	1	15.8	11.502	0.03841	0	0
P21333 FLNA FLN FL	18	30	30	28	17.3	280.74	41.072	531820000	4E+06
A0A7I2V HNRNPA2B1	13	2	2	2	15.8	18.051	2.2793	49084000	4E+06
A0A8C8 ITGA6	2	1	1	1	0.8	117.8	0.275	0	0
A0A024FWARS hCG_2	2	1	1	1	3.4	53.165	1.9998	0	0
E9PQ96 RPS3	11	1	1	1	14.3	10.279	0.47502	13729000	2E+06
P23526 AHCY SAHH	3	2	2	2	6	47.716	0.17762	18538000	926890
E9PS23 CFL1	5	1	1	1	13.3	10.08	0.70237	12103000	2E+06
Q8NHX` HCC5	6	1	1	1	1.8	75.786	1.3974	3218600	76634
V9HW2` HEL-S-123m	9	5	5	5	11.2	59.75	4.4368	56550000	2E+06
K7EK45 PTBP1	5	1	1	1	9	27.114	3.8369	15497000	2E+06
Q53YD7 EEF1G hCG_2	6	5	5	5	10.8	50.118	4.1558	66695000	3E+06
C9JXA2 EPHA3	5	1	1	1	1.7	103.14	0.27399	0	0
Q9BV00 LMOD1	6	1	1	1	3.7	29.962	0.25663	0	0
E9PN91 EEF1D	18	1	1	1	11.3	11.616	0.72677	20263000	3E+06
P30101 PDIA3 ERP57	19	4	4	4	9.5	56.782	9.7861	13132000	452830
B4DEX8 MAT2A hCG_	8	2	2	2	7.2	39.71	0.18197	9324200	582760
G8JLB6 HNRNPH1	31	4	4	4	11.7	51.229	95.006	127910000	7E+06
F5GXD8 STIP1	5	1	1	1	6.5	15.637	0.48691	6579100	657910
Q86XU5 MYH9	9	3	3	3	2.8	158.75	2.4003	14014000	189380
H3BU31 RPL4	8	2	2	2	15.8	18.966	3.3818	54929000	9E+06
F2Z393 TALDO1	3	1	1	1	3.5	35.328	1.6824	7685900	512390
A0A7I2V HSPA9	16	3	3	3	6.6	66.601	1.1776	21664000	637160
M0QZU: RPL13A	9	1	1	1	23.9	4.7176	1.48	896740	298910
B8ZZ51 MDH1	5	1	1	1	5.9	18.689	0.08647	6533200	816650
Q2VIR3 EIF2S3B	2	1	1	1	3.4	51.228	0.59798	2910600	138600
A8K2M0 PSMC4 hCG_	2	1	1	1	3.3	47.366	0.01751	2396400	95856
Q5FWG` IQGAP1	9	1	1	1	4.8	30.925	4.3938	0	0
A0A6I8P CAPZB	8	1	1	1	23.7	6.7564	0.05437	5382600	2E+06
A0A0U1 FASN	2	1	1	1	0.4	273.2	0.1738	0	0
B4DUR8 CCT3	6	4	4	4	9	55.674	31.999	33095000	1E+06
E9PQ34 SERPINH1	11	1	1	1	9.2	15.301	0.146	3241100	360120
Q7Z759 CCT8 hCG_1	5	3	3	3	7.2	54.106	1.3904	17693000	552900
A0A2R8` RPS6KA3	7	1	1	1	7.4	31.615	0.26961	0	0
K7ESJ2 CNN1	7	1	1	1	17.4	13.001	0.03065	0	0
M0ROY6 HNRNPM	8	1	1	1	11.1	10.739	0.56586	5030100	1E+06
J3QL07 KPNA2	10	1	1	1	24	5.5029	0.32895	2349200	783070
A0A024C MYOM2 hCG	2	1	1	1	1	164.91	0.03692	0	0

A0A087>FGF8 hCG_2E	5	1	1	1	10.7	16.254	0.21237	0	0
R4GN08 ARPC4	5	1	1	1	10.4	9.405	0.28742	3500800	875190
P60842 EIF4A1 DDX2	26	6	6	6	17.2	46.153	3.1731	69124000	3E+06
E5RIP1 RPS20	4	1	1	1	23.9	5.4292	0.20576	5580100	3E+06
A0A6Q8 PRPS1	7	1	1	1	10.9	10.257	0.11896	0	0
P61568 ERVK11-1	1	1	1	1	8.9	21.461	1.7095	0	0
B4DUQ1 HNRNPK	8	5	5	5	16.2	48.51	42.781	68749000	3E+06
M0QX7E RPS16	6	1	1	1	20	5.5575	0.94806	9085400	2E+06
P62258 YWHAE	40	4	4	2	19.2	29.174	3.8241	42882000	3E+06
J3JS69 RPS18 hCG_1	2	1	1	1	8.5	9.7672	0.15807	0	0
P62318 SNRPD3	1	2	2	2	15.1	13.916	1.4932	60398000	1E+07
Q96IR1 RPS4X	4	1	1	1	3.7	27.259	0.48349	18156000	1E+06
A2A3R7 RPS6	4	1	1	1	13.2	10.415	0.1445	0	0
H0YFC6 RAN	6	1	1	1	10.7	11.568	0.84549	0	0
Q5VVC8 RPL11	4	1	1	1	8.4	19.024	1.3221	6985900	776210
E9PP36 RPL8	4	1	1	1	7.4	16.18	0.4577	9939200	2E+06
D0PNI1 YWHAZ HEL-	9	3	1	1	13.1	27.745	2.7681	12209000	872100
J3KSP2 RPL38	5	1	1	1	61.9	2.559	0.09541	5651700	6E+06
D6RAU2 RACK1	14	1	1	1	7.5	16.403	0.26968	4013000	445890
P63261 ACTG1 ACTC	11	21	1	1	78.1	41.792	114	599330000	3E+07
P68032 ACTC1 ACTC	19	11	1	1	30	42.019	2.0669	408970000	2E+07
P68363 TUBA1B	42	9	9	9	29.9	50.151	246.39	672620000	3E+07
P68371 TUBB4B TUBI	23	11	2	2	33.3	49.83	1.8601	55263000	3E+06
F8VQ14 CCT2	6	6	6	6	17.8	44.812	5.8202	77045000	3E+06
P81605 DCD AIDD D	1	2	2	2	22.7	11.284	0.82937	33107000	7E+06
G3V279 ERH	3	1	1	1	15.5	8.2033	5.5308	9580400	2E+06
A0A0S2>RBM10	5	2	2	2	3.2	94.369	0.64529	12021000	343460
Q00839 HNRNPU C1C	17	2	2	2	4	90.583	1.5492	15196000	410700
A0A8I5K SET	8	3	3	3	14.6	26.779	1.2205	44903000	6E+06
E9PNN3 PTS	5	1	1	1	11.7	8.9132	0.04693	5149000	1E+06
A0A0A0I PRDX1	8	2	2	2	21.6	10.676	2.1169	32647000	4E+06
Q96AS4 SPIRE1	4	1	1	1	7.6	14.952	0.13851	0	0
Q8N437 RASGEF1B	4	1	1	1	6.2	24.13	0.01268	0	0
F8VRG3 TWF1	2	1	1	1	22	6.4352	0.16594	6074900	3E+06
B4DY09 ILF2	5	1	1	1	2.8	38.91	0.0861	9053000	532530
A0A1B0C SPTAN1	9	1	1	1	50	2.5288	0.60724	3691400	2E+06
D6RF44 HNRNPD	11	1	1	1	12.6	12.553	0.66664	14175000	2E+06
H0YBT4 DPYSL3	13	1	1	1	10.5	21.846	4.3997	0	0
H0YEV2 CTTN	8	1	1	1	10.9	14.311	0.07015	7471600	2E+06
Q14721 KCNB1	1	1	1	1	1.2	95.877	0.20968	0	0
C9IY94 SEPTIN2	10	1	1	1	9.9	13.737	0.34624	0	0
H3BUD4 PLCL1	4	1	1	1	1.8	114.69	0.39285	0	0
A0A024F STK38 hCG_1	9	7	7	7	24.7	54.19	23.991	264470000	1E+07
F8W0G4 PCBP2	16	2	2	2	15.2	16.637	3.1982	0	0
D6RH11 ELF2	5	1	1	1	4.2	21.998	0.2261	0	0
Q15750 TAB1 MAP3K	3	1	1	1	2.2	54.643	0.00922	4620800	184830
F2Z2I2 PFKFB3	11	4	4	4	9.5	52.584	0.81909	23991000	727000
B7WPD9 KIF26B	3	1	1	1	0.5	184.68	0.96439	0	0
H3BLT5 BLTP1	8	1	1	1	1	152.56	1.0768	134740000	2E+06
Q2M3C7 SPHKAP KIA/	1	1	1	1	0.6	186.45	5.9131	2722100	31289
Q58FF8 HSP90AB2P I	1	3	1	1	9.7	44.348	0.00933	5384000	269200
H0YH72 FAM133B	3	1	1	1	11.7	13.46	0.03003	0	0
A0A8Q3 ZC3H12B	3	1	1	1	3.8	41.699	0.29427	0	0
E9PHX9 TTC39A	7	1	1	1	10.7	12.67	0.05192	0	0
H0YBN5 VIRMA	2	1	1	1	1.9	81.979	0.40088	0	0
Q6PIU2 NCEH1 AAD/	2	1	1	1	2.9	45.807	0.12426	2318800	115940
Q6W3E5 GDPD4 GDE6	1	1	1	1	1.6	71.995	0.02698	0	0
Q6ZMY6 WDR88 PQW	1	1	1	1	4.2	52.62	0.06715	0	0
Q6ZR08 DNAH12 DH0	1	1	1	1	0.4	356.94	0.02015	0	0

D3DN77 hCG_2022730	6	2	2	2	7.8	30.768	1.8834	3774200	209680
A0A6Q8 PRSS41	3	1	1	1	5.2	34.107	0.01737	2516600	125830
G3XAL8 ZGRF1 hCG_2	2	1	1	1	1.5	104.99	0.05463	0	0
A0A804 KIAA0825	2	1	1	1	1.1	147.76	0.08444	0	0
Q05BG6 FAM178A	4	1	1	1	2.4	38.363	0.1279	0	0
A3KMG4 DOCK3	3	1	1	1	2.5	69.178	0.08677	0	0
A0A024 SYCP3 hCG_2	2	1	1	1	4.7	27.728	1.0914	0	0
Q8NA29 MFSD2A MF9	1	1	1	1	1.7	60.17	0.02511	0	0
A0A024 hCG_1811539	5	1	1	1	3.1	36.177	0.16546	1960400	108910
Q8NDH2 CCDC168 C1	1	1	1	1	0.2	801.9	0.00723	0	0
Q9NTA6 DKFZp761I21	8	1	1	1	3	69.294	0.09533	0	0
Q96AG4 LRRC59 PRO	1	1	1	1	2.9	34.93	0.30907	0	0
K7EQH1 ARK2N	3	1	1	1	6.2	15.512	0.09733	0	0
E5RIY4 MTERF3	3	1	1	1	11	20.098	0.01522	0	0
H0Y7V4 DNAH8	3	1	1	1	0.9	478.87	0.23574	0	0
Q96JF6 ZNF594 KIAA	3	1	1	1	3.5	93.906	0.02764	0	0
H0YHI9 MYRFL	3	1	1	1	9.3	42.441	0.0791	0	0
Q96N20 ZNF75A	2	1	1	1	5.4	34.693	0.14615	0	0
Q8ND87 DKFZp434K0	2	1	1	1	9.7	15.64	0.11553	0	0
Q96PG2 MS4A10 CD2	1	1	1	1	3.4	29.747	0.72492	91255000	7E+06
D3DR32 MPHOSPH1 I	2	1	1	1	0.7	206.16	0.00376	0	0
H0UIC2 hCG_1794678	3	1	1	1	1.2	121.98	0.01067	0	0
A0A087 MAGI1	8	1	1	1	0.9	106.9	0.0101	0	0
B3KW33 OSBPL9 hCG	5	2	2	2	4.4	58.497	0.69987	234170000	1E+07
Q53SW3 DPYSL5	2	1	1	1	4.6	52.093	0.136	0	0
A0A024 WDR77 hCG_	4	7	7	7	33.9	36.724	77.041	992340000	9E+07
Q9BRS2 RIOK1 RIO1	1	1	1	1	1.8	65.582	0.54977	15638000	601450
B4DKD1 TTYH2	3	1	1	1	1.6	56.32	0.01467	26252000	2E+06
Q9BXX2 ANKRD30B	3	1	1	1	1.1	158.05	0.01417	14891000	215820
B3KTY4 SLITRK2 hCG	3	1	1	1	1.1	81.553	0.02825	0	0
A0A024 ASCC2 hCG_	3	1	1	1	1.9	73.197	5.7868	0	0
Q9H444 CHMP4B C2C	1	1	1	1	6.7	24.95	0.14374	0	0
Q9H8M2 CNNM2 ACC	1	1	1	1	1.7	96.622	0.10513	0	0
Q9HB55 CYP3A43	1	1	1	1	2.4	57.669	0.0333	0	0
Q9HB75 PIDD1 LRDD	1	1	1	1	1.4	99.711	0.19992	5643500	148510
A0A0U1 RTN4	8	1	1	1	6.3	14.088	0.32972	9177000	1E+06
C9JTK6 OLA1	5	2	2	2	22.4	12.302	1.429	6680500	954350
Q6AI22 DKFZp686H1	2	1	1	1	0.4	318.87	1.2782	0	0
Q9NYR8 RDH8 PRRD	2	1	1	1	2.3	33.755	0.07222	0	0
Q50KP5 Hosa(Japanes	2	1	1	1	5.1	33.424	0.23861	0	0
E9PDR5 IBTK	5	1	1	1	0.9	127.23	0.05937	0	0
M0R151 KLK11 TLSP	4	1	1	1	6.4	11.979	0.32659	0	0
F8WDA1 PRKAG2	5	1	1	1	5.2	24.639	0.00621	0	0
A0A090 ACCN3 hCG_	4	1	1	1	3	58.905	0.03885	0	0
Q9UJC5 SH3BGR2 F/	1	1	1	1	13.1	12.326	1.2785	4832300	966460
Q9ULL5 PRR12 KIAA1	1	1	1	1	0.6	211.04	0.06382	0	0
Q9UPZ6 THSD7A KIA/	1	1	1	1	1	185.36	0.77986	0	0
Q86VX4 SMC3	3	1	1	1	0.8	141.51	0.59567	0	0
E7ETR0 RUVBL1	5	1	1	1	3.5	34.782	0.06945	0	0
H0Y3R0 MYO5A	17	1	1	1	4.1	30.727	0.0038	0	0
A0A024 SPIN hCG_29	2	1	1	1	3.8	29.6	0.07552	3945700	263040
A0A024 GTF3A hCG_	1	1	1	1	11.2	46.045	0.02342	0	0
D6R904 TPM3	11	3	3	3	37.9	11.017	1.4229	28939000	5E+06
V9GZ54 MSN	4	1	1	1	15.5	12.123	0.68453	52242000	6E+06
A0ZTA0 TPAP	1	1	1	1	2.4	71.826	0.01682	0	0
A1A508 PRSS3	6	2	1	1	8.5	26.487	8.8827	96242000	1E+07
A4D1Y7 LOC401324 t	1	1	1	1	15.5	15.577	0.36326	0	0
C9JM33 IFNAR2	2	1	1	1	7.4	23.485	0.0555	0	0
D6RIA1 ZNF451	1	1	1	1	17.7	18.042	0.02924	0	0

Q05DR4 CALD1	11	2	2	2	17	17.766	0.25594	10767000	1E+06
E9PI01 L3MBTL3	1	1	1	1	14.1	17.804	0.05091	339230000	4E+07
F8WBX5 SCIN	1	1	1	1	16.7	11.714	0.13183	0	0
H0YBY2 MSR1	1	1	1	1	13.4	15.92	0.18311	0	0
H0YFP6 PMFBP1	1	1	1	1	3.4	58.725	0.0946	33471000	1E+06
H0YGJ0 ACSF3	1	1	1	1	9.4	11.959	1.5414	0	0
L0R512 NCAM2	1	1	1	1	21.4	4.7347	0.40463	54399000	2E+07
L8E8P2 NAP1L2	1	1	1	1	16.7	6.479	0.29373	0	0
Q9BZU2 hCG_1990398	1	1	1	1	34.8	5.7908	0.07579	0	0
Q9UHS1 hCG_1747076	1	1	1	1	24.5	5.5194	0.06835	0	0

Related Manuscript File- Table 2

Fasta headers	GeneName	Number of proteins	Peptides	Razor + unique peptides	Unique peptides	Sequence coverage [%]	Mol. weight [kDa]	Score	Intensity	iBAQ	
Q5JTZ9	AARS2	AARS	2	38	38	38	56.2	107.34	323	1E+10	2E+08
A0A7I2V5	HNRNPA1	19	1	1	1	13.3	8.6005	0.62	5E+06	8E+05	
K7EKN5	PALM3	4	1	1	1	1.8	65.289	0.2	0	0	
H0YNA0	ANXA2	16	1	1	1	13.9	9.0581	0.01	3E+06	6E+05	
O14513	NCKAP5 ERH	1	1	1	1	0.5	208.53	0.12	0	0	
O14744	PRMT5 HRM	15	28	28	28	63.6	72.683	144	4E+09	1E+08	
F8WAR6	KIF3C	7	1	1	1	2.4	75.829	0.14	0	0	
H0YAN8	ARHGEF10	3	1	1	1	1.2	109.65	0.02	0	0	
A0A0S2Z5	LSM1	2	1	1	1	9.8	15.179	0.16	0	0	
H7C3F9	ARPC2	4	1	1	1	6.7	18.654	0.58	5E+06	5E+05	
O43506	ADAM20	2	1	1	1	1.8	81.602	0.04	0	0	
E1CKY7	PPP1R12B sr	2	1	1	1	6.5	21.367	0.03	0	0	
H0Y483	PFKFB3	22	1	1	1	7	16.466	0.1	7E+06	6E+05	
A0A024R6	DNAJA2 hCC	2	1	1	1	6.1	45.745	0.12	1E+07	6E+05	
Q86YU9	FLJ00414	3	1	1	1	0.7	123.5	0.19	0	0	
O75054	IGSF3 EW13 h	1	1	1	1	1.1	135.19	0.02	0	0	
A0A804HI	FLNB	7	8	6	5	4.6	243.09	5.29	4E+07	4E+05	
C9JIR6	PPM1B	12	8	8	8	31	41.83	25.3	3E+08	1E+07	
O95025	SEMA3D UN	1	1	1	1	1.7	89.65	0.18	0	0	
H0YBB4	UBR5	4	1	1	1	11.3	20.474	0.53	6E+06	1E+06	
O43698	ZFS-6	18	1	1	1	8.1	11.514	0.46	0	0	
O95969	SCGB1D2 LIF	1	1	1	1	14.4	9.925	0.01	0	0	
P01859	IGHG2	6	1	1	1	2.8	35.9	0.72	1E+07	1E+06	
P03952	KLKB1 KLK3	3	1	1	1	1.4	71.369	0.11	3E+08	7E+06	
P05109	S100A8 CAG	1	1	1	1	11.8	10.834	0.01	3E+06	4E+05	
P05141	SLC25A5 AA	9	5	5	2	18.1	32.852	83.3	2E+08	1E+07	
Q0QEN7	ATP5B	3	1	1	1	3.4	48.113	0.29	5E+06	2E+05	
Q5U077	HEL-S-281 L	3	1	1	1	2.7	36.638	0.69	5E+06	3E+05	
Q8TBA7	HSP90AA1	14	8	4	4	13	73.826	0.72	8E+07	3E+06	
P08238	HSP90AB1 H	27	10	10	6	16.3	83.263	7.02	5E+08	1E+07	
A0A0S2Z3	FCGR3A	5	1	1	1	3.4	22.696	0.05	0	0	
P0C7P3	SLFN14	1	1	1	1	1.3	103.91	0.62	0	0	
P0CG39	POTEJ	1	3	1	1	4.5	117.39	0.32	2E+07	4E+05	
A0A0G2JI	HSPA1B	19	26	26	14	55	70.108	323	8E+09	2E+08	
A0A7P0TE	HSPA5	5	7	6	6	15.6	69.01	7.14	8E+07	3E+06	
P11142	HSPA8 HSC7	25	22	20	6	41.8	70.897	225	3E+09	1E+08	
Q6I9V5	SLC25A6 hCC	3	4	1	1	14.8	32.866	0.33	5E+06	3E+05	
B3GN61	CDH1	6	1	1	1	4.7	83.822	0.07	6E+07	2E+06	
A0A090NI	ERP70 tcag7	4	1	1	1	1.6	72.932	0.16	0	0	
J3QLE5	SNRPN	11	1	1	1	4.7	17.546	0.06	3E+06	3E+05	
C9K028	NME1	10	1	1	1	20	6.5266	0.14	6E+06	6E+06	
A4D2A4	ZNF3 tcag7.6	6	1	1	1	3.4	44.163	0	0	0	
P17987	TCP1 CCT1 C	10	7	7	7	13.8	60.343	13	8E+07	3E+06	
Q6PJ54	NEB	4	1	1	1	3.9	39.435	0.12	0	0	
Q5QPQ9	CDK11A	23	1	1	1	27	4.5582	0.02	0	0	
P21333	FLNA FLN FL	19	26	26	22	15.6	280.74	83.9	6E+08	5E+06	
A0A8C8KI	ITGA6	2	1	1	1	0.8	117.8	0.49	0	0	
E9PQ96	RPS3	11	2	2	2	24.2	10.279	1.25	2E+07	3E+06	
Q8NHX6	HCC5	6	1	1	1	1.8	75.786	0.54	3E+06	76561	
K7EK45	PTBP1	5	1	1	1	9	27.114	10.2	2E+06	3E+05	
F8VPD4	CAD	5	3	3	3	2.3	236.02	2.2	4E+07	4E+05	
P29373	CRABP2	1	1	1	1	10.9	15.693	0	0	0	
B4DEW2	MAPK4	3	1	1	1	5.3	42.093	0.04	0	0	
P31689	DNAJA1 DN	3	2	2	2	6.5	44.868	0.18	3E+07	1E+06	
E5RGH4	HNRNPH1	13	1	1	1	17	11.181	46.9	1E+08	2E+07	

F5GXD8	STIP1	5	1	1	1	6.5	15.637	0	9E+06	9E+05
O14992	HS24/p52	11	3	3	2	9.3	52.367	25.6	4E+07	1E+06
P42166	TMPO LAP2	1	1	1	1	1.3	75.491	0.03	0	0
P42338	PIK3CB PIK3C	1	1	1	1	1.4	122.76	0.11	7E+07	1E+06
F5GX32	RAD52	6	1	1	1	2.3	32.507	0.64	0	0
A0A6I8PR	CAPZB	8	1	1	1	23.7	6.7564	0.94	9E+06	3E+06
A0A0U1R	FASN	2	1	1	1	0.4	273.2	0.09	0	0
B4DUR8	CCT3	6	4	4	4	9	55.674	3.54	8E+07	3E+06
A0A384M	TUFM hCG_1	2	1	1	1	4.2	49.874	0.57	1E+07	5E+05
H3BN04	MTHFS	2	1	1	1	22.7	4.9367	0.42	0	0
Q7Z759	CCT8 hCG_1	5	7	7	7	15.1	54.106	7.1	1E+08	3E+06
A0A8I5KY	FMR1	33	1	1	1	2.3	45.683	0.3	0	0
F6U4U2	NEK2	2	1	1	1	7.2	45.216	0.15	0	0
M0R0Y6	HNRNPM	8	1	1	1	16.2	10.739	2.06	0	0
Q5T8R3	SLC16A1	7	1	1	1	4.1	31.689	0.4	4E+06	6E+05
Q9UDQ3	WUGSC:H_D	3	1	1	1	1.6	84.396	0.1	0	0
A0A024R	HSPA2 hCG_	28	9	3	3	16.6	70.02	0.68	5E+07	1E+06
D3DV75	ADAR hCG_1	9	1	1	1	2.5	98.744	-0	5E+06	1E+05
J3QKZ9	EIF4A1	14	1	1	1	8.2	14.595	0.05	3E+06	3E+05
E5RIP1	RPS20	4	1	1	1	23.9	5.4292	0.01	6E+06	3E+06
G3V4F7	SRP54	5	1	1	1	4.3	48.67	0.01	0	0
A0A0A0M	ACTR3C	12	1	1	1	5.8	21.312	11	4E+06	5E+05
Q5T6W2	HNRNPK	6	2	2	2	7.1	41.807	1.14	2E+07	9E+05
M0QX76	RPS16	6	1	1	1	20	5.5575	7.65	3E+06	7E+05
G9K388	YWHAE/FAM	30	2	2	2	5.5	41.224	4.03	1E+07	8E+05
P62318	SNRPD3	1	2	2	2	15.1	13.916	2.84	4E+07	7E+06
H0YFC6	RAN	6	1	1	1	10.7	11.568	0.27	0	0
A0A804G	ACTG1	11	16	1	1	60.4	40.904	59.4	7E+08	4E+07
P68032	ACTC1 ACTC	19	8	1	1	19.9	42.019	0.98	1E+09	6E+07
F5H5D3	TUBA1C	43	10	10	10	27.4	57.73	249	3E+08	1E+07
K7EKP9	DLG4	14	1	1	1	4.5	21.095	0	0	0
F8VQ14	CCT2	6	8	8	8	26	44.812	38.9	2E+08	8E+06
Q6PII6	TMF1	5	1	1	1	2.4	58.381	0.37	0	0
G3V279	ERH	3	1	1	1	15.5	8.2033	0.01	4E+06	9E+05
Q01804	OTUD4 HIN-	1	1	1	1	1.7	124.04	0.06	3E+06	49556
Q6PJM5	CALD1	12	2	2	2	9.2	36.692	0.66	3E+07	2E+06
A0A0A0M	PRDX1	5	1	1	1	10.3	10.676	0.35	1E+07	2E+06
Q96AS4	SPIRE1	4	1	1	1	7.6	14.952	0.17	0	0
A0A8I5KU	LRRC74A	4	1	1	1	5.4	43.241	0	2E+08	9E+06
C9J9F8	CFAP210	3	1	1	1	7.6	19.547	0.27	3E+07	4E+06
A0A0U1R	ROCK1	3	1	1	1	1.8	133.07	0	0	0
H7C2L1	ITGA9	2	1	1	1	14.5	8.0801	0.06	0	0
E5RG28	DPYS	2	1	1	1	12	16.242	0	6E+07	9E+06
C9JYU3	LRRC32	3	1	1	1	11.8	20.153	0.01	1E+07	2E+06
A0A024R	STK38 hCG_	6	8	8	7	27.5	54.19	89	3E+08	1E+07
H3BV80	RNPS1	7	1	1	1	7.6	24.561	0.01	0	0
Q15772	SPEG APEG1	1	1	1	1	0.2	354.28	0.04	0	0
Q17RC7	EXOC3L4 C1	1	1	1	1	1.1	79.895	0.16	0	0
A0A024R	MTERFD3 hC	2	1	1	1	5.2	44.413	0.15	0	0
Q4G0P3	HYDIN HYDI	1	1	1	1	0.3	575.89	0	0	0
Q562R1	ACTBL2	1	3	1	1	11.2	42.003	0.15	0	0
Q58FF8	HSP90AB2P	1	2	1	1	7.1	44.348	2.03	1E+07	7E+05
A0A024R	RP11-98F14.	4	1	1	1	3.1	33.783	0.33	0	0
Q8N9U5	ZBTB41 hCG	2	1	1	1	1.6	49.624	0.01	0	0
Q5T7W0	ZNF618 KIA/	1	1	1	1	1.9	104.95	0.03	0	0
A0A024R	KIAA0649 hC	2	1	1	1	0.8	127.32	0.01	0	0
A1L0S9	CROCC	4	1	1	1	1	129.52	0.47	0	0
Q5HYB0	DKFZp686P1	7	1	1	1	0.7	114.49	0.13	2E+07	4E+05
H7C3M6	FAM78B	3	1	1	1	7.5	29	0.01	0	0

A0A024R5	ATAD2 hCG_	2	1	1	1	1.3	158.55	0.07	0	0
Q3LIB9	Nbla00144	3	1	1	1	1.6	50.817	0.08	0	0
D3DUW2	MGC24039 f	3	1	1	1	5.3	84.622	0.21	0	0
A0A8I5KT	USP53	3	1	1	1	2.5	86.533	0.01	0	0
F8WE42	NOL8	6	1	1	1	1.3	108.45	0.03	0	0
Q9H8S9	MOB1A C2o	2	1	1	1	5.1	25.079	0.01	0	0
H0YB11	DGLUCY	9	1	1	1	6.5	29.36	0.02	0	0
Q7Z3J3	RGPD4 RGP4	1	1	1	1	1.2	197.29	0.06	3E+08	3E+06
Q86SU0	ILDR1	1	1	1	1	2.7	62.814	0.08	0	0
Q86W68	USP37	3	1	1	1	5.3	33.39	0.45	0	0
Q86W42	THOC6 WDF	1	1	1	1	5.9	37.535	0.04	0	0
H3BQ67	LONP2	6	1	1	1	8.3	18.931	0.18	0	0
A7E2D6	NAV2	2	1	1	1	0.5	261.72	0.04	0	0
Q8IZS2	MRDS1	4	1	1	1	7	20.993	0.06	0	0
Q8IZU8	DSEL C1orf	1	1	1	1	1.2	139.24	0.03	0	0
Q149N2	DOCK4	6	1	1	1	0.8	127.53	0.06	5E+06	83279
Q8N365	CIART C1orf	1	1	1	1	2.3	41.442	0.06	0	0
F5H777	CCDC82	11	1	1	1	8.4	30.485	0.17	9E+06	9E+05
Q8N8Z8	ZNF441	1	1	1	1	3.5	80.135	0.06	3E+08	1E+07
Q8N9C0	IGSF22	1	1	1	1	2.1	100.4	0.01	0	0
H0YFG0	ATG16L2	5	1	1	1	4.4	34.408	0.03	0	0
F5GXG5	EXPH5	6	1	1	1	1.3	133.5	0.59	0	0
A0A023HI	TET1 CXXC6	2	1	1	1	0.7	142.8	0	0	0
B2RU30	DEFB105B	3	1	1	1	17.9	8.9234	0.08	8E+06	4E+06
B2CM70	GSDML	3	1	1	1	3.5	45.893	0.03	0	0
F8VNT5	PIP4K2C	5	1	1	1	6.2	17.952	0.07	0	0
D3DP61	KM-HN-1 hC	5	1	1	1	2.8	68.046	0.54	0	0
A0A024RI	PSPC1 hCG_	3	1	1	1	3.1	45.57	0.12	0	0
B2CML4	APOBEC3DE	2	1	1	1	2.6	46.598	0.04	0	0
A0A8Q3S	INTS4	18	1	1	1	3.6	39.962	0.01	0	0
H0YHI9	MYRFL	3	1	1	1	9.3	42.441	0.23	0	0
A0A6I8PT	ODAD1 FLJ3	6	2	2	2	2.8	66.813	0.03	2E+06	57473
Q96PG2	MS4A10 CD2	1	1	1	1	3.4	29.747	0.72	2E+08	1E+07
H7C1T5	FLACC1	3	1	1	1	7.3	14.909	0.13	0	0
D3DR32	MPHOSPH1	2	1	1	1	0.7	206.16	0.14	0	0
A0A075B7	ATAD5	2	1	1	1	2.3	137.99	0.08	0	0
A0A024R2	VPS13A hCG	3	1	1	1	0.4	339.18	0.11	0	0
Q6IBT3	CCT7	6	5	5	5	15.1	59.328	7.73	9E+07	3E+06
Q9BPX7	C7orf25	2	1	1	1	6.2	46.45	0.06	5E+07	3E+06
A0A024R0	WDR77 hCG	4	8	8	8	36.5	36.724	304	1E+09	1E+08
Q9BRS2	RIOK1 RIO1	2	2	2	2	3.7	65.582	0.42	3E+07	1E+06
Q9BVM2	DPCD	1	1	1	1	6.4	23.239	0.07	0	0
B5MCQ4	DGCR6L	2	1	1	1	3.8	20.225	0.64	3E+06	3E+05
Q9BYX7	POTEKP ACT	1	4	1	1	15.2	42.016	0.02	9E+07	5E+06
Q9H0F5	RNF38	1	1	1	1	1.4	57.595	0.18	0	0
Q9H444	CHMP4B C2C	1	1	1	1	6.7	24.95	0.03	0	0
Q9H4Z2	ZNF335	1	1	1	1	0.5	144.89	0.06	0	0
A0A0A0M	NOL6	3	1	1	1	0.8	112.16	0.09	0	0
H0Y9I8	CPLANE1	3	1	1	1	1	255.48	0.01	6E+06	63141
Q9H8M5	CNNM2 ACE	1	1	1	1	1.7	96.622	0.09	0	0
Q3LIF4	Nbla00271	3	1	1	1	3.7	57.025	0.28	0	0
A0A5J6D5	CNGB3	3	1	1	1	3.7	64.999	0.05	0	0
A0A0C4D	ACOXL	4	1	1	1	3.1	40.503	1.89	0	0
Q9NX36	DNAJC28 C2	1	1	1	1	3.4	45.805	0.01	8E+06	3E+05
I3L0K0	DUS2	12	1	1	1	35.6	8.2518	0.02	0	0
Q9NXL2	ARHGEF38	1	1	1	1	2.6	89.077	0.04	0	0
Q9NY74	ETAA1 ETAA	1	1	1	1	1	103.44	0.4	1E+07	2E+05
MOR151	KLK11 TLSP	4	1	1	1	6.4	11.979	0.8	0	0
B1AKM8	PISD	6	1	1	1	11.8	26.282	0	3E+07	2E+06

F8WDA1	PRKAG2	5	1	1	1	5.2	24.639	0.67	0	0
C9IY64	SFMBT1	3	1	1	1	11.6	13.775	0.05	0	0
H0YKQ8	SLC12A6	9	1	1	1	2.1	103.82	0.05	0	0
Q9UIU6	SIX4	1	1	1	1	1.7	82.932	0.11	0	0
Q9UJC5	SH3BGRL2 F,	1	1	1	1	13.1	12.326	0.83	5E+06	1E+06
Q9UKA4	AKAP11 AKA	1	1	1	1	0.9	210.51	0.03	0	0
Q9UNE7	STUB1 CHIP	3	2	2	2	7.6	34.856	0.1	2E+07	8E+05
Q86VX4	SMC3	3	1	1	1	0.8	141.51	1.09	0	0
K7EK06	FARSA	4	1	1	1	6.5	18.389	0.76	3E+06	5E+05
H0YGN4	STK38L	3	2	1	1	26.4	9.9724	0.33	1E+07	3E+06
Q9Y490	TLN1 KIAA1C	2	1	1	1	0.7	269.76	0.33	0	0
J3QL92	MTCL1	4	1	1	1	2.6	55.249	0.25	0	0
E5RIY7	ZNF706	5	1	1	1	23.8	4.6133	0.64	0	0
C9JQE8	NCOR2	8	1	1	1	0.8	160.39	0.03	0	0
A0A044P\	MISP3	1	1	1	1	3.3	39.004	0.14	0	0
A0A0A0M	SLC4A7	1	1	1	1	2.2	134.57	0.05	1E+07	2E+05
A0A2R8Y\	ANAPC1	1	1	1	1	4.2	29.315	0.15	0	0
Q92967	ZF6	4	1	1	1	3	49.12	0.02	0	0
A0A669K\	CRELD1	1	1	1	1	13.1	15.433	0.02	2E+06	3E+05
A0A7I2V2	HSPD1	19	3	3	3	11.6	57.758	2.14	3E+07	9E+05
V9GZ54	MSN	4	1	1	1	15.5	12.123	0.09	3E+07	3E+06
A1A508	PRSS3	6	2	1	1	8.5	26.487	0.02	2E+08	2E+07
D6W5R0	hCG_174180	2	1	1	1	0.7	147.28	0.24	0	0
Q9Y3Z0	DKFZp564J0.	2	1	1	1	7.1	13.866	0.18	0	0
C9J590	REPIN1	1	1	1	1	22.8	15.306	0.08	2E+08	5E+07
D6RA62	RPS6KA2	1	1	1	1	6.4	16.39	0.44	0	0
D6RBT2	FAM47E	1	1	1	1	8.3	15.265	0.14	0	0
E9PK54	HSPA8	1	11	1	1	72.1	19.955	0.43	3E+07	2E+06
F8VNX9	RAB3IP	1	1	1	1	5.5	17.845	0.22	0	0
F8VR53	LZTFL1	1	1	1	1	20.9	5.1149	0	0	0
H0YDR1	SULF1	1	1	1	1	5.3	15.569	0.29	0	0
H0YDW3	APOL3	1	1	1	1	17.8	5.1599	0.13	0	0
H0YGJ0	ACSF3	1	1	1	1	9.4	11.959	1.54	0	0
H3BTZ8	CIAPIN1	1	1	1	1	18.1	7.8627	0.13	0	0
H7BZA6	NOSTRIN	1	1	1	1	45	2.3366	0.01	0	NaN
K7EN51	NEDD4L	1	1	1	1	25.7	4.3432	0.18	0	NaN
L0R4X4	EHBP1L1	1	1	1	1	9.9	14.07	0.4	2E+07	3E+06
L0R5A4	DNAH11	1	1	1	1	24.4	5.0093	0.03	0	0
Q7Z6E5	DKFZp779G2	1	1	1	1	0.9	115.71	0.03	0	0
Q8WWP3	GYPA	1	1	1	1	29.5	14.343	0.05	3E+06	4E+05
Q9BZU2	hCG_199039	1	1	1	1	34.8	5.7908	0.13	0	0